# When Sample Selection Bias Precipitates Model Collapse

**Xinbao Qiao** [1 2 3]  **Xianglong Du** [4 5]  **Wei Liu** [4 5]  **Jingqi Zhang** [6]  **Peihua Mai** [1 2 6]  **Meng Zhang** [7]  **Yan Pang** [1 2 6]

## Abstract

The proliferation of recursive training on synthetic data can alleviate data scarcity but risks model collapse, where repeated training erodes distributional tails and homogenizes outputs. Data selection is widely viewed as a remedy, yet its reliability depends critically on the reference distribution used by the verifier. We show that in low-resource verification regimes, where each verifier observes only a small, fragmented, and biased slice of the target manifold, selection itself becomes biased. This situation naturally arises in low-resource data silos such as healthcare consortia or proprietary financial institutions, where raw data cannot be pooled and local references are inherently incomplete. As a result, selection preferentially retains samples aligned with the local manifold while pruning globally relevant tail modes, turning from a safeguard against collapse into a mechanism that precipitates it. We theoretically prove that such siloed selection accelerates collapse and induces power-law diversity decay. As an initial mitigation, we construct Wasserstein proxy references from multiple silos without sharing raw data. Empirical results confirm that local-reference selection fails on skewed distributions, whereas collaborative proxy references mitigate diversity degradation, suggesting that recursive synthetic-data pipelines require particular caution when real-data coverage is fragmented or scarce.

This work was done in part while Xinbao Qiao was with the National University of Singapore (Chongqing) Research Institute, Zhejiang University, and The Chinese University of Hong Kong. [1]National University of Singapore (Chongqing) Research Institute [2]Chongqing Key Laboratory of Trusted Perception and Interaction Technology for Intelligent and Connected Vehicles, Chongqing, China [3]The Chinese University of Hong Kong [4]State Key Laboratory of Intelligent Vehicle Safety Technology, Chongqing, China [5]CHONGQING CHANGAN AUTOMOBILE Co., Ltd, Chongqing, China [6]National University of Singapore [7]Zhejiang University. Correspondence to: Yan Pang <james-pang@nus.edu.sg>, Meng Zhang <mengzhang@intl.zju.edu.cn>, Xinbao Qiao <xinbaoqiao@cuhk.edu.hk,xinbaoqiao@zju.edu.cn>.

*Proceedings of the 43rd International Conference on Machine Learning*, Seoul, South Korea. PMLR 306, 2026. Copyright 2026 by the author(s).

## 1. Introduction

The digital ecosystem is currently saturated with indistinguishable synthetic data, which is subsequently harvested for the training of future models. When generative models are recursively trained on the outputs of their predecessors, they undergo a degenerative process that is not merely a stagnation of quality, but an active decay of statistical fidelity. This phenomenon, observed empirically (Bertrand et al., 2024) and described theoretically (Kazdan et al., 2025), is widely termed *model collapse* in prior works.[1] Qualitatively, it describes a self-consuming loop wherein subsequent model generations trained on synthetic data progressively dissociate from the tails of the true underlying distribution, converging toward a distorted representation of reality as probability density functions narrow. Quantitatively, as the number of generations increases, the variance of the distribution shrinks, and the Wasserstein distance between the synthetic and true distributions diverges (Kazdan et al., 2025; Shidani et al., 2025). The implications of these self-consuming loops are profound, ranging from loss of diversity (Zhou et al., 2025; Wyllie et al., 2024; Taori & Hashimoto, 2023) to potential training instability and failures (Bertrand et al., 2024; Alemohammad et al., 2024).

The current consensus in the research community underscores the critical importance of data selection (Feng et al., 2025; Dohmatob et al., 2025a; Fu et al., 2025; Yi et al., 2026): the active filtering of synthetic data to eliminate low-quality samples. Intuitively, if a perfect verifier (modeled as an external oracle or high-quality filter capable of distinguishing between synthetic data that preserves the true distribution and synthetic data that introduces noise) exists, recursive training can be stabilized. Indeed, under such ideal conditions, performance could potentially be optimized to surpass that of models trained solely on raw data (Shi et al., 2025). Therefore, numerous data selection methodologies have been developed to mitigate model collapse. For instance, in the domain of language models, Feng et al. (2025) utilize metrics such as the ROUGE score to quantify the alignment between generated outputs and ground-truth data, retaining the samples with the highest fidelity.

---

[1]While the literature on model collapse encompasses a variety of interpretations, we focus on the definition of *Collapsing Variance* (Shumailov et al., 2024; Kazdan et al., 2025; Schaeffer et al., 2025). See Appendix A for a review of related works.

However, consider low-resource settings such as isolated institutions, including hospitals and banks, that operate under strict privacy regulations. In such data-scarce and siloed environments, these entities often resort to synthetic data for recursive self-improvement in order to sustain model performance (Huang et al., 2023). Consequently, their verification mechanisms are inherently local, operating only on partial and biased slices of the global distribution. When a generative model is vetted against such restricted criteria, the selected synthetic data fails to capture global diversity. Instead, it reflects the verifier's limited local prior (Yi et al., 2026). While prior theoretical works (Ferbach et al., 2024; Wei & Zhang, 2025) have demonstrated related pathologies from other perspectives, such as the vanishing variance induced by preference-based data curation, a critical gap remains in understanding the structural impact of data silos:

> **(Q1)** *How does model collapse manifest when sample selection is confined to data-scarce environments?*

In response to **Q1**, Section 3 analyzes biased sample selection dynamics, focusing on data-scarce and low-resource environments increasingly reliant on synthetic augmentation. Combining theoretical intuition with empirical findings in Subsection 3.1, we demonstrate that biased selection mechanisms governed by local priors function as inherently biased filters that are blind to the global manifold, ultimately resulting in a loss of diversity. This process is similar to the diversity contraction observed in data curation based on human preferences (Ferbach et al., 2024; Wei & Zhang, 2025), yet it is passively driven by inherent constraints. Far from ensuring stability, Subsection 3.2 shows that insular selection prunes the diversity essential for recursive training, steering the model toward collapsed data diversity at an asymptotic power-law rate. Although Subsection 3.3 shows that equipping the verifier with global ground truth offers a theoretical remedy, intrinsic data scarcity and low-resource constraints preclude this solution, presenting a dilemma:

> **(Q2)** *How can we verify synthetic data against a global reference distribution that no single entity possesses, while operating under data scarcity?*

In response to **Q2**, we bridge this gap by proposing a collaborative framework inspired by current work (Li et al., 2024). Although selection bias is well documented (Ferbach et al., 2024), effective countermeasures for data silos are notably absent. By utilizing the properties of Wasserstein geometry in Subsection 4.2, we coordinate multiple data silos without direct data exchange to compute proxies: geodesic interpolations in Subsection 4.3 or Wasserstein barycenters in Subsection 4.4. These proxies serve as a collective reference, enabling multiple parties to score synthetic data, rather than relying on a single biased silo. Section 6 demonstrates a significant reduction in model collapse within low-resource communities, especially those characterized by data silos.

## 2. Preliminaries

**Self-Consuming Training Loops.** Following standard formulations (Alemohammad et al., 2024; Shumailov et al., 2023), we define a self-consuming loop as an iterative process where a model $\mathcal{M}_t$ is trained on a dataset $\mathbf{X}_t$ derived from the synthetic output $\hat{\mathbf{X}}_t$ of its predecessor $\mathcal{M}_{t-1}$. The literature categorizes such analyses into three paradigms:

- *Replace Paradigm.* The training set consists exclusively of new synthetic samples generated in the preceding round ($\mathbf{X}_t = \hat{\mathbf{X}}_t$). Existing analysis (Proposition 1) demonstrates that this paradigm induces catastrophic *model collapse*, characterized by progressive variance shrinkage and tail information loss (Alemohammad et al., 2024; Shumailov et al., 2023).
- *Accumulate Paradigm.* Prior literature suggests augmenting the initial real data $\mathbf{X}_0$ with all subsequent generations ($\mathbf{X}_t = \mathbf{X}_{t-1} \cup \hat{\mathbf{X}}_t$). Existing analysis (Proposition 2) demonstrates that this paradigm prevents variance divergence, thereby ensuring recursive stability (Kazdan et al., 2025; Dey & Donoho, 2024).
- *Accumulate-Subsample Paradigm.* To mitigate the prohibitive costs of full accumulation, this method utilizes a fixed-size subset sampled from the accumulated pool (Shi et al., 2025; Kazdan et al., 2025; Dey & Donoho, 2024). Empirically, this alternative mitigates model collapse while satisfying computational constraints, particularly when paired with robust data selection mechanisms (Shi et al., 2025).

**Multivariate Gaussian Analysis Framework.** We adopt the theoretical analysis framework established in the current literature (Shumailov et al., 2024; Alemohammad et al., 2024; Bertrand et al., 2024; Kazdan et al., 2025), distinguishing between two paradigms: *Replace* and *Accumulate*.

*The Replace Paradigm.* In this framework, the functional approximation step involves generating $n$ synthetic data points using the fitted parameters $\mathbf{X}_{i,t} \sim \mathcal{N}(\boldsymbol{\mu}_{t-1}, \boldsymbol{\Sigma}_{t-1})$ for $t \in [0, T]$; the corresponding distribution is hereafter referred to as $\mathcal{N}_t$. The recursive process is thus defined as:

$$\text{Sampling: } \mathbf{X}_{i,t} = \boldsymbol{\mu}_{t-1} + \boldsymbol{\Sigma}_{t-1}^{1/2}\mathbf{z}_{i,t}$$

$$\text{Learning: } \begin{cases} \boldsymbol{\mu}_t = \dfrac{1}{n}\sum_{i=1}^{n}\mathbf{X}_{i,t}, \\ \boldsymbol{\Sigma}_t = \dfrac{1}{n-1}\sum_{i=1}^{n}(\mathbf{X}_{i,t} - \boldsymbol{\mu}_t)^{\otimes 2}, \end{cases} \quad (1)$$

where $\mathbf{z}_{i,t} \sim \mathcal{N}(\mathbf{0}, \mathbf{I}_d)$ represents the stochastic term, and we define the product $(\mathbf{X}_{i,t}-\boldsymbol{\mu}_t)(\mathbf{X}_{i,t}-\boldsymbol{\mu}_t)^\top \triangleq (\mathbf{X}_{i,t}-\boldsymbol{\mu}_t)^{\otimes 2}$. Note that in the case of maximum likelihood estimation, the result is instead a biased variance estimator (Shumailov et al., 2024). This recursive dependency precipitates model collapse, which is characterized by (Shumailov et al., 2024):

**Proposition 1** (**Replace Paradigm**, Shumailov et al.)**.**
*Under the Replace paradigm, as iteration $t \to \infty$, the estimator statistics satisfy:*

$$\Sigma_t \xrightarrow{a.s.} \mathbf{0}. \tag{2}$$

*Let $\mathbb{W}_2$ denote the Wasserstein-2 distance. We have*

$$\mathbb{E}[\mathbb{W}_2^2(\mathcal{N}_t, \mathcal{N}_0)] \to \infty. \tag{3}$$

Please refer to Subsection B.2 for the comprehensive proof. Proposition 1 indicates that the fitted distribution collapses to a point mass (Dirac delta), marking the elimination of diversity. The diverging Wasserstein distance reflects the loss of information compared to the real data manifold.

*The Accumulate Paradigm* aggregates samples from all prior generations. For clarity, we designate the parameters as $\bar{\boldsymbol{\mu}}$ and $\bar{\boldsymbol{\Sigma}}$. Kazdan et al. (2025) characterized the process as:

**Sampling:** $\mathbf{X}_{i,t} = \bar{\boldsymbol{\mu}}_{t-1} + \bar{\boldsymbol{\Sigma}}_{t-1}^{1/2} \mathbf{z}_{i,t}$,

**Learning:**
$$\begin{cases} \bar{\boldsymbol{\mu}}_t = \dfrac{1}{(t+1)n} \displaystyle\sum_{\tau=0}^{t} \sum_{i=1}^{n} \mathbf{X}_{i,\tau}, \\ \bar{\boldsymbol{\Sigma}}_t = \dfrac{1}{(t+1)n-1} \displaystyle\sum_{\tau=0}^{t} \sum_{i=1}^{n} (\mathbf{X}_{i,\tau} - \bar{\boldsymbol{\mu}}_t)^{\otimes 2}. \end{cases} \tag{4}$$

**Proposition 2** (**Accumulate Paradigm**, Kazdan et al.)**.**
*Under the Accumulate paradigm, as iteration $t \to \infty$, we have the following convergence in expectation:*

$$\mathbb{E}[(\bar{\boldsymbol{\mu}}_t - \bar{\boldsymbol{\mu}}_0)^2] \to (1 - \alpha_n)\bar{\boldsymbol{\Sigma}}_0, \tag{5}$$

$$\mathbb{E}[\bar{\boldsymbol{\Sigma}}_t] \to \alpha_n \bar{\boldsymbol{\Sigma}}_0, \quad \text{where } \alpha_n = \frac{\sin(\pi/\sqrt{n})}{\pi/\sqrt{n}}. \tag{6}$$

*Consequently, the Wasserstein distance stabilizes at:*

$$\mathbb{E}[\mathbb{W}_2^2(\mathcal{N}_t, \mathcal{N}_0)] \to 2(1 - \sqrt{\alpha_n})\mathrm{Tr}(\bar{\boldsymbol{\Sigma}}_0). \tag{7}$$

Please refer to Subsection B.3 for the comprehensive proof. Proposition 2 establishes that data accumulation precludes collapse. Now we formally define the Wasserstein distance.

**Wasserstein Distance.** Let $\mathcal{P}_p(\mathbb{R}^d)$ denote the space of probability measures with finite $p$-th moments on the metric space $\mathbb{R}^d$. For any $\mathcal{P}, \mathcal{Q} \in \mathcal{P}_p$, the $p$-$\mathbb{W}_p$ distance reads:

$$\mathbb{W}_p(\mathcal{P}, \mathcal{Q}) = \left( \inf_{\pi \in \Pi(\mathcal{P}, \mathcal{Q})} \mathbb{E}_{(\mathbf{x}, \mathbf{y}) \sim \pi}[d^p(\mathbf{x}, \mathbf{y})] \right)^{1/p}, \tag{8}$$

where $\Pi(\mathcal{P}, \mathcal{Q})$ denotes the set of couplings with marginals $\mathcal{P}, \mathcal{Q}$. The ground metric $d(\mathbf{x}, \mathbf{y})$ defines the distance between samples $\mathbf{x}, \mathbf{y} \in \mathbb{R}^d$. We defer specific definitions of $d(\mathbf{x}, \mathbf{y})$ for vision and language to Subsection C.1.

## 3. Theoretical Intuitions

Although Proposition 2 guarantees stability under unbiased accumulation, we show that data selection is double-edged: when the reference is local and fragmented, biased selection precipitates variance collapse (Theorem 1), induces an explicit asymptotic decay rate (Theorem 2), and incurs a downstream Wasserstein discrepancy cost (Theorem 3).

### 3.1. Selection Bias Precipitates Model Collapse

**Assumption 1.** *For analytical tractability, we characterize the selection mechanism via a score function $U(\mathbf{x}) : \mathbb{R}^d \to \mathbb{R}$, which is locally concave around a target state $\mathbf{x} = \mathbf{u}^*$, with initialization lying within the local basin of attraction.*

Assumption 1 serves as a tractable formalism to analyze biased selection mechanisms, encompassing strategies ranging from metric-based data pruning (Shi et al., 2025) (e.g., selecting images closest to the centroid (He et al., 2023) or covariance (Rezaei et al., 2026) of real features (Feng et al., 2025)) in data-scarce and low-resource environments to active preference optimization strategies (e.g., Best-of-N sampling (Gui et al., 2024)). While the biases in these scenarios originate from distinct motivations, characterized respectively by environmental constraints (specifically the restriction to a local target $\mathbf{u}^*$) and intentional preference curation (specifically the restriction to a preferred target $\mathbf{u}^*$), this mathematical abstraction captures their shared core goal: prioritizing samples that are proximal to a preferred ideal.

At generation $t$, we define the bounded selection region $\mathcal{R}_t \subset \mathbb{R}^d$ as a high-utility neighborhood enclosing the target $\mathbf{u}^*$, dynamically calibrated to select the top-$\alpha$ probability mass of the current sampling distribution, where $\alpha \in (0, 1)$ represents the selection ratio, acting as a filtering budget (e.g., selecting the top-$n$ candidates from $N$ generated data implies $\alpha = n/N$). Therefore, the selected data follow a truncated multivariate normal distribution, denoted as $\mathbf{X}_{i,t} \sim \mathcal{TN}(\bar{\boldsymbol{\mu}}_{t-1}, \bar{\boldsymbol{\Sigma}}_{t-1}, \mathcal{R}_t)$. Formally, the Accumulate paradigm with $n$ selected samples is formalized as:

**Sampling:** $\tilde{\mathbf{X}}_{i,t} \sim \mathcal{N}(\bar{\boldsymbol{\mu}}_{t-1}, \bar{\boldsymbol{\Sigma}}_{t-1})$,
**Selecting:** $\mathbf{X}_{i,t} \sim \mathcal{TN}(\bar{\boldsymbol{\mu}}_{t-1}, \bar{\boldsymbol{\Sigma}}_{t-1}, \mathcal{R}_t)$,

$$\text{s.t.} \int_{\mathcal{R}_t} p(\mathbf{x} | \bar{\boldsymbol{\mu}}_{t-1}, \bar{\boldsymbol{\Sigma}}_{t-1}) d\mathbf{x} = \alpha,$$

**Learning:**
$$\begin{cases} \bar{\boldsymbol{\mu}}_t = \dfrac{1}{(t+1)n} \displaystyle\sum_{\tau=0}^{t} \sum_{i=1}^{n} \mathbf{X}_{i,\tau}, \\ \bar{\boldsymbol{\Sigma}}_t = \dfrac{1}{(t+1)n-1} \displaystyle\sum_{\tau=0}^{t} \sum_{i=1}^{n} (\mathbf{X}_{i,\tau} - \bar{\boldsymbol{\mu}}_t)^{\otimes 2}. \end{cases} \tag{9}$$

We demonstrate that biased selection breaks the stability of the Accumulate paradigm, leading to a new form of collapse.

**Theorem 1 (Selection Bias Precipitates Collapse).**
*Consider the Accumulate paradigm with top-$\alpha$ selection toward an ideal $\mathbf{u}^*$. As iteration $t \to \infty$, the estimator statistics behave as follows:*

$$\|\bar{\boldsymbol{\mu}}_t - \mathbf{u}^*\|^2 \xrightarrow{a.s.} 0. \tag{10}$$

$$\bar{\boldsymbol{\Sigma}}_t \xrightarrow{a.s.} \mathbf{0}. \tag{11}$$

*The Wasserstein-2 distance converges to:*

$$\mathbb{E}[\mathbb{W}_2^2(\mathcal{N}_t, \mathcal{N}_0)] \to \|\mathbf{u}^* - \bar{\boldsymbol{\mu}}_0\|^2 + \mathrm{Tr}(\bar{\boldsymbol{\Sigma}}_0) \tag{12}$$

Please refer to Subsection B.4 for the comprehensive proof. Theorem 1 shows that while the mean aligns with the target $\mathbf{u}^*$ and the Wasserstein distance eventually stabilizes, the variance inexorably collapses from a diversity perspective. This result formalizes how local-reference selection can turn fidelity to a silo into diversity loss: although the mean aligns with $\mathbf{u}^*$, the variance needed to represent modes outside the local reference is progressively erased. Furthermore, Theorem 1 resonates with concurrent analyses (Ferbach et al., 2024; Wei & Zhang, 2025), which demonstrate that optimizing for human preferences inadvertently leads to bias amplification. We advance beyond qualitative observations to quantify the collapse rate of variance dissipation.

### 3.2. The Asymptotic Rate of Model Collapse

To explicitly derive the collapse rate of the variance, we standardize the selection process to the isotropic frame. Let $\mathcal{D}_{t-1} \subset \mathbb{R}^d$ denote the *standardized selection region*:

$$\mathcal{D}_{t-1} = \{\mathbf{z} \in \mathbb{R}^d \mid \bar{\boldsymbol{\mu}}_{t-1} + \bar{\boldsymbol{\Sigma}}_{t-1}^{1/2}\mathbf{z} \in \mathcal{R}_t\}. \tag{13}$$

By strict enforcement of the selection ratio, the probability measure of this region satisfies $\int_{\mathcal{D}_{t-1}} \phi_d(\mathbf{z})d\mathbf{z} = \alpha$. Consequently, the retained samples $\mathbf{X}_{i,t}$ can be reparameterized via the truncated variable $\boldsymbol{\eta}_{i,t} \sim \mathcal{TN}(\mathbf{0}, \mathbf{I}_d, \mathcal{D}_{t-1})$ as:

$$\mathbf{X}_{i,t} = \bar{\boldsymbol{\mu}}_{t-1} + \bar{\boldsymbol{\Sigma}}_{t-1}^{1/2}\boldsymbol{\eta}_{i,t}. \tag{14}$$

We characterize the statistics of $\boldsymbol{\eta}_{i,t}$ via its first two moments conditioned on the filtration $\{\mathcal{F}_t\}_{t\geq 0}$, where $\mathcal{F}_t$ represents the information available up to time $t$.

- **Mean Drift ($\boldsymbol{a}_{t-1}$):** Defined as $\boldsymbol{a}_{t-1} \triangleq \mathbb{E}[\boldsymbol{\eta}_{i,t}|\mathcal{F}_{t-1}]$. Geometrically, the vector $\boldsymbol{a}_{t-1}$ denotes the directional force driven by the asymmetry of the selection region, propelling the empirical mean towards the target $\mathbf{u}^*$.
- **Covariance Contraction ($\mathbf{B}_{t-1}$):** Defined as $\mathbf{B}_{t-1} \triangleq \mathrm{Cov}(\boldsymbol{\eta}_{i,t}|\mathcal{F}_{t-1})$. The matrix $\mathbf{B}_{t-1}$ captures the reduction in uncertainty due to biased data selection.

Accordingly, we define the dissipation matrix $\boldsymbol{\Psi}_{t-1}$ as:

$$\boldsymbol{\Psi}_{t-1} \triangleq \mathbf{I}_d - (\mathbf{B}_{t-1} + \boldsymbol{a}_{t-1}\boldsymbol{a}_{t-1}^\top). \tag{15}$$

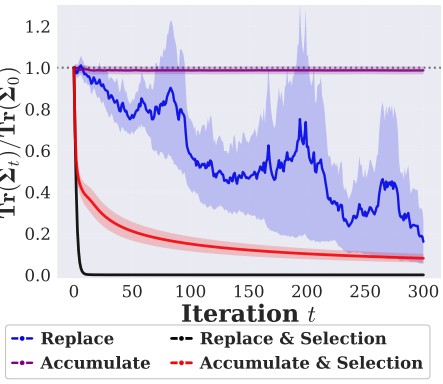

*Figure 1.* **Multivariate Gaussian Modeling**. We observe that *sample selection bias precipitates model collapse* in both Replace and Accumulate paradigms. In this setting, diversity under the Accumulate paradigm with selection dissipates at a power-law rate.

Under the local-basin analysis above, the first-order dissipation matrix admits a uniform positive spectral gap. In particular, there exists a constant $\psi > 0$ such that $\lambda_{\min}(\boldsymbol{\Psi}_{t-1}) \geq \psi$ along the trajectory.

**Theorem 2 (Asymptotic Rate of Model Collapse).**
*Assume that the trajectory remains in the local basin for all iterations. As iteration $t \to \infty$,*

$$\mathrm{Tr}(\bar{\boldsymbol{\Sigma}}_t) = \mathcal{O}_{a.s.}\left(t^{-\psi}\right). \tag{16}$$

Please refer to Subsection B.5 for the proof. Theorem 2 demonstrates a *power-law* rate. Intuitively, this shows a two-phase collapse dynamic: ① an initial phase of rapid homogenization driven by strict sample filtering, ② followed by slow, asymptotic convergence to a Dirac point mass. This quantification reveals the tension: stricter alignment with the local target $\mathbf{u}^*$ can improve local fidelity while increasing the dissipation magnitude $\lambda_{\min}(\boldsymbol{\Psi}_\infty)$. In data-scarce or low-resource environments, this means that a verifier with limited reference coverage may accelerate tail erosion by selecting samples that appear locally high-quality.

Empirically, we validate these findings in Figure 1 using the Multivariate Gaussian Modeling framework in Kazdan et al. (2025); Shumailov et al. (2024). Monitoring the ratio $\mathrm{Tr}(\bar{\boldsymbol{\Sigma}}_t)/\mathrm{Tr}(\bar{\boldsymbol{\Sigma}}_0)$, we compare the Replace paradigm ($n = 200$ samples per iteration) against the Accumulate paradigm. When incorporating a selection mechanism that filters the top-$n$ samples closest to a target state $\mathbf{u}^*$, we observe diversity collapse in both paradigms. Specifically, the Replace paradigm suffers from rapid variance depletion; the Accumulate paradigm exhibits the power-law decay predicted in Theorems 1 and 2: it mirrors the rapid initial collapse of the Replace paradigm but subsequently transitions into a protracted asymptotic decay. Due to space limitations, we defer additional results on non-Gaussian distributions and extended configurations to Subsection C.4.

## 3.3. The Wasserstein Cost of Model Collapse

While our variance-decay analysis leverages Gaussian approximations to obtain explicit dynamics, the impact of collapse on the generalization performance of the downstream prediction task does not hinge on such parametric assumptions. To quantify this effect under minimal distributional structure, we adopt standard regularity conditions commonly used in learning-theoretic optimal transport generalization (Courty et al., 2017; Redko et al., 2017).

**Assumption 2.** *Let $\mathcal{H}$ be a hypothesis class of predictors $h : \mathcal{X} \rightarrow \mathcal{Y}$, and we assume the following regularity conditions.*

- **Hypothesis Lipschitzness.** *Every $h$ is $\epsilon$-Lipschitz w.r.t. $d_{\mathcal{X}}$, i.e., $\|h(x) - h(x')\| \leq \epsilon\, d_{\mathcal{X}}(x, x')$ for all $x, x' \in \mathcal{X}$.*
- **Loss Lipschitzness.** *The loss $\mathcal{L}(\hat{y}, y)$ is $\ell$-Lipschitz in both inputs w.r.t. the bounded task-metric $d_{\mathcal{Y}}$, i.e., $|\mathcal{L}(\hat{y}_1, y_1) - \mathcal{L}(\hat{y}_2, y_2)| \leq \ell(d_{\mathcal{Y}}(\hat{y}_1, \hat{y}_2) + d_{\mathcal{Y}}(y_1, y_2))$ for all $\hat{y}, y \in \mathcal{Y}$.*
- **$(\epsilon, \delta)$-Probabilistic Cross-Lipschitzness.** *Let $g^* : \mathcal{X} \rightarrow \mathcal{Y}$ denote the global oracle mapping (ground truth). The local verification process imposes a biased effective supervision signal $g_t : \mathcal{X} \rightarrow \mathcal{Y}$ on the selected samples. We assume the local view $g_t$ and the global view $g^*$ satisfy $(\epsilon, \delta)$-probabilistic cross-Lipschitzness (Just et al., 2023).*

Let $\mathcal{R}_{\mathcal{D}}(h; g) \triangleq \mathbb{E}_{x \sim \mathcal{D}}[\mathcal{L}(h(x), g(x))]$ denote the expected risk of predictor $h$ under distribution $\mathcal{D}$. Following Just et al. (2023), we have the following generalization cost:

> **Theorem 3** (**Wasserstein Cost of Model Collapse**).
> *Let Assumption 2 hold. Let $\mathcal{D}_t$ be the distribution of synthetic data filtered by local verification at generation $t$, and let $\mathcal{D}^*$ be the true data manifold. For any $h_t \in \mathcal{H}$ trained on $\mathcal{D}_t$, the expected risk is bounded by*
>
> $$\begin{aligned} \mathcal{R}_{\mathcal{D}^*}(h_t; g^*) \leq \\ 2\ell\epsilon\, \mathbb{W}_p(\mathcal{D}_t, \mathcal{D}^*) + \mathcal{R}_{\mathcal{D}_t}(h_t; g_t) + \mathcal{O}(\ell\,\delta). \end{aligned} \quad (17)$$

Please refer to Subsection B.6 for the comprehensive proof. Theorem 3 decomposes the expected risk on the true manifold $\mathcal{D}^*$ into three components: (i) the distributional discrepancy measured by the Wasserstein distance $\mathbb{W}_p(\mathcal{D}_t, \mathcal{D}^*)$; (ii) the expected risk on the current filtered distribution $\mathcal{D}_t$; and (iii) an error term due to probabilistic violations. When $h_t$ is well-optimized on the local synthetic data $\mathcal{D}_t$, the second term becomes negligible. Consequently, the generalization performance is primarily dominated by the first term. Theorem 3 suggests that access to ground truth could theoretically preclude mode collapse. However, low-resource and sparsity constraints render such access infeasible in siloed environments. Unlike preference-based curation, where bias stems from explicit human objectives and is mitigable via established techniques (Grover et al., 2019; Chen et al., 2024), the verification bias in data silos is an intrinsic consequence of fragmented access to the global distribution.

## 4. Methodology

### 4.1. Wasserstein-Gradient-Based Selection

Inspired by Kessler et al. (2025); Li et al. (2024), we delineate the theoretical foundation of the selection mechanism. The core objective is to quantify the contribution of individual synthetic samples to the global distributional alignment. Specifically, given a synthetic dataset $\mathcal{P} = \{x_i\}_{i=1}^n$ and a reference real dataset $\mathcal{Q}$, we examine the sensitivity of the Wasserstein distance $\mathbb{W}_p(\mathcal{P}, \mathcal{Q})$ with respect to perturbations of $x_i$. Conceptually, this is similar to influence functions (IF) (Koh & Liang, 2017) which were adopted for adversarial curation strategies in self-consuming generative models (Wei & Zhang, 2025). A key distinction is that, while IF-based methods rely on linear approximations for infinitesimal perturbations, the discrete Wasserstein distance is formulated as a linear programming problem (LP). According to the Sensitivity Theorem (Bertsekas, 1997), the gradients derived from the LP dual solution remain valid for perturbations within a local polytope. This allows us to reliably predict the variation in Wasserstein distance induced by re-weighting a sample without the need for re-calculation.

Similar to Li et al. (2024), we utilize the first-order variation induced by the Kantorovich dual potentials. Recall the dual formulation: $\mathbb{W}_p(\mathcal{P}, \mathcal{Q}) = \sup_{(f,g) \in \Phi_c} (\langle f, \mathcal{P} \rangle + \langle g, \mathcal{Q} \rangle)$, where $\Phi_c$ denotes the set of admissible potentials satisfying $f(x) + g(y) \leq d^p(x, y)$. The optimal dual potential $f^*$ serves as the subgradient of the transport cost with respect to the probability mass $P$, i.e., $\nabla_{\mathcal{P}} \mathbb{W}_p(\mathcal{P}, \mathcal{Q}) = (f^*)^\top$.

To eliminate the degree of freedom arising from the translational invariance of the simplex constraint ($\sum_{i=1}^N P = 1$), the zero-sum convention is adopted to fix the subgradients, following the strategy in Just et al. (2023). Consequently, for a sample $x_i \in \mathcal{P}$, the calibrated gradients are derived by:

$$\mathcal{S}(x_i) \triangleq \frac{\partial \mathbb{W}_p(\mathcal{P}, \mathcal{Q})}{\partial \mathcal{P}(x_i)} = f^*(x_i) - \frac{1}{N-1} \sum_{j \neq i}^N f^*(x_j). \quad (18)$$

**Remark 1.** *Equation (18) yields intriguing interpretations. Specifically, a **positive score** $\mathcal{S}(x_i)$ implies that removing the sample $x_i$ reduces the total discrepancy $\mathbb{W}_p(\mathcal{P}, \mathcal{Q})$, whereas a **negative score** suggests that its removal would exacerbate the divergence. Consequently, this drives us to prune samples exhibiting positive scores to shift the distribution $\mathcal{P}$ closer to the target manifold $\mathcal{Q}$ in Wasserstein distance.*

Intuitively, the synthetic dataset $\mathcal{P}$ could be transmitted to decentralized parties holding real data shards $\mathcal{Q}_k$ ($k = 1, \ldots, K$) to compute the sensitivity scores $\mathcal{S}_k(x_i)$. However, recent studies (Ganev & De Cristofaro, 2025; Chen et al., 2020; van Breugel et al., 2023) highlight that synthetic data remains susceptible to privacy leakage. This critical limitation motivates privacy-preserving methods based on the following Wasserstein geometry property.

## 4.2. Wasserstein Geometry Property

We first introduce intrinsic geometric properties in $\mathcal{P}_p(\mathbb{R}^d)$.

**Property 1** (**Wasserstein Barycenter**, Aguéh & Carlier (2011))**.** *Let $\{\mathcal{Q}_k\}_{k=1}^K \subset \mathcal{P}_p(\mathbb{R}^d)$ denote a collection of square-integrable probability measures with weights $\{\lambda_k\}_{k=1}^K$ such that $\sum_{k=1}^K \lambda_k = 1$, where typically $\lambda_k = 1/K$. The Wasserstein barycenter $\mathcal{Q}^*$ is characterized as:*

$$\mathcal{Q}^* = \arg\min_{\mathcal{Q} \in \mathcal{P}_p(\mathbb{R}^d)} \sum_{k=1}^K \lambda_k \mathbb{W}_p^p(\mathcal{Q}, \mathcal{Q}_k). \quad (19)$$

**Property 2** (**McCann's Interpolation**, McCann (1997); Rakotomamonjy et al. (2024))**.** *For any two continuous measures $\mathcal{P}_0, \mathcal{P}_1 \in \mathcal{P}_p(\mathbb{R}^d)$, the displacement interpolation $\mathcal{P}_t$ for $t \in [0,1]$ defines the optimal trajectory between them. It is constructed via a transport map $\mathbf{T}$ pushing $\mathcal{P}_0$ to $\mathcal{P}_1$:*

$$\mathcal{P}_t = ((1-t)\mathbf{I}_d + t\mathbf{T})_{\#}\mathcal{P}_0. \quad (20)$$

*In the discrete setting, we approximate the intractable map $\mathbf{T}$ using barycentric mapping (Courty et al., 2018). The interpolation is realized by the trajectory $x_i(t) = (1-t)x_i + t \cdot n(\mathbf{P}^\star \mathbf{X}')_i$, where the barycentric projection $n(\mathbf{P}^\star \mathbf{X}')_i$ acts as the empirical proxy for $\mathbf{T}(x_i)$ (Peyré et al., 2019).*

**Property 3** (**Wasserstein Geodesics**, Ambrosio et al. (2005); Kolouri et al. (2017))**.** *Let $\mathbf{T}$ be the optimal transport map pushing $\mathcal{P}$ to $\mathcal{Q}$. For any $t \in (0,1)$, the intermediate interpolant $\xi^*$ is constructed as $\xi^* = ((1-t)\mathbf{I}_d + t\mathbf{T})_{\#}\mathcal{P}$. This interpolant satisfies:*

$$\mathbb{W}_p(\mathcal{P}, \mathcal{Q}) = \mathbb{W}_p(\mathcal{P}, \xi^*) + \mathbb{W}_p(\xi^*, \mathcal{Q}). \quad (21)$$

**Main Intuition.** Li et al. (2024) show that Wasserstein distances and barycenters can be computed without sharing raw data by constructing a proxy interpolation $\xi^*$ (via Property 2) on the geodesic between two distributions $\mathcal{P}$ and $\mathcal{Q}$; this interpolant satisfies the geodesic property in Property 3.

## 4.3. Collaborative Geodesic Interpolation Calculation

We now present `Scheme I`. Motivated by Theorem 1, by broadening the selection criteria to encompass multiple target preferences rather than a solitary preference $u^*$, one can expand the selection space $\mathcal{R}_t$, thereby mitigating collapse.

We utilize Property 3 to avoid direct data communication. When $\xi_k^*$ is an interpolation measure (computed by Property 2) on the geodesic between $\mathcal{P}$ and $\mathcal{Q}_k$, the dual formulation is $\mathbb{W}_p(\mathcal{P}, \mathcal{Q}_k) = \sup_{(f_k, g_k) \in \Phi_c} (\langle f_k, \mathcal{P} \rangle + \langle g_k, \xi_k^* \rangle) + \sup_{(h_k, j_k) \in \Phi_c} (\langle h_k, \xi_k^* \rangle + \langle j_k, \mathcal{Q}_k \rangle)$.

We thus have $\nabla_{\mathcal{P}} \mathbb{W}_p(\mathcal{P}, \mathcal{Q}_k) \approx \nabla_{\mathcal{P}} \mathbb{W}_p(\mathcal{P}, \xi_k^*) = (f^*)^\top$, which implies we can compute $\mathcal{S}_k(x_i)$ using the proxy interpolations $\xi_k^*$ without directly accessing real data $\mathcal{Q}_k$.

$$\mathcal{S}_k(x_i) = \frac{\partial \mathbb{W}_p(\mathcal{P}, \xi_k^*)}{\partial \mathcal{P}(x_i)} = f^*(x_i) - \frac{1}{N-1} \sum_{j \neq i}^N f^*(x_j) \quad (22)$$

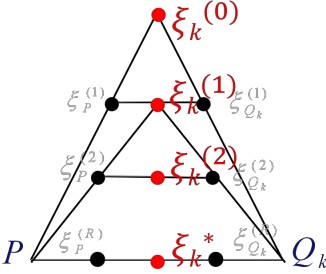

*Figure 2.* **Collaborative Geodesic Interpolation Calculation.** Holders of synthetic data $\mathcal{P}$ and real data $\mathcal{Q}_k$ jointly compute the proxy measure $\xi_k^*$ along the geodesic without raw data exchange. $\xi_{\mathcal{P}}^{(r)}$ and $\xi_{\mathcal{Q}_k}^{(r)}$ denote intermediate interpolants between the proxy measure $\xi_k^{(r)}$ and the distributions $\mathcal{P}$ and $\mathcal{Q}_k$, respectively.

Therefore, the key problem is finding interpolations on the Wasserstein geodesics between $\mathcal{P}$ and $\mathcal{Q}_k$ without sharing $\mathcal{P}$ and $\mathcal{Q}_k$. As illustrated in Figure 2, we abstract the overall process as a geometric procedure: For round $r = 1, \ldots, R$:

**Initialization.** An arbitrary proxy measure $\xi_k^{(0)}$ is initialized, typically as a Gaussian distribution or a selection from a public dataset proximate to the target distribution.

1. **Interpolation.** The interpolations $\xi_{\mathcal{P}}^{(r)}$ and $\xi_{\mathcal{Q}_k}^{(r)}$ are computed from the proxy $\xi_k^{(r)}$ to $\mathcal{P}$ and $\mathcal{Q}_k$, respectively, i.e., $\xi_{\mathcal{P}}^{(r)} \in \arg\min_\xi \mathcal{W}_p(\mathcal{P}, \xi) + \mathcal{W}_p(\xi, \xi_k^{(r)})$ and $\xi_{\mathcal{Q}_k}^{(r)} \in \arg\min_\xi \mathcal{W}_p(\mathcal{Q}_k, \xi) + \mathcal{W}_p(\xi, \xi_k^{(r)})$.

2. **Communication.** The interpolant $\xi_{\mathcal{P}}^{(r)}$ is transmitted by the party holding $\mathcal{P}$ to the party holding $\mathcal{Q}_k$.

3. **Update.** The updated proxy measure $\xi_k^{(r+1)}$ is calculated via an interpolation measure between $\xi_{\mathcal{P}}^{(r)}$ and $\xi_{\mathcal{Q}_k}^{(r)}$ and is subsequently returned to the synthetic data party.

**Theorem 4** (**Monotonicity and Interpolation Convergence**, Rakotomamonjy et al. (2024))**.** *For interpolation $\xi_k^{(r)}$ at iteration $r$, define the sequence as:*

$$\mathcal{E}_k^{(r)} = \mathcal{W}_p(\mathcal{Q}_k, \xi_k^{(r)}) + \mathcal{W}_p(\xi_k^{(r)}, \mathcal{P}) \quad (23)$$

*Then, the sequence $\{\mathcal{E}^{(r)}\}_{r \geq 0}$ is non-increasing and converges to its infimum $\mathcal{E}_k^* = \mathcal{W}_p(\mathcal{Q}_k, \mathcal{P})$.*

Please refer to Subsection B.7 for the comprehensive proof. The above process is fully consistent with the process in Rakotomamonjy et al. (2024) for computing the Wasserstein distance, while we use $\xi_k^*$ as a proxy signal of $\mathcal{Q}_k$ for data selection. Specifically, let the candidate synthetic dataset be represented as an empirical measure $\mathcal{P} = \frac{1}{N} \sum_{i=1}^N \delta_{x_i}$. For each candidate sample $x_i$, each party $k$ contributes a score $\mathcal{S}_k(x_i)$; colloquially, every party scores $x_i$. We select a subset of indices $\mathcal{I}$ with cardinality constraint $|\mathcal{I}| \leq n < N$, yielding the filtered distribution $\mathcal{P}_{\mathcal{I}} = \frac{1}{n} \sum_{i \in \mathcal{I}} \delta_{x_i}$.

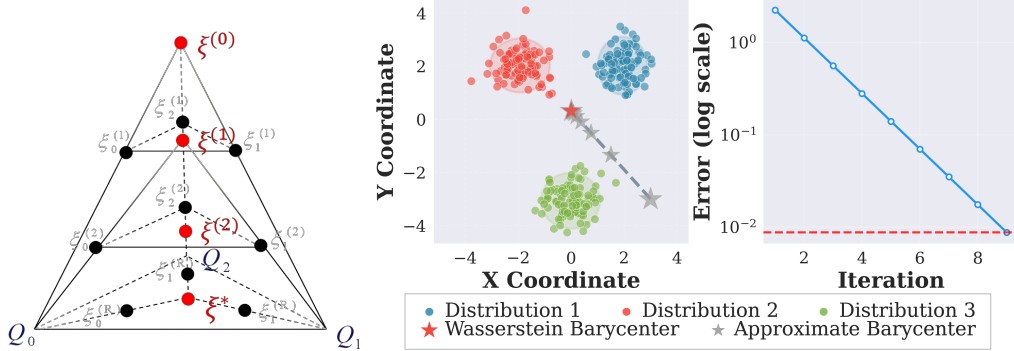

*Figure 3.* **Collaborative Wasserstein Barycenter Estimation.** (**Left**) Schematic illustration of the iterative interpolation process for computing the approximate barycenter $\xi^{(r)}$. (**Middle**) Empirical visualization of the optimization trajectory, demonstrating the convergence from an arbitrary initialization to the barycenter. (**Right**) Numerical convergence results showing the monotonic decrease.

To ensure $\mathcal{P}_{\mathcal{I}}$ covers the distributional modes of all parties while avoiding redundancy, we draw inspiration from recommendation systems that maximize the coverage of diverse user interests (Krause & Golovin, 2014) and mitigate social bias in synthetic data (Mehrotra & Vishnoi, 2023). We formulate the selection as a *monotone submodular maximization problem* by modeling the diminishing return of adding similar local synthetic data for each individual party. Colloquially, this means that multiple parties score synthetic data, instead of a single biased data silo.

$$\underset{\mathcal{I} \subseteq \{1, \dots, N\}}{\text{maximize}} \quad \sum_{k=1}^{K} g\Big( \sum_{i \in \mathcal{I}} (1 - \tilde{\mathcal{S}}_k(x_i)) \Big) \quad \text{s.t. } |\mathcal{I}| \leq n, \quad (24)$$

where $\tilde{\mathcal{S}}_k(x_i)$ denotes the normalized score (min-max normalization) and $g : \mathbb{R}_{\geq 0} \to \mathbb{R}$ is a non-decreasing concave function (e.g., $g(z) = \log(1 + z)$) that penalizes redundancy. Since the composition of a concave function with a non-negative modular sum is guaranteed to be submodular (Bach, 2013), this problem can be solved using a standard greedy algorithm, which provides a guarantee of $(1 - 1/e)$-approximation to the optimum (Nemhauser et al., 1978).

However, this approach is limited in scalability, as any modification to the synthetic dataset $\mathcal{P}$ necessitates a complete re-computation of the interpolation. Therefore, we propose an offline framework using a Wasserstein barycenter.

### 4.4. Collaborative Wasserstein Barycenter Estimation

We now present Scheme II. Motivated by Theorem 3, which suggests that if the filtered distribution approximates the ground truth, we can theoretically mitigate mode collapse. To achieve this, we compute a proxy for the ground truth distribution, i.e., the Wasserstein barycenter (Property 1). Similar to Subsection 4.3, we abstract the process as the following geometric procedure illustrated in Figure 3:
**Initialization.** A central server initializes an estimated barycenter $\xi^{(0)}$ with multiple parties holding real data distributions $\mathcal{Q}_k$ for $k = 0, 1, 2$. For round $r = 0, \dots, R - 1$:

1. **Interpolation.** The server broadcasts the current estimate $\xi^{(r)}$. Each party $k$ computes the geodesic interpolation $\xi_k^{(r)}$ between $\xi^{(r)}$ and its local distribution $\mathcal{Q}_k$, i.e., $\xi_k^{(r)} \in \arg\min_\xi \mathcal{W}_p(\mathcal{Q}_k, \xi) + \mathcal{W}_p(\xi, \xi^{(r)})$.

2. **Communication.** The server aggregates the interpolations $\{\xi_k^{(r)}\}_k$ from all participating parties.

3. **Update.** The server updates the estimated barycenter $\xi^{(r+1)} = \sum_{k=1}^{K} 1/K \cdot \xi_k^{(r)}$.

---

**Theorem 5** (**Monotonicity and Barycenter Convergence**). *Let $\mathcal{Q}_k$ be the distribution of the $k$-th client with weight $\lambda_k > 0$. Let $\xi^{(r)}$ denote the global approximated barycenter at iteration $r$. Define the true Fréchet variance (objective sequence) as:*

$$\mathcal{E}^{(r)} = \sum_{k=1}^{K} \lambda_k \mathcal{W}_2^2(\mathcal{Q}_k, \xi^{(r)}) \qquad (25)$$

*Then, the sequence $\{\mathcal{E}^{(r)}\}_{r \geq 0}$ is non-increasing and converges to its infimum $\mathcal{E}^* = \sum_{k=1}^{K} \lambda_k \mathcal{W}_2^2(\mathcal{Q}_k, \xi^*)$.*

---

Please refer to Subsection B.8 for the comprehensive proof. Empirically, Figure 3 shows that our method yields an approximation that is virtually indistinguishable from the true Wasserstein barycenter computed by the standard free-support algorithm (Cuturi & Doucet, 2014).

By utilizing the calibrated gradient $\mathcal{S}(x_i) = \frac{\partial \mathbb{W}_p(\mathcal{P}, \xi^*)}{\partial \mathcal{P}(x_i)} = f^*(x_i) - \frac{1}{N-1} \sum_{j \neq i}^{N} f^*(x_j)$, we identify top-$\alpha$ samples aligned with the global ground truth. Crucially, this decouples the proxy estimation from the synthetic data generation, allowing $\xi^*$ to be reused even as synthetic data $\mathcal{P}$ changes.

**Remark 2.** *As stated in Rakotomamonjy et al. (2024), compressing a dataset into a proxy makes the reconstruction problem ill-posed. To seek formal guarantees beyond implicit protection, one can further integrate the differential privacy framework (Lê Tien et al., 2019) (Subsection C.6).*

*Table 1.* **Computational Time Scalability Analysis**. Wall-clock time in seconds for one round of data selection under varying candidate set size $N$, reference set size $M$, and number of clients $K$. Unless otherwise varied, we fix $N = 100{,}000$, $M = 5{,}000$, and $K = 10$.

| Candidate Set Size | | | Reference Set Size per Client | | | Number of Clients | | |
|---|---|---|---|---|---|---|---|---|
| $N$ | Scheme I (s) | Scheme II (s) | $M$ | Scheme I (s) | Scheme II (s) | $K$ | Scheme I (s) | Scheme II (s) |
| 10,000 | 6.7 | 2.1 | 1,000 | 24.8 | 20.8 | 5 | 45.0 | 41.5 |
| 50,000 | 14.4 | 10.3 | 2,000 | 29.1 | 25.4 | 10 | 45.5 | 41.5 |
| 100,000 | 24.7 | 20.8 | 5,000 | 45.5 | 41.6 | 20 | 45.8 | 41.6 |
| 150,000 | 34.9 | 31.0 | 8,000 | 64.5 | 60.7 | 50 | 47.5 | 41.8 |
| 200,000 | 45.1 | 41.4 | 10,000 | 79.6 | 75.8 | 100 | 49.2 | 42.1 |

## 5. Computational Complexity

We further evaluate the scalability of the proposed selection mechanisms by measuring the wall-clock time required for one round of data selection. CIFAR-10 is used as the base dataset, and the candidate pool is enlarged to $N = 200{,}000$ by replicating the original 50,000 images. We compare two variants: *Interpolation Proxy* (Algorithm 1) and *Barycenter Proxy* (Algorithm 2). The benchmark follows a parallel simulation protocol, where client-side computations are assumed to run concurrently, matching the federated setting.

Unless otherwise specified, the default configuration is $N = 200{,}000$ candidate samples, $M = 5{,}000$ reference samples per client, $K = 10$ clients, and $R = 10$ interpolation rounds. For each experiment, we vary one parameter while keeping the remaining parameters fixed. The reported runtime includes GPU data transfer, distance matrix computation, and selection or transport operations, while excluding initial data loading and communication overhead.

We now summarize the leading computational cost. Let $N = |\mathcal{P}|$ denote the size of the synthetic candidate pool, $M = |\mathcal{Q}_k|$ the size of each local real dataset, $S$ the support size of the proxy distribution, $L$ the number of Sinkhorn scaling iterations, $R$ the number of interpolation rounds in Scheme I, $T$ the number of barycenter estimation rounds in Scheme II, and $n$ the selection budget. The complexity is reported as parallel wall-clock complexity.

**Theorem 6** (**Computational Complexity**). *Under Sinkhorn-based OT with $L$ scaling iterations, the leading parallel complexities of our schemes are*

$$\mathcal{T}_{Scheme\ I} = \mathcal{O}(RL(N + M + S)S + nNK),$$
$$\mathcal{T}_{Scheme\ II} = \mathcal{O}(TLMS + LNS). \qquad (26)$$

The first term in $\mathcal{T}_{Scheme\ I}$ comes from the collaborative geodesic interpolation between the synthetic distribution $\mathcal{P}$ and each local real distribution $\mathcal{Q}_k$. Since this proxy depends on the current candidate pool, a new synthetic pool generally requires recomputing the interpolation. The second term corresponds to greedy sample selection, where marginal gains are evaluated across $K$ client-side scores.

By contrast, Scheme II separates proxy estimation from candidate selection. Its barycenter estimation cost, $TLMS$, depends on the local real distributions and the barycenter support size, but not on the synthetic candidate size $N$. Once the barycenter proxy is obtained, scoring a new synthetic pool requires only a single Sinkhorn-based pass of cost $LNS$. This decoupling is important in iterative synthetic data generation: when $\mathcal{P}$ changes, Scheme II can reuse the barycenter proxy, whereas Scheme I must recompute the interpolation proxy. As shown in Table 1, both methods scale approximately linearly with $N$ and $M$, consistent with Theorem 6. However, their behavior differs more clearly as $K$ increases. Scheme I becomes more expensive because it repeatedly interacts with client-specific interpolation proxies and aggregates client-wise selection scores. In contrast, Scheme II remains nearly flat in the parallel setting, since the barycenter estimation is decoupled from the synthetic candidate pool and can be amortized across multiple rounds of synthetic data generation. This explains why the empirical gap in wall-clock time is larger than what a single-round comparison alone suggests.

## 6. Experiments

We follow the settings (baselines, configurations, and metrics) in Shidani et al. (2025); Rezaei et al. (2026); Bertrand et al. (2024). Due to space constraints, detailed settings and supplementary results are deferred to Appendix C.

**BASELINES** Following Rezaei et al. (2026), CovMatch (Rezaei et al., 2026) uses a greedy strategy to match the feature covariance of real data; CenterMatch (He et al., 2023) selects images closest to the centroid of real features; and K-means (Lin et al., 2023) clusters the generated pool into groups and selects images closest to each cluster center.

**METRICS** Following Bertrand et al. (2024), we report Fréchet Inception Distance (**FID**) (Heusel et al., 2017) for quality, as well as **Precision** for fidelity and **Recall** for diversity based on Inception-V3 (Kynkäänniemi et al., 2019).

**SETUP** Following Shi et al. (2025), we conduct experiments with DDPM (Ho et al., 2020) across **CIFAR-10** (Krizhevsky et al., 2009), **CelebA** (Liu et al., 2015), and **STL-10** (Coates

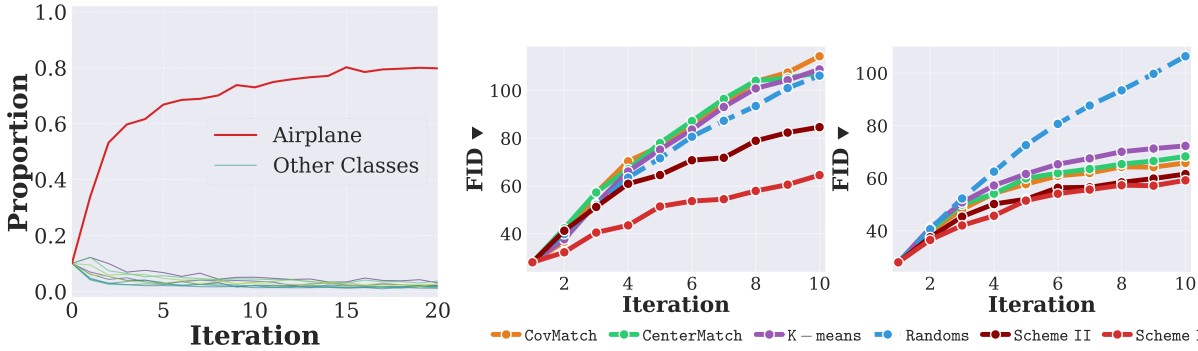

*Figure 4.* **Evolution of the proportion** of the Airplane class and the nine other classes using the Airplane class as a local reference dataset (Left). **Evolution of FID** (↓) of the generated images using ExDir $(1, 0.1)$ (Middle) and IID partition (Right) as local reference datasets.

*Table 2.* **Results after 10 iterations** with ExDir $(1, 0.1)$ reference set. Colors indicate the 1st, 2nd, and 3rd best results.

| Method | CIFAR-10 | | | STL-10 | | | CelebA | | |
|---|---|---|---|---|---|---|---|---|---|
| | FID ↓ | Precision ↑ | Recall ↑ | FID ↓ | Precision ↑ | Recall ↑ | FID ↓ | Precision ↑ | Recall ↑ |
| `Random` | 106 | 0.53 | 0.48 | 95 | 0.49 | 0.53 | 96 | 0.51 | 0.28 |
| `K-means` (Lin et al., 2023) | 102 | 0.56 | 0.40 | 89 | 0.54 | 0.53 | 87 | 0.59 | 0.48 |
| `CenterMatch` (He et al., 2023) | 116 | 0.50 | 0.35 | 111 | 0.57 | 0.58 | 87 | 0.64 | 0.46 |
| `CovMatch` (Rezaei et al., 2026) | 115 | 0.51 | 0.47 | 131 | 0.59 | 0.55 | 92 | 0.65 | 0.51 |
| **Scheme II** (Subsection 4.4) | 85 | 0.57 | 0.57 | 69 | 0.57 | 0.63 | 75 | 0.70 | 0.62 |
| **Scheme I** (Subsection 4.3) | 71 | 0.60 | 0.58 | 65 | 0.66 | 0.71 | 69 | 0.69 | 0.71 |

et al., 2011). In the main text, we primarily investigate the *Accumulate-Subsample* paradigm: we select $n$ instances from $N = 4n$ generated candidates to augment the data pool, and subsequently subsample $n$ instances for training from the pool. Since Dirichlet partitions yield weak initial models, we first utilize the training set of size $n = 50,000$ to establish a strong generator. Note that the training set is not added to the accumulated pool; instead, the data is distributed among 10 parties to serve as local reference sets. In the specific case of STL-10, we allocate the 5,000 labeled samples across the 10 parties.

Figure 4 (Left) illustrates the proportion of each class in the training set predicted by a pre-trained VGG11 (92.39% test accuracy) when only the Airplane class is used as the reference dataset for `CenterMatch`. The results highlight that when the selection mechanism is guided by a skewed local prior, the diversity deteriorates rapidly, leading to homogenization. Figure 4 (Middle & Right) illustrates that while selection baselines mitigate collapse in IID scenarios, they surprisingly lag behind `Random` in non-IID scenarios.

Table 2 shows that baselines with poor performance on CIFAR-10 and STL-10 achieve better results on face generation. Intuitively, this stems from the highly structured nature of face data: even with a biased reference set, the filtered images still retain the basic characteristics.

## 7. Lessons Learned

**Sample Selection Bias Accelerates Collapse in Data Silos.** Although local filtering is introduced to mitigate model collapse, optimizing synthetic data solely against local reference signals can have the opposite effect. In siloed environments, the local real-data distribution provides only a partial view of the target distribution. As a result, filtering synthetic samples by their agreement with local ground truth induces a confirmation-bias effect: samples that deviate from the local reference, including valid but underrepresented modes, are more likely to be discarded. This gradually narrows the synthetic training distribution and accelerates diversity loss across recursive generations.

**Low-Resource Regimes Are Especially Vulnerable.** This failure mode becomes more pronounced when real-data coverage is scarce or fragmented. In low-resource regimes, tail regions are weakly represented even before synthetic augmentation begins, so local-reference selection can easily confuse rare but valid modes with low-quality generations. The filtering process therefore does not merely remove noise; it systematically suppresses underrepresented regions of the target distribution. Data scarcity is thus amplified into persistent tail pruning, making low-resource silos particularly susceptible to collapse.

## Impact Statement

**Advancing the Sustainability of Generative AI.** This research addresses a critical existential risk facing the future of artificial intelligence: model collapse driven by recursive training on synthetic data. As the digital ecosystem becomes saturated with machine-generated content, future models risk degenerating into homogenized representations of reality that lose statistical fidelity. By theoretically quantifying the diversity (variance) decay rate as a power law and providing a geometric solution to mitigate it, this work offers a pathway to sustain the self-improvement of Generative AI systems without necessitating a continuous influx of human-generated data, which is becoming a scarce resource.

**Ethical Considerations and Limitations.** There remains a risk that if the collaborative proxy is poisoned or biased by a majority of participating nodes, the selection mechanism could enforce a collective bias rather than a true ground truth. Therefore, deployment of these methods requires careful governance to ensure the participating entities in a siloed network represent a sufficiently diverse cross-section of the target domain. Although the algorithmic dynamics regarding adversarial attacks and defenses are intriguing, this setting exceeds the scope of this study, which is dedicated to investigating model collapse within data silos and taking the first step toward its resolution by shifting from collaborative learning to collaborative evaluation.

## Acknowledgements

This work was supported in part by the Chongqing Key Laboratory of Trusted Perception and Interaction Technology for Intelligent and Connected Vehicles, the State Key Laboratory of Intelligent Vehicle Safety Technology, Chongqing Changan Automobile Co., Ltd., and the Chongqing Natural Science Foundation (Grant No. CSTB2024NSCQ-LZX0172), and in part by the National Natural Science Foundation of China (Grant No. 62572433).

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

# Table of Contents

# A. Related Work

## A.1. Model Collapse

The phenomenon of model collapse is not a monolith but rather a spectrum of pathological behaviors that have been analyzed from different theoretical perspectives. A significant body of literature characterizes collapse primarily through the lens of prediction error or population risk on the original data distribution. Within this framework, collapse is often defined as a catastrophic, nonlinear degradation of model performance, rendering the model functionally useless after a limited number of recursive generations (Schaeffer et al., 2025). A more rigorous mathematical formulation extends this to the asymptotic divergence of the test loss, describing a scenario where the model drifts indefinitely away from the ground-truth manifold (Shumailov et al., 2024; Dohmatob et al., 2024). While these risk-based definitions provide a high-level signal of failure, they tend to obscure the underlying geometric deformations

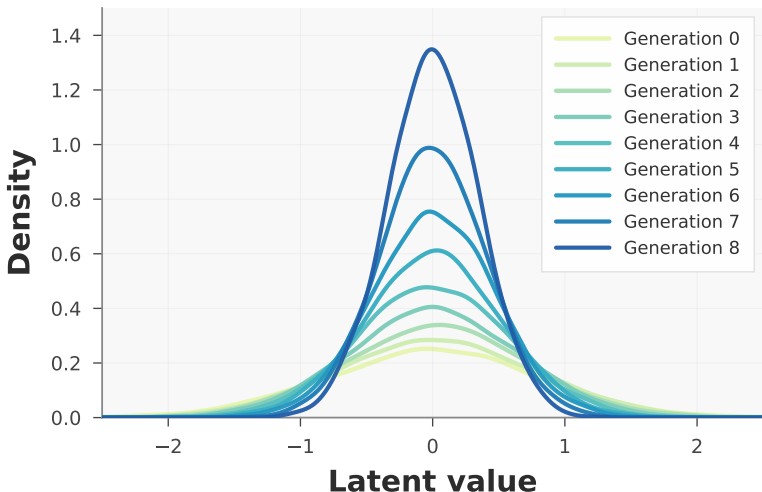

*Figure 5.* How latent-space distributions evolve over the course of training using generated data. As seen in the Gaussian instance, the model tends to ignore extreme values (tails) and gravitate toward the mean.

occurring within the learned distribution, leading to recent calls for more granular definitions based on distribution shifts (Schaeffer et al., 2025).

Complementary to risk-based metrics, a distinct stream of research focuses on the statistical degeneration of the generated data's topology. This perspective characterizes collapse as a transition from generalization to memorization, where the model's output diversity shrinks significantly (Shi et al., 2025). As shown in Figure 5, the core of this phenomenon is *Variance Collapse*, where recursive training causes the variance of the learned distribution to contract asymptotically toward zero, effectively reducing the distribution to a point estimate (Alemohammad et al., 2024). This process is frequently preceded by the erosion of distributional tails, where low-probability events (the "long tail") are statistically washed away, leaving only a homogenized core (Hataya et al., 2023; Shumailov et al., 2024). In multi-modal settings, this manifests as mode dropping or mode entanglement, simplifying the complex data manifold into a few high-density regions (Alemohammad et al., 2024). Importantly, such tail erosion is not merely a statistical artifact: it can have uneven social and linguistic consequences. Jarvis et al. (2026) argue that low-resource languages and marginalized communities often occupy precisely these underrepresented regions of the data distribution, and thus may experience model collapse earlier and more severely than high-resource communities. This perspective reframes collapse as a distributional inequity problem, where the loss of rare modes corresponds to the erasure of culturally, linguistically, or demographically underrepresented content.

To understand the mechanics of this decay, recent works have analyzed the interplay between loss functions and high-dimensional geometry. Theoretical underpinnings attribute variance collapse to the smoothing nature of standard loss functions (e.g., MSE or cross-entropy), which encourage models to approximate the conditional mean of the data and act as low-pass filters that dampen high-frequency variations (Mobahi et al., 2020). This effect is exacerbated in high-dimensional regimes. As the dimensionality increases, the volume of the distribution support grows exponentially, making the tails increasingly difficult to cover with synthetic samples. Consequently, the covariance shift between the target distribution and the synthetic approximation becomes the dominant factor in generalization error, accelerating the collapse process (Rezaei et al., 2026). The severity of these pathological behaviors is intrinsically linked to the training paradigm employed. Theoretical analyses typically distinguish between *Replace* and *Accumulate* strategies. Under the *Replace* paradigm, where training data is entirely substituted by synthetic samples at each generation, variance collapse is theoretically proven to be inevitable regardless of model capacity (Shumailov et al., 2024; Dohmatob et al., 2024; Mobahi et al., 2020). Conversely, the *Accumulate* paradigm (where synthetic data is appended to historical real data) has been shown to offer asymptotic stability (Kazdan et al., 2025). Empirical studies suggest that maintaining access to the original data (or a high-quality subset) anchors the distribution, preventing the unbounded drift of the mean and variance (Gerstgrasser et al., 2024).

## A.2. Data Selection

Recognizing the inevitability of model collapse under naive recursive training, the research community has pivoted toward *data selection* (Dohmatob et al., 2025b) as a primary mitigation strategy. Existing approaches can be broadly categorized into three methodological families: fidelity-based filtering, diversity-preserving alignment, and oracle-guided verification.

The intuitive approach involves curating synthetic data based on scalar quality metrics, such as perplexity, log-likelihood, or confidence scores. Classic strategies employ truncation or rejection sampling to discard samples that deviate significantly from the model's high-density regions (Alemohammad et al., 2024). More recent advances leverage *coreset selection* techniques, aiming to identify a minimal subset of synthetic data that approximates the gradient or loss landscape of the full distribution (Pooladzandi et al., 2022). While empirical studies show that pruning data based on hardness or quality metrics can beat power-law scaling (Sorscher et al., 2022), these quality-centric methods often inadvertently accelerate *Variance Collapse*. By systematically favoring safe modal samples (Mai et al., 2026), human preferences reinforce the very homogenization that characterizes the collapse regime (Alemohammad et al., 2024).

To counteract variance reduction, a second stream of research focuses on maximizing distributional coverage. From an information-theoretic perspective, (Shi et al., 2025) identify a critical phase transition from generalization to memorization, proposing entropy thresholds to maintain manifold richness. Geometrically, (Rezaei et al., 2026) prove that for linear models, generalization error is dominated by covariance shift, establishing *Covariance Matching* as an asymptotically optimal strategy. Moreover, recent works have adopted *Influence Functions* to quantify the causal effect of individual samples on reasoning capabilities (Mlodozeniec et al., 2025; Lin et al., 2025; Qiao et al., 2026; Humane et al., 2025; Qiao et al., 2025).

The theoretically grounded line of inquiry posits that scaling up with synthetic data is only viable given a reliable verification mechanism. (Feng et al., 2025) formalized this hypothesis, demonstrating that access to a high-precision verifier (e.g., an external oracle) allows synthetic data training to surpass baselines trained solely on real data. Similarly, (Shoshan et al., 2023) explore using generative models to synthesize validation sets for model selection, contingent on the ability to calibrate synthetic error against real-domain risk. Under this framework, the efficacy of selection is strictly bounded by the precision and recall of the external supervisor relative to the ground truth.

**Research Gap: The Absence of Global Verification.** A critical, often unstated assumption unifies the diverse strategies discussed above: they all presume access to a centralized "ground truth" or global statistics. Specifically, coreset selection (Pooladzandi et al., 2022) and covariance matching (Rezaei et al., 2026) require access to the full target distribution to compute gradients or statistics, while verification frameworks (Feng et al., 2025; Shoshan et al., 2023) assume an oracle capable of distinguishing real from synthetic distributions. In real-world decentralized deployments, such as fragmented data silos, this assumption is fundamentally invalid. Local entities possess only partial, biased views of the global distribution. Consequently, applying existing selection strategies locally leads to local selection bias.

## A.3. Optimal Transport

Optimal Transport (OT) provides a rigorous geometric framework for comparing probability distributions, offering distinct advantages over statistical divergences by capturing the underlying metric space structure (Villani et al., 2008). While computational advancements such as Sinkhorn iterations (Cuturi, 2013) have enabled OT applications in high-dimensional machine learning, standard algorithms require centralized access to raw samples, rendering them inapplicable in privacy-sensitive environments (Mai et al., 2025). To address these constraints, recent scholarship has extended OT to decentralized settings. In the context of *metric estimation*, (Rakotomamonjy et al., 2024) proposed the federated Wasserstein Distance, utilizing geodesic interpolants to approximate transport costs across disparate clients without exchanging raw data. The most closely related work to ours is Li et al. (2024), which inspires our study by leveraging Wasserstein barycenters for data valuation in FL. From an *optimization* perspective, (Dvurechenskii et al., 2018) established distributed algorithms for computing barycenters over networks via consensus protocols. Regarding *privacy*, (Lê Tien et al., 2019) introduced Differentially Private OT, employing randomized projections to optimize transportation plans under privacy budgets.

A fundamental distinction exists between these prior works and our proposed framework. Existing literature utilizes distributed OT primarily for *metric estimation* (accurately estimating the global $\mathbb{W}_2$ distance (Rakotomamonjy et al., 2024)) or *adaptation* (learning a mapping between source and target domains (Lê Tien et al., 2019)). In contrast, our work repurposes the OT geometry for *gradient-based data selection*. By computing the calibrated gradient of the Wasserstein distance with respect to individual synthetic samples, we quantify the marginal contribution of each sample to ground-truth manifold alignment. This transforms distributed OT from a passive metric into an active selection mechanism.

# B. Proof

## B.1. Auxiliary Lemmata

Before presenting the main proofs, we introduce several key technical lemmas required for our analysis, including Lemmas 3 and 4, which are used in the proof of Theorem 1.

**Lemma 1.** *Consider a selection mechanism on a multivariate normal distribution $\mathcal{N}(\boldsymbol{\mu}, \boldsymbol{\Sigma})$ defined by a fixed, locally concave utility function $U(\mathbf{x})$ maximized at $\mathbf{u}^*$. Let $\mathcal{R}$ be the high-utility selection region preserving probability mass $\alpha \in (0, 1)$. Let $\boldsymbol{\Delta} \triangleq \mathbb{E}[\mathbf{x} \mid \mathbf{x} \in \mathcal{R}] - \boldsymbol{\mu}$ be the mean drift vector. Assuming $\boldsymbol{\mu}$ is in a local neighborhood of $\mathbf{u}^*$, there exists a constant $\kappa > 0$ such that the drift acts as a restoring force in the standardized (Mahalanobis) metric:*

$$(\boldsymbol{\mu} - \mathbf{u}^*)^\top \boldsymbol{\Sigma}^{-1} \boldsymbol{\Delta} \leq -\kappa \|\boldsymbol{\mu} - \mathbf{u}^*\|_{\boldsymbol{\Sigma}^{-1}}^2 \tag{27}$$

*Proof.* We analyze the drift by strictly operating in the standardized isotropic frame to avoid the non-commutativity issues of matrix products in the parameter space.

**1. Transformation to Standardized Space:** Let $\mathbf{z} = \boldsymbol{\Sigma}^{-1/2}(\mathbf{x} - \boldsymbol{\mu})$ be the whitening transformation. The selection region $\mathcal{R}$ corresponds to a region $\mathcal{D}(\mathbf{c})$ in the standardized space, where $\mathbf{c} = \boldsymbol{\Sigma}^{-1/2}(\boldsymbol{\mu} - \mathbf{u}^*)$ is the standardized error vector.

The mean drift vector in the parameter space is given by $\boldsymbol{\Delta} = \boldsymbol{\Sigma}^{1/2} \boldsymbol{a}(\mathbf{c})$, where $\boldsymbol{a}(\mathbf{c}) = \mathbb{E}[\mathbf{z} \mid \mathbf{z} \in \mathcal{D}(\mathbf{c})]$ is the drift in the standardized frame. We aim to show that $\mathbf{c}$ and $\boldsymbol{a}(\mathbf{c})$ are opposed.

**2. Gradient Analysis (Strict Contraction):** The Jacobian of the standardized drift is related to the conditional covariance:

$$\nabla_{\mathbf{c}} \boldsymbol{a}(\mathbf{c}) = \text{Cov}(\mathbf{z} \mid \mathbf{z} \in \mathcal{D}(\mathbf{c})) - \mathbf{I}_d. \tag{28}$$

Since the utility function $U(\mathbf{x})$ is locally concave, the selection induces a log-concave constraint. By applying the Brascamp-Lieb inequality (or properties of log-concave measures), the conditional covariance is strictly contracted relative to the identity matrix. Thus, there exists $\kappa > 0$ such that:

$$\text{Cov}(\mathbf{z} \mid \mathbf{z} \in \mathcal{D}(\mathbf{c})) \preceq (1 - \kappa)\mathbf{I}_d \implies \nabla_{\mathbf{c}} \boldsymbol{a}(\mathbf{c}) \preceq -\kappa \mathbf{I}_d. \tag{29}$$

**3. Inner Product Bound:** Instead of approximating the vector $\boldsymbol{a}(\mathbf{c})$, we apply the Mean Value Theorem directly to the inner product $\mathbf{c}^\top \boldsymbol{a}(\mathbf{c})$. Note that $\boldsymbol{a}(\mathbf{0}) = \mathbf{0}$ due to the local symmetry around the optimum.

$$\mathbf{c}^\top \boldsymbol{a}(\mathbf{c}) = \mathbf{c}^\top \left( \int_0^1 \nabla \boldsymbol{a}(t\mathbf{c}) dt \right) \mathbf{c}. \tag{30}$$

Since the Jacobian is uniformly bounded by $-\kappa \mathbf{I}_d$, the quadratic form is bounded as:

$$\mathbf{c}^\top \boldsymbol{a}(\mathbf{c}) \leq -\kappa \|\mathbf{c}\|^2. \tag{31}$$

**4. Mapping back to Parameter Space:** We substitute the standardized variables back into the Mahalanobis inner product:

$$(\boldsymbol{\mu} - \mathbf{u}^*)^\top \boldsymbol{\Sigma}^{-1} \boldsymbol{\Delta} = (\boldsymbol{\Sigma}^{1/2} \mathbf{c})^\top \boldsymbol{\Sigma}^{-1} (\boldsymbol{\Sigma}^{1/2} \boldsymbol{a}(\mathbf{c})) \tag{32}$$

$$= \mathbf{c}^\top \boldsymbol{\Sigma}^{1/2} \boldsymbol{\Sigma}^{-1} \boldsymbol{\Sigma}^{1/2} \boldsymbol{a}(\mathbf{c}) \tag{33}$$

$$= \mathbf{c}^\top \boldsymbol{a}(\mathbf{c}) \tag{34}$$

$$\leq -\kappa \|\mathbf{c}\|^2 \tag{35}$$

$$= -\kappa \|\boldsymbol{\mu} - \mathbf{u}^*\|_{\boldsymbol{\Sigma}^{-1}}^2. \tag{36}$$

This confirms that the drift acts as a restoring force in the natural geometry induced by the covariance $\boldsymbol{\Sigma}$. □

**Lemma 2** (Robbins–Siegmund Theorem, Robbins & Siegmund (1971)). *Let $(\Omega, \mathcal{F}, P)$ be a probability space equipped with a filtration $\{\mathcal{F}_n\}_{n \geq 0}$, where $\mathcal{F}_n$ represents the information available up to time $n$. Let $\{z_n\}_{n \geq 0}$, $\{a_n\}_{n \geq 0}$, $\{x_n\}_{n \geq 0}$ and $\{y_n\}_{n \geq 0}$ be nonnegative $\mathcal{F}_n$-measurable random variables. Assume that, almost surely,*

$$\mathbb{E}_n[z_{n+1} \mid \mathcal{F}_n] \leq (1 + a_n)z_n + x_n - y_n, \tag{RS}$$

*and that*

$$\sum_{n=0}^{\infty} a_n < \infty \quad a.s., \qquad \sum_{n=0}^{\infty} x_n < \infty \quad a.s. \tag{37}$$

*Then,*

1. *$\{z_n\}$ converges almost surely to a finite random variable $z_\infty$.*

2. *$\sum_{n=0}^{\infty} y_n < \infty \quad a.s.$*

**Lemma 3** (Special Case of equation RS, Liu et al. (2025)). *Maintain the setting and notation of Lemma 2. Let $\{T_n\}_{n \geq 0}$ be a deterministic sequence with $0 < T_n < 1$ for all $n$. Let $\alpha > 0$ and $\eta > 0$ be constants and suppose the sequences are identified as $a_n = 0$, $x_n = \eta T_n^2$, $y_n = \alpha T_n z_n$. This yields the following recursion:*

$$\mathbb{E}_n[z_{n+1} \mid \mathcal{F}_n] \leq (1 - \alpha T_n) z_n + \eta T_n^2. \tag{38}$$

*If this recursion satisfies the Robbins–Monro condition*

$$\sum_{n=0}^{\infty} T_n = \infty \qquad and \qquad \sum_{n=0}^{\infty} T_n^2 < \infty \tag{39}$$

*Then*

$$z_n \xrightarrow{a.s.} 0 \tag{40}$$

**Lemma 4** (Ruhe's Trace Inequality, Ruhe (1970)). *Let $\mathbf{A}$ and $\mathbf{B}$ be $n \times n$ positive semidefinite Hermitian matrices with eigenvalues denoted by $a_1 \geq \cdots \geq a_n \geq 0$ and $b_1 \geq \cdots \geq b_n \geq 0$, respectively. Then, the trace of their product is bounded by:*

$$\sum_{i=1}^{n} a_i b_{n-i+1} \leq \mathrm{Tr}(\mathbf{AB}) \leq \sum_{i=1}^{n} a_i b_i. \tag{41}$$

## B.2. Proof of Proposition 1

**Proposition 1** (**Replace Paradigm**, Shumailov et al.). *Under the Replace paradigm, as iteration $t \to \infty$, the estimator statistics satisfy:*

$$\boldsymbol{\Sigma}_t \xrightarrow{a.s.} \mathbf{0}. \tag{2}$$

*Let $\mathbb{W}_2$ denote the Wasserstein-2 distance. We have*

$$\mathbb{E}[\mathbb{W}_2^2(\mathcal{N}_t, \mathcal{N}_0)] \to \infty. \tag{3}$$

*Proof.* We analyze the two claims separately: the divergence of the Wasserstein distance and the collapse of the variance.

**1. Wasserstein Divergence ($\mathbb{E}[\mathbb{W}_2^2] \to \infty$).** First, observe that the Wasserstein-2 distance between the approximation $\mathcal{N}(\bar{\boldsymbol{\mu}}_t, \bar{\boldsymbol{\Sigma}}_t)$ and the true distribution $\mathcal{N}_0$ is lower-bounded by the Euclidean distance of their means:

$$\mathbb{W}_2^2(\mathcal{N}(\bar{\boldsymbol{\mu}}_t, \bar{\boldsymbol{\Sigma}}_t), \mathcal{N}_0) \geq \|\bar{\boldsymbol{\mu}}_t - \bar{\boldsymbol{\mu}}_0\|^2. \tag{42}$$

Let $\hat{\boldsymbol{\mu}}_0$ and $\hat{\boldsymbol{\Sigma}}_0$ be the empirical mean and covariance estimated from the initial samples of $\mathcal{N}_0$. By the triangle inequality:

$$\mathbb{W}_2^2(\mathcal{N}(\bar{\boldsymbol{\mu}}_t, \bar{\boldsymbol{\Sigma}}_t), \mathcal{N}_0) + \mathbb{W}_2^2(\mathcal{N}_0, \mathcal{N}(\hat{\boldsymbol{\mu}}_0, \hat{\boldsymbol{\Sigma}}_0)) \geq \frac{1}{2}\mathbb{W}_2^2(\mathcal{N}(\bar{\boldsymbol{\mu}}_t, \bar{\boldsymbol{\Sigma}}_t), \mathcal{N}(\hat{\boldsymbol{\mu}}_0, \hat{\boldsymbol{\Sigma}}_0)). \tag{43}$$

Rearranging this yields:

$$\mathbb{W}_2^2(\mathcal{N}(\bar{\boldsymbol{\mu}}_t, \bar{\boldsymbol{\Sigma}}_t), \mathcal{N}_0) \geq \frac{1}{2}\|\bar{\boldsymbol{\mu}}_t - \hat{\boldsymbol{\mu}}_0\|^2 - \mathbb{W}_2^2(\mathcal{N}_0, \mathcal{N}(\hat{\boldsymbol{\mu}}_0, \hat{\boldsymbol{\Sigma}}_0)). \tag{44}$$

Under the Replace paradigm, the sampling and fitting process matches the setting of Shumailov et al. (2024) with unbiased estimators. Referring to the results of Eq. (21) in the Supplementary information of Shumailov et al. (2024), we have

$$\mathbb{E}[\|\bar{\boldsymbol{\mu}}_t - \hat{\boldsymbol{\mu}}_0\|^2] \geq \mathrm{Tr}(\bar{\boldsymbol{\Sigma}}_0) \sum_{\tau=1}^{t} \frac{1}{n}. \tag{45}$$

Since the sample size is constant $n$ at each generation, the term sums to $\frac{t}{n}$. Thus, as $t \to \infty$:

$$\mathbb{E}[\mathbb{W}_2^2(\mathcal{N}(\bar{\boldsymbol{\mu}}_t, \bar{\boldsymbol{\Sigma}}_t), \mathcal{N}_0)] \geq \frac{\mathrm{Tr}(\bar{\boldsymbol{\Sigma}}_0)}{2n}t - \mathbb{W}_2^2(\mathcal{N}_0, \mathcal{N}(\hat{\boldsymbol{\mu}}_0, \hat{\boldsymbol{\Sigma}}_0)) \to \infty. \tag{46}$$

**2. Variance Collapse ($\bar{\boldsymbol{\Sigma}}_t \xrightarrow{a.s.} \mathbf{0}$).** The proof relies on the martingale properties of the covariance trace. Consider the recursive update step where samples are generated as $\mathbf{X}_{i,t} = \bar{\boldsymbol{\Sigma}}_{t-1}^{1/2}\mathbf{z}_{i,t} + \bar{\boldsymbol{\mu}}_{t-1}$ with $\mathbf{z}_{i,t} \sim \mathcal{N}(\mathbf{0}, \mathbf{I}_d)$.

The trace of the covariance matrix, $\mathrm{Tr}(\bar{\boldsymbol{\Sigma}}_t)$, forms a lower-bounded supermartingale. By Doob's Martingale Convergence Theorem, it must converge to a random variable $\boldsymbol{\Sigma}_\infty$:

$$\mathrm{Tr}(\bar{\boldsymbol{\Sigma}}_t) \longrightarrow \boldsymbol{\Sigma}_\infty. \tag{47}$$

Specifically, the trace evolves multiplicatively as $\mathrm{Tr}(\bar{\boldsymbol{\Sigma}}_t) = Q_t \mathrm{Tr}(\bar{\boldsymbol{\Sigma}}_{t-1})$, where $Q_t$ is a generalized $\chi^2$ random variable (with expectation $\mathbb{E}[Q_t] = 1$) representing the sum of $d$ independent $\chi^2$ variables weighted by the eigenvalues of $\bar{\boldsymbol{\Sigma}}_{t-1}$.

For any generation $t$, at least one weight is significant, ensuring that $P(|Q_t - 1| > \epsilon) > c > 0$ for some constant $c$. Consequently, for the product $\mathrm{Tr}(\bar{\boldsymbol{\Sigma}}_t) = \mathrm{Tr}(\bar{\boldsymbol{\Sigma}}_0) \prod_{j=1}^{t} Q_j$ to converge to a finite limit $\boldsymbol{\Sigma}_\infty$, it must be that $\boldsymbol{\Sigma}_\infty = 0$ almost surely (otherwise the fluctuations of $Q_t$ would prevent convergence). Since all matrix norms are equivalent, $\mathrm{Tr}(\bar{\boldsymbol{\Sigma}}_t) \to 0$ implies:

$$\bar{\boldsymbol{\Sigma}}_t \xrightarrow{a.s.} \mathbf{0}. \tag{48}$$

$\square$

## B.3. Proof of Proposition 2

**Proposition 2** (**Accumulate Paradigm**, Kazdan et al.). *Under the Accumulate paradigm, as iteration $t \to \infty$, we have the following convergence in expectation:*

$$\mathbb{E}[(\bar{\boldsymbol{\mu}}_t - \bar{\boldsymbol{\mu}}_0)^2] \to (1 - \alpha_n)\bar{\boldsymbol{\Sigma}}_0, \tag{5}$$

$$\mathbb{E}[\bar{\boldsymbol{\Sigma}}_t] \to \alpha_n \bar{\boldsymbol{\Sigma}}_0, \quad \text{where } \alpha_n = \frac{\sin(\pi/\sqrt{n})}{\pi/\sqrt{n}}. \tag{6}$$

*Consequently, the Wasserstein distance stabilizes at:*

$$\mathbb{E}[\mathbb{W}_2^2(\mathcal{N}_t, \mathcal{N}_0)] \to 2(1 - \sqrt{\alpha_n})\mathrm{Tr}(\bar{\boldsymbol{\Sigma}}_0). \tag{7}$$

*Proof.* In this section, we provide a rigorous derivation of the convergence of the Accumulate paradigm in the multivariate setting. We define the recursive process as follows:

**Sampling Step:** At generation $t$, data are sampled from the estimated distribution of the previous generation:

$$X_{i,t} = \bar{\boldsymbol{\mu}}_{t-1} + \bar{\boldsymbol{\Sigma}}_{t-1}^{1/2} z_{i,t}, \quad \text{where } z_{i,t} \sim \mathcal{N}(0, \mathbf{I}_d), \quad i = 1, \ldots, n \tag{49}$$

**Learning Step:** The parameters are updated using all accumulated historical data from generation $0$ to $t$:

$$\bar{\boldsymbol{\mu}}_t = \frac{1}{tn} \sum_{\tau=1}^{t} \sum_{i=1}^{n} X_{i,\tau} \tag{50}$$

$$\bar{\boldsymbol{\Sigma}}_t = \frac{1}{tn} \sum_{\tau=1}^{t} \sum_{i=1}^{n} (X_{i,\tau} - \bar{\boldsymbol{\mu}}_t)(X_{i,\tau} - \bar{\boldsymbol{\mu}}_t)^\top \tag{51}$$

### B.3.1. RECURSIVE RELATION OF THE MEAN

We first derive the recursive relationship between $\bar{\boldsymbol{\mu}}_t$ and $\bar{\boldsymbol{\mu}}_{t-1}$. Decomposing the summation into the history (generations 1 to $t-1$) and the current generation $t$:

$$\bar{\boldsymbol{\mu}}_t = \frac{1}{tn} \left( \sum_{\tau=1}^{t-1} \sum_{i=1}^{n} X_{i,\tau} + \sum_{i=1}^{n} X_{i,t} \right) \tag{52}$$

Observing that the first term sums to $n(t-1)\bar{\boldsymbol{\mu}}_{t-1}$, and substituting the sampling equation for $X_{i,t}$:

$$\bar{\boldsymbol{\mu}}_t = \frac{1}{tn} \left[ n(t-1)\bar{\boldsymbol{\mu}}_{t-1} + \sum_{i=1}^{n} (\bar{\boldsymbol{\mu}}_{t-1} + \bar{\boldsymbol{\Sigma}}_{t-1}^{1/2} z_{i,t}) \right] \tag{53}$$

$$= \frac{1}{tn} \left[ n(t-1)\bar{\boldsymbol{\mu}}_{t-1} + n\bar{\boldsymbol{\mu}}_{t-1} + n\bar{\boldsymbol{\Sigma}}_{t-1}^{1/2} \bar{z}_t \right] \tag{54}$$

$$= \frac{1}{tn} \left[ nt\bar{\boldsymbol{\mu}}_{t-1} + n\bar{\boldsymbol{\Sigma}}_{t-1}^{1/2} \bar{z}_t \right] \tag{55}$$

where $\bar{z}_t = \frac{1}{n} \sum_{i=1}^{n} z_{i,t} \sim \mathcal{N}(0, \frac{1}{n}\mathbf{I}_d)$. This yields the recurrence:

$$\bar{\boldsymbol{\mu}}_t = \bar{\boldsymbol{\mu}}_{t-1} + \bar{\boldsymbol{\Sigma}}_{t-1}^{1/2} \frac{\bar{z}_t}{t} \tag{56}$$

Unrolling this recurrence leads to the general form:

$$\bar{\boldsymbol{\mu}}_t = \bar{\boldsymbol{\mu}}_0 + \sum_{k=1}^{t} \bar{\boldsymbol{\Sigma}}_{k-1}^{1/2} \frac{\bar{z}_k}{k} \tag{57}$$

Taking the expectation, since $\mathbb{E}[\bar{z}_k] = 0$, we have $\mathbb{E}[\bar{\boldsymbol{\mu}}_t] = \bar{\boldsymbol{\mu}}_0$.

### B.3.2. RECURSIVE RELATION OF THE COVARIANCE

We aim to find the expected covariance matrix $\mathbb{E}[\bar{\boldsymbol{\Sigma}}_t]$. Using the law of total expectation, we first compute $\mathbb{E}[\bar{\boldsymbol{\Sigma}}_t|\mathcal{F}_{t-1}]$. Using the multivariate decomposition identity $\sum_i^N (x-c)(x-c)^\top = \sum_i^N (x-\bar{x})(x-\bar{x})^\top + N(\bar{x}-c)(\bar{x}-c)^\top$, we decompose the total covariance into *Within-Group Covariance* (A) and *Between-Group Deviation* (B):

$$nt\bar{\boldsymbol{\Sigma}}_t = \sum_{\tau=1}^{t} \left[ \underbrace{\sum_{i=1}^{n}(X_{i,\tau}-\bar{X}_\tau)(X_{i,\tau}-\bar{X}_\tau)^\top}_{\text{Term } A_\tau : \text{Within}} + \underbrace{\sum_{i=1}^{n}(\bar{X}_\tau-\bar{\boldsymbol{\mu}}_t)(\bar{X}_\tau-\bar{\boldsymbol{\mu}}_t)^\top}_{\text{Term } B_\tau : \text{Between}} \right] \tag{58}$$

where $\bar{X}_\tau$ is the sample mean of generation $\tau$. Substituting $X_{i,\tau} = \bar{\boldsymbol{\mu}}_{\tau-1} + \bar{\boldsymbol{\Sigma}}_{\tau-1}^{1/2} z_{i,\tau}$, we have $\bar{X}_\tau = \bar{\boldsymbol{\mu}}_{\tau-1} + \bar{\boldsymbol{\Sigma}}_{\tau-1}^{1/2}\bar{z}_\tau$. The within-group term simplifies using $S_\tau = \sum_{i=1}^{n}(z_{i,\tau}-\bar{z}_\tau)(z_{i,\tau}-\bar{z}_\tau)^\top$:

$$\text{Term } A_\tau = \bar{\boldsymbol{\Sigma}}_{\tau-1}^{1/2} S_\tau (\bar{\boldsymbol{\Sigma}}_{\tau-1}^{1/2})^\top \tag{59}$$

We calculate the conditional expectation by splitting the summation into historical terms ($\tau < t$) and the new term ($\tau = t$).

**Step 1: Historical Contribution ($\tau < t$).** For $\tau < t$, $\bar{\boldsymbol{\Sigma}}_{\tau-1}$ is fixed in $\mathcal{F}_{t-1}$.

- **Within Covariance:** $\mathbb{E}[\text{Term } A_\tau] = (n-1)\bar{\boldsymbol{\Sigma}}_{\tau-1}$.

- **Between Deviation:** We express the deviation vector:

$$\bar{X}_\tau - \bar{\boldsymbol{\mu}}_t = (\bar{X}_\tau - \bar{\boldsymbol{\mu}}_{t-1}) - (\bar{\boldsymbol{\mu}}_t - \bar{\boldsymbol{\mu}}_{t-1}) = (\bar{X}_\tau - \bar{\boldsymbol{\mu}}_{t-1}) - \bar{\boldsymbol{\Sigma}}_{t-1}^{1/2}\frac{\bar{z}_t}{t} \tag{60}$$

Squaring this (outer product) and taking the expectation, the cross-term vanishes due to the independence of $\bar{z}_t$ from historical data. The random part contributes:

$$\mathbb{E}\left[ n\left(\bar{\boldsymbol{\Sigma}}_{t-1}^{1/2}\frac{\bar{z}_t}{t}\right)\left(\bar{\boldsymbol{\Sigma}}_{t-1}^{1/2}\frac{\bar{z}_t}{t}\right)^\top \right] = \frac{n}{t^2}\bar{\boldsymbol{\Sigma}}_{t-1}^{1/2}\frac{\mathbf{I}_d}{n}(\bar{\boldsymbol{\Sigma}}_{t-1}^{1/2})^\top = \frac{1}{t^2}\bar{\boldsymbol{\Sigma}}_{t-1} \tag{61}$$

Thus, the total expectation for one historical step $\tau$ is:

$$\mathbb{E}[\text{Term } A_\tau] = \mathbb{E}\left[\sum_{i=1}^{n}(X_{i,\tau}-\bar{\boldsymbol{\mu}}_t)(X_{i,\tau}-\bar{\boldsymbol{\mu}}_t)^\top\right] = n\bar{\boldsymbol{\Sigma}}_{t-1} + \frac{1}{t^2}\bar{\boldsymbol{\Sigma}}_{t-1} \tag{62}$$

Summing over the $t-1$ historical steps and using the definition of the previous total covariance $\bar{\boldsymbol{\Sigma}}_{t-1}$:

$$\sum_{\tau=1}^{t-1}\mathbb{E}[\text{Term } A_\tau] = \sum_{\tau=1}^{t-1}\mathbb{E}\left[\sum_{i=1}^{n}(X_{i,\tau}-\bar{\boldsymbol{\mu}}_t)(X_{i,\tau}-\bar{\boldsymbol{\mu}}_t)^\top\right] = n(t-1)\bar{\boldsymbol{\Sigma}}_{t-1} + \frac{t-1}{t^2}\bar{\boldsymbol{\Sigma}}_{t-1} \tag{63}$$

**Step 2: New Data Contribution ($\tau = t$).** For the current step $\tau = t$:

- **Within Covariance:** $\mathbb{E}[\text{Term } A_t] = (n-1)\bar{\boldsymbol{\Sigma}}_{t-1}$.

- **Between Deviation:** Using Equation (56), we have $\bar{X}_t - \bar{\boldsymbol{\mu}}_t = \bar{\boldsymbol{\Sigma}}_{t-1}^{1/2}\bar{z}_t(1-\frac{1}{t})$. Taking the expectation:

$$\mathbb{E}[\text{Term } B_t] = n\left(1-\frac{1}{t}\right)^2 \bar{\boldsymbol{\Sigma}}_{t-1}^{1/2}\frac{\mathbf{I}_d}{n}(\bar{\boldsymbol{\Sigma}}_{t-1}^{1/2})^\top = \left(\frac{t-1}{t}\right)^2\bar{\boldsymbol{\Sigma}}_{t-1} \tag{64}$$

**Step 3: Aggregation** Summing all components:

$$\mathbb{E}[nt\bar{\boldsymbol{\Sigma}}_t|\mathcal{F}_{t-1}] = \left[\underbrace{n(t-1)}_{\text{History Base}} + \underbrace{(t-1)/t^2}_{\text{History Drift}} + \underbrace{(n-1)}_{\text{New Within}} + \underbrace{(t-1)^2/t^2}_{\text{New Between}}\right]\bar{\boldsymbol{\Sigma}}_{t-1} \tag{65}$$

$$= \left[n(t-1) + n - 1 + \frac{t-1+(t-1)^2}{t^2}\right]\bar{\boldsymbol{\Sigma}}_{t-1} \tag{66}$$

$$= \left[nt - 1 + \frac{t-1}{t}\right]\bar{\boldsymbol{\Sigma}}_{t-1} = \left[nt - \frac{1}{t}\right]\bar{\boldsymbol{\Sigma}}_{t-1} \tag{67}$$

Dividing by $nt$:

$$\mathbb{E}[\bar{\boldsymbol{\Sigma}}_t|\mathcal{F}_{t-1}] = \left(1 - \frac{1}{nt^2}\right)\bar{\boldsymbol{\Sigma}}_{t-1} \tag{68}$$

Since $1 - \frac{1}{nt^2} < 1$, we have $\mathbb{E}[\bar{\boldsymbol{\Sigma}}_t|\mathcal{F}_{t-1}] \preceq \bar{\boldsymbol{\Sigma}}_{t-1}$, which implies that $\{\bar{\boldsymbol{\Sigma}}_t\}$ is a nonnegative supermartingale. By the Martingale Convergence Theorem, $\bar{\boldsymbol{\Sigma}}_t$ converges almost surely to a limiting random matrix $\bar{\boldsymbol{\Sigma}}_\infty$.

Taking the total expectation recursively yields the infinite product:

$$\mathbb{E}[\bar{\boldsymbol{\Sigma}}_t] = \bar{\boldsymbol{\Sigma}}_0 \prod_{k=1}^{t}\left(1 - \frac{1}{nk^2}\right) \tag{69}$$

As $t \to \infty$, using Euler's infinite product formula for the sinc function $\frac{\sin(\pi x)}{\pi x} = \prod(1 - \frac{x^2}{k^2})$, with $x = 1/\sqrt{n}$:

$$\lim_{t\to\infty}\mathbb{E}[\bar{\boldsymbol{\Sigma}}_t] = \frac{\sin(\pi/\sqrt{n})}{\pi/\sqrt{n}}\bar{\boldsymbol{\Sigma}}_0 = \alpha_n\bar{\boldsymbol{\Sigma}}_0 \tag{70}$$

### B.3.3. WASSERSTEIN DISTANCE

The squared 2-Wasserstein distance between $\mathcal{N}(\bar{\boldsymbol{\mu}}_t, \bar{\boldsymbol{\Sigma}}_t)$ and $\mathcal{N}(\bar{\boldsymbol{\mu}}_0, \bar{\boldsymbol{\Sigma}}_0)$ is given by:

$$\mathbb{E}[\mathbb{W}_2^2] = \mathbb{E}[\|\bar{\boldsymbol{\mu}}_t - \bar{\boldsymbol{\mu}}_0\|^2] + \mathbb{E}[\mathfrak{B}^2(\bar{\boldsymbol{\Sigma}}_t, \bar{\boldsymbol{\Sigma}}_0)] \tag{71}$$

**Mean Shift Term:** Recall $\bar{\boldsymbol{\mu}}_t - \bar{\boldsymbol{\mu}}_0 = \sum_{k=1}^{t}\bar{\boldsymbol{\Sigma}}_{k-1}^{1/2}\frac{\bar{z}_k}{k}$. The terms $\bar{z}_k$ are independent with variance $\mathbf{I}_d/n$. Thus:

$$\mathbb{E}[\|\bar{\boldsymbol{\mu}}_t - \bar{\boldsymbol{\mu}}_0\|^2] = \sum_{k=1}^{t}\frac{1}{k^2}\text{Tr}\left(\frac{1}{n}\bar{\boldsymbol{\Sigma}}_{k-1}^{1/2}\mathbf{I}_d(\bar{\boldsymbol{\Sigma}}_{k-1}^{1/2})^\top\right) = \sum_{k=1}^{t}\frac{1}{nk^2}\text{Tr}(\mathbb{E}[\bar{\boldsymbol{\Sigma}}_{k-1}]) \tag{72}$$

Using the product form $\mathbb{E}[\bar{\boldsymbol{\Sigma}}_{k-1}] = P_{k-1}\bar{\boldsymbol{\Sigma}}_0$ where $P_k = \prod_{\tau=1}^{k}(1 - \frac{1}{n\tau^2})$, we construct a telescoping sum. Notice that $P_k = P_{k-1}(1 - \frac{1}{nk^2}) \implies P_{k-1} - P_k = \frac{1}{nk^2}P_{k-1}$. Substituting this into the summation:

$$\mathbb{E}[\|\bar{\boldsymbol{\mu}}_t - \bar{\boldsymbol{\mu}}_0\|^2] = \text{Tr}(\bar{\boldsymbol{\Sigma}}_0)\sum_{k=1}^{t}(P_{k-1} - P_k) \tag{73}$$

$$= \text{Tr}(\bar{\boldsymbol{\Sigma}}_0)(P_0 - P_t) = \text{Tr}(\bar{\boldsymbol{\Sigma}}_0)(1 - \alpha_n) \tag{74}$$

**Covariance Distance Term:** As $t \to \infty$, $\bar{\boldsymbol{\Sigma}}_t \to \alpha_n\bar{\boldsymbol{\Sigma}}_0$ almost surely. The Bures metric term becomes:

$$\lim_{t\to\infty}\mathfrak{B}^2(\bar{\boldsymbol{\Sigma}}_t, \bar{\boldsymbol{\Sigma}}_0) = \text{Tr}(\alpha_n\bar{\boldsymbol{\Sigma}}_0 + \bar{\boldsymbol{\Sigma}}_0 - 2(\alpha_n\bar{\boldsymbol{\Sigma}}_0^{1/2}\bar{\boldsymbol{\Sigma}}_0\bar{\boldsymbol{\Sigma}}_0^{1/2})^{1/2}) \tag{75}$$

$$= \text{Tr}((1 + \alpha_n)\bar{\boldsymbol{\Sigma}}_0 - 2\sqrt{\alpha_n}\bar{\boldsymbol{\Sigma}}_0) \tag{76}$$

$$= (1 + \alpha_n - 2\sqrt{\alpha_n})\text{Tr}(\bar{\boldsymbol{\Sigma}}_0) = (1 - \sqrt{\alpha_n})^2\text{Tr}(\bar{\boldsymbol{\Sigma}}_0) \tag{77}$$

Combining both terms:

$$\lim_{t\to\infty}\mathbb{E}[\mathbb{W}_2^2] = (1 - \alpha_n)\text{Tr}(\bar{\boldsymbol{\Sigma}}_0) + (1 - \sqrt{\alpha_n})^2\text{Tr}(\bar{\boldsymbol{\Sigma}}_0) = 2(1 - \sqrt{\alpha_n})\text{Tr}(\bar{\boldsymbol{\Sigma}}_0) \tag{78}$$

$$\square$$

## B.4. Proof of Theorem 1

**Theorem 1** (**Selection Bias Precipitates Collapse**). *Consider the Accumulate paradigm with top-$\alpha$ selection toward an ideal $\mathbf{u}^*$. As iteration $t \to \infty$, the estimator statistics behave as follows:*

$$\|\bar{\boldsymbol{\mu}}_t - \mathbf{u}^*\|^2 \xrightarrow{a.s.} 0. \tag{10}$$

$$\bar{\boldsymbol{\Sigma}}_t \xrightarrow{a.s.} \mathbf{0}. \tag{11}$$

*The Wasserstein-2 distance converges to:*

$$\mathbb{E}[\mathbb{W}_2^2(\mathcal{N}_t, \mathcal{N}_0)] \to \|\mathbf{u}^* - \bar{\boldsymbol{\mu}}_0\|^2 + \mathrm{Tr}(\bar{\boldsymbol{\Sigma}}_0) \tag{12}$$

*Proof.* In this section, we provide the complete derivation for the Accumulate paradigm under biased selection (top-$\alpha$ selection based on a local utility $U(x)$). We analyze the evolution of the mean vector and the covariance matrix to prove the convergence of the mean and the collapse of the variance.

**Step 1: Sampling and Selection** At generation $t$, we first generate raw samples from the previous estimate:

$$\tilde{X}_{i,t} \sim \mathcal{N}(\bar{\boldsymbol{\mu}}_{t-1}, \bar{\boldsymbol{\Sigma}}_{t-1}), \quad i = 1, \ldots, n \tag{79}$$

The selection mechanism retains samples within a high-utility region $\mathcal{R}_t \subset \mathbb{R}^d$ enclosing the target $u^*$, such that the acceptance probability is $\alpha$. The selected samples $X_{i,t}$ follow a truncated multivariate normal distribution:

$$X_{i,t} \sim \mathcal{TN}(\bar{\boldsymbol{\mu}}_{t-1}, \bar{\boldsymbol{\Sigma}}_{t-1}, \mathcal{R}_t) \tag{80}$$

To facilitate analysis, we utilize the standardized reparameterization. Let $\mathcal{D}_{t-1}$ be the standardized selection region:

$$\mathcal{D}_{t-1} = \{\mathbf{z} \in \mathbb{R}^d \mid \bar{\boldsymbol{\mu}}_{t-1} + \bar{\boldsymbol{\Sigma}}_{t-1}^{1/2}\mathbf{z} \in \mathcal{R}_t\} \tag{81}$$

The selected samples can be expressed as:

$$X_{i,t} = \bar{\boldsymbol{\mu}}_{t-1} + \bar{\boldsymbol{\Sigma}}_{t-1}^{1/2}\boldsymbol{\eta}_{i,t} \tag{82}$$

where $\boldsymbol{\eta}_{i,t} \sim \mathcal{TN}(\mathbf{0}, \mathbf{I}_d, \mathcal{D}_{t-1})$ is a standard truncated multivariate normal vector.

**Step 2: Key Moments** We characterize the statistics of the standardized truncated noise $\boldsymbol{\eta}_{i,t}$:

1. **Mean Drift Vector:** $\boldsymbol{a}_{t-1} \triangleq \mathbb{E}[\boldsymbol{\eta}_{i,t}|\mathcal{F}_{t-1}]$.

   This vector represents the directional force toward the high-utility region.

2. **Covariance Contraction Matrix:** $\mathbf{B}_{t-1} \triangleq \mathrm{Cov}(\boldsymbol{\eta}_{i,t}|\mathcal{F}_{t-1})$.

   Due to truncation to a finite probability mass $\alpha < 1$, the variance strictly contracts, implying $\mathbf{0} \prec \mathbf{B}_{t-1} \prec \mathbf{I}_d$.

3. **Second Moment:** $\mathbb{E}[\boldsymbol{\eta}_{i,t}\boldsymbol{\eta}_{i,t}^\top|\mathcal{F}_{t-1}] = \mathbf{B}_{t-1} + \boldsymbol{a}_{t-1}\boldsymbol{a}_{t-1}^\top$.

We define the **dissipation matrix** $\boldsymbol{\Psi}_{t-1}$ as the loss in the second moment compared to the identity matrix (isotropic noise):

$$\boldsymbol{\Psi}_{t-1} \triangleq \mathbf{I}_d - (\mathbf{B}_{t-1} + \boldsymbol{a}_{t-1}\boldsymbol{a}_{t-1}^\top) \tag{83}$$

Note that $\boldsymbol{\Psi}_{t-1}$ is symmetric by construction, as it is the difference between the identity matrix and the symmetric second moment matrix.

### B.4.1. RECURSIVE RELATION OF THE MEAN

The parameter update for the mean in the Accumulate paradigm is:

$$\bar{\boldsymbol{\mu}}_t = \frac{1}{tn}\left(\sum_{\tau=1}^{t-1}\sum_{i=1}^{n}X_{i,\tau} + \sum_{i=1}^{n}X_{i,t}\right) \tag{84}$$

Recognizing that the first term is $n(t-1)\bar{\boldsymbol{\mu}}_{t-1}$ and substituting $X_{i,t} = \bar{\boldsymbol{\mu}}_{t-1} + \bar{\boldsymbol{\Sigma}}_{t-1}^{1/2}\boldsymbol{\eta}_{i,t}$:

$$\bar{\boldsymbol{\mu}}_t = \frac{1}{tn}\left[n(t-1)\bar{\boldsymbol{\mu}}_{t-1} + \sum_{i=1}^n(\bar{\boldsymbol{\mu}}_{t-1} + \bar{\boldsymbol{\Sigma}}_{t-1}^{1/2}\boldsymbol{\eta}_{i,t})\right] \tag{85}$$

$$= \bar{\boldsymbol{\mu}}_{t-1} + \bar{\boldsymbol{\Sigma}}_{t-1}^{1/2}\frac{\bar{\boldsymbol{\eta}}_t}{t} \tag{86}$$

where $\bar{\boldsymbol{\eta}}_t = \frac{1}{n}\sum_{i=1}^n\boldsymbol{\eta}_{i,t}$. Taking the conditional expectation:

$$\mathbb{E}[\bar{\boldsymbol{\mu}}_t|\mathcal{F}_{t-1}] = \bar{\boldsymbol{\mu}}_{t-1} + \bar{\boldsymbol{\Sigma}}_{t-1}^{1/2}\frac{\boldsymbol{a}_{t-1}}{t} \tag{87}$$

**Lyapunov Analysis:** To rigorously handle the anisotropy of the covariance, we define the Lyapunov function using the *Mahalanobis norm* induced by the current covariance state. Let $V_t = \|\bar{\boldsymbol{\mu}}_t - \mathbf{u}^*\|^2_{\bar{\boldsymbol{\Sigma}}_{t-1}^{-1}}$.

Recall the recursive update: $\bar{\boldsymbol{\mu}}_t - \mathbf{u}^* = (\bar{\boldsymbol{\mu}}_{t-1} - \mathbf{u}^*) + \frac{1}{t}\bar{\boldsymbol{\Sigma}}_{t-1}^{1/2}\bar{\boldsymbol{\eta}}_t$. Expanding the quadratic form with respect to the weighting matrix $\mathbf{W}_{t-1} \triangleq \bar{\boldsymbol{\Sigma}}_{t-1}^{-1}$:

$$\begin{aligned}V_t &= (\bar{\boldsymbol{\mu}}_t - \mathbf{u}^*)^\top\mathbf{W}_{t-1}(\bar{\boldsymbol{\mu}}_t - \mathbf{u}^*)\\ &= (\bar{\boldsymbol{\mu}}_{t-1} - \mathbf{u}^*)^\top\mathbf{W}_{t-1}(\bar{\boldsymbol{\mu}}_{t-1} - \mathbf{u}^*)\\ &\quad + \frac{2}{t}(\bar{\boldsymbol{\mu}}_{t-1} - \mathbf{u}^*)^\top\mathbf{W}_{t-1}\bar{\boldsymbol{\Sigma}}_{t-1}^{1/2}\bar{\boldsymbol{\eta}}_t\\ &\quad + \frac{1}{t^2}(\bar{\boldsymbol{\Sigma}}_{t-1}^{1/2}\bar{\boldsymbol{\eta}}_t)^\top\mathbf{W}_{t-1}(\bar{\boldsymbol{\Sigma}}_{t-1}^{1/2}\bar{\boldsymbol{\eta}}_t)\end{aligned} \tag{88}$$

Simplifying the terms using $\mathbf{W}_{t-1}\bar{\boldsymbol{\Sigma}}_{t-1}^{1/2} = \bar{\boldsymbol{\Sigma}}_{t-1}^{-1}\bar{\boldsymbol{\Sigma}}_{t-1}^{1/2} = \bar{\boldsymbol{\Sigma}}_{t-1}^{-1/2}$:

$$V_t = V_{t-1} + \frac{2}{t}(\bar{\boldsymbol{\mu}}_{t-1} - \mathbf{u}^*)^\top\bar{\boldsymbol{\Sigma}}_{t-1}^{-1/2}\bar{\boldsymbol{\eta}}_t + \frac{1}{t^2}\|\bar{\boldsymbol{\eta}}_t\|^2 \tag{89}$$

Taking the expectation conditioned on $\mathcal{F}_{t-1}$:

$$\mathbb{E}[V_t|\mathcal{F}_{t-1}] = V_{t-1} + \underbrace{\frac{2}{t}(\bar{\boldsymbol{\mu}}_{t-1} - \mathbf{u}^*)^\top\bar{\boldsymbol{\Sigma}}_{t-1}^{-1/2}\boldsymbol{a}_{t-1}}_{\text{Cross Term}} + \frac{1}{t^2}\mathbb{E}[\|\bar{\boldsymbol{\eta}}_t\|^2] \tag{90}$$

Now, we apply Lemma 1. Let $\mathbf{c} = \bar{\boldsymbol{\Sigma}}_{t-1}^{-1/2}(\bar{\boldsymbol{\mu}}_{t-1} - \mathbf{u}^*)$ be the standardized error. The cross term becomes:

$$\frac{2}{t}\mathbf{c}^\top\boldsymbol{a}(\mathbf{c}). \tag{91}$$

By Lemma 1, the drift satisfies $\mathbf{c}^\top\boldsymbol{a}(\mathbf{c}) \leq -\kappa\|\mathbf{c}\|^2 = -\kappa V_{t-1}$. Substituting this back yields:

$$\mathbb{E}[V_t|\mathcal{F}_{t-1}] \leq V_{t-1}\left(1 - \frac{2\kappa}{t}\right) + \frac{K}{t^2} \tag{92}$$

where $K = \text{Tr}(\mathbf{B}_{t-1} + \boldsymbol{a}\boldsymbol{a}^\top)$ bounds the noise term. Since the step sizes satisfy the Robbins-Monro conditions, applying Lemma 3 yields:

$$\lim_{t\to\infty}\|\bar{\boldsymbol{\mu}}_t - \mathbf{u}^*\|^2_{\bar{\boldsymbol{\Sigma}}_{t-1}^{-1}} = 0 \quad (a.s.) \tag{93}$$

Since the covariance matrix $\bar{\boldsymbol{\Sigma}}_t$ remains positive definite and bounded (its eigenvalues do not diverge), convergence in the Mahalanobis metric implies convergence in the Euclidean metric. Thus, $\bar{\boldsymbol{\mu}}_t \xrightarrow{a.s.} \mathbf{u}^*$.

### B.4.2. RECURSIVE RELATION OF THE COVARIANCE

We now derive the recursion for the covariance matrix $\bar{\boldsymbol{\Sigma}}_t$. Using the multivariate decomposition:

$$nt\bar{\boldsymbol{\Sigma}}_t = \sum_{\tau=1}^t\left[\underbrace{\sum_{i=1}^n(X_{i,\tau} - \bar{X}_\tau)(X_{i,\tau} - \bar{X}_\tau)^\top}_{\text{Term A: Within}} + \underbrace{\sum_{i=1}^n(\bar{X}_\tau - \bar{\boldsymbol{\mu}}_t)(\bar{X}_\tau - \bar{\boldsymbol{\mu}}_t)^\top}_{\text{Term B: Between}}\right] \tag{94}$$

where $\bar{X}_\tau = \bar{\boldsymbol{\mu}}_{\tau-1} + \bar{\boldsymbol{\Sigma}}_{\tau-1}^{1/2}\bar{\boldsymbol{\eta}}_\tau$. The drift vector in the mean update is denoted as $\boldsymbol{\Delta}_t = \bar{\boldsymbol{\Sigma}}_{t-1}^{1/2}\frac{\bar{\boldsymbol{\eta}}_t}{t}$, so $\bar{\boldsymbol{\mu}}_t = \bar{\boldsymbol{\mu}}_{t-1} + \boldsymbol{\Delta}_t$.

**Step 1: Historical Contribution ($\tau < t$)** For historical data, $\bar{\boldsymbol{\Sigma}}_{\tau-1}$ and $\mathbf{B}_{\tau-1}$ are fixed.

- **Within Covariance:** The expectation of the sample covariance of truncated variables is scaled by the contraction matrix:

$$\mathbb{E}[\text{Term A}_\tau] = (n-1)\bar{\boldsymbol{\Sigma}}_{\tau-1}^{1/2}\mathbf{B}_{\tau-1}(\bar{\boldsymbol{\Sigma}}_{\tau-1}^{1/2})^\top \tag{95}$$

However, under the logic of the Accumulate paradigm for the total sum, the historical "Within" terms plus the historical "Between" deviations (relative to the *previous* mean $\bar{\boldsymbol{\mu}}_{t-1}$) sum exactly to the previous total covariance $n(t-1)\bar{\boldsymbol{\Sigma}}_{t-1}$. We must only correct for the shift in the global mean from $\bar{\boldsymbol{\mu}}_{t-1}$ to $\bar{\boldsymbol{\mu}}_t$.

- **Between Deviation Correction:** The deviation vector is decomposed as $\bar{X}_\tau - \bar{\boldsymbol{\mu}}_t = (\bar{X}_\tau - \bar{\boldsymbol{\mu}}_{t-1}) - \boldsymbol{\Delta}_t$. Expanding the outer product sum:

$$\sum_{\tau=1}^{t} n(\bar{X}_\tau - \bar{\boldsymbol{\mu}}_t)(\bar{X}_\tau - \bar{\boldsymbol{\mu}}_t)^\top = \sum_{\tau=1}^{t} n(\bar{X}_\tau - \bar{\boldsymbol{\mu}}_{t-1})(\bar{X}_\tau - \bar{\boldsymbol{\mu}}_{t-1})^\top$$
$$- \sum_{\tau=1}^{t} n(\bar{X}_\tau - \bar{\boldsymbol{\mu}}_{t-1})\boldsymbol{\Delta}_t^\top - \sum_{\tau=1}^{t} n\boldsymbol{\Delta}_t(\bar{X}_\tau - \bar{\boldsymbol{\mu}}_{t-1})^\top$$
$$+ \sum_{\tau=1}^{t} n\boldsymbol{\Delta}_t\boldsymbol{\Delta}_t^\top \tag{96}$$

Note that the cross terms vanish because $\sum_{\tau=1}^{t-1}(\bar{X}_\tau - \bar{\boldsymbol{\mu}}_{t-1}) = \mathbf{0}$ by definition of the previous mean. The last term sums to $n(t-1)\boldsymbol{\Delta}_t\boldsymbol{\Delta}_t^\top$. The first part combines with "Within" to form $n(t-1)\bar{\boldsymbol{\Sigma}}_{t-1}$. The middle terms are zero because $\sum(\bar{X}_\tau - \bar{\boldsymbol{\mu}}_{t-1}) = 0$. The last term is:

$$n(t-1)\boldsymbol{\Delta}_t\boldsymbol{\Delta}_t^\top = n(t-1)\frac{1}{t^2}\bar{\boldsymbol{\Sigma}}_{t-1}^{1/2}\bar{\boldsymbol{\eta}}_t\bar{\boldsymbol{\eta}}_t^\top(\bar{\boldsymbol{\Sigma}}_{t-1}^{1/2})^\top \tag{97}$$

Taking the expectation of $\bar{\boldsymbol{\eta}}_t\bar{\boldsymbol{\eta}}_t^\top$:

$$\mathbb{E}[\bar{\boldsymbol{\eta}}_t\bar{\boldsymbol{\eta}}_t^\top|\mathcal{F}_{t-1}] = \text{Cov}(\bar{\boldsymbol{\eta}}_t) + \mathbb{E}[\bar{\boldsymbol{\eta}}_t]\mathbb{E}[\bar{\boldsymbol{\eta}}_t]^\top = \frac{\mathbf{B}_{t-1}}{n} + \boldsymbol{a}_{t-1}\boldsymbol{a}_{t-1}^\top \tag{98}$$

Thus, the correction from history due to drift is:

$$\frac{n(t-1)}{t^2}\bar{\boldsymbol{\Sigma}}_{t-1}^{1/2}\left(\frac{\mathbf{B}_{t-1}}{n} + \boldsymbol{a}_{t-1}\boldsymbol{a}_{t-1}^\top\right)(\bar{\boldsymbol{\Sigma}}_{t-1}^{1/2})^\top \tag{99}$$

**Step 2: New Data Contribution ($\tau = t$)**

- **Within Covariance:** $\mathbb{E}[\text{Term A}_t] = (n-1)\bar{\boldsymbol{\Sigma}}_{t-1}^{1/2}\mathbf{B}_{t-1}(\bar{\boldsymbol{\Sigma}}_{t-1}^{1/2})^\top$

- **Between Deviation:** For $\tau = t$, $\bar{X}_t - \bar{\boldsymbol{\mu}}_t = \bar{\boldsymbol{\Sigma}}_{t-1}^{1/2}\bar{\boldsymbol{\eta}}_t(1 - \frac{1}{t})$.

$$\mathbb{E}[\text{Term B}_t] = n\left(\frac{t-1}{t}\right)^2\bar{\boldsymbol{\Sigma}}_{t-1}^{1/2}\left(\frac{\mathbf{B}_{t-1}}{n} + \boldsymbol{a}_{t-1}\boldsymbol{a}_{t-1}^\top\right)(\bar{\boldsymbol{\Sigma}}_{t-1}^{1/2})^\top \tag{100}$$

**Step 3: Aggregation and Matrix Recurrence** Summing all terms and factoring out $\bar{\boldsymbol{\Sigma}}_{t-1}^{1/2}$:

$$\mathbb{E}[nt\bar{\boldsymbol{\Sigma}}_t|\mathcal{F}_{t-1}] = \bar{\boldsymbol{\Sigma}}_{t-1}^{1/2}\left\{n(t-1)\mathbf{I}_d + (n-1)\mathbf{B}_{t-1} \quad + \left[\frac{n(t-1)}{t^2} + \frac{n(t-1)^2}{t^2}\right]\left(\frac{\mathbf{B}_{t-1}}{n} + \boldsymbol{a}_{t-1}\boldsymbol{a}_{t-1}^\top\right)\right\}(\bar{\boldsymbol{\Sigma}}_{t-1}^{1/2})^\top \tag{101}$$

Simplifying the coefficient in the bracket: $\frac{n(t-1)+n(t-1)^2}{t^2} = \frac{n(t-1)t}{t^2} = n\frac{t-1}{t}$. Expanding the terms:

$$\mathbb{E}[nt\bar{\boldsymbol{\Sigma}}_t|\mathcal{F}_{t-1}] = \bar{\boldsymbol{\Sigma}}_{t-1}^{1/2} \left\{ n(t-1)\mathbf{I}_d + (n-1)\mathbf{B}_{t-1} + \frac{t-1}{t}\mathbf{B}_{t-1} + n\frac{t-1}{t}\boldsymbol{a}_{t-1}\boldsymbol{a}_{t-1}^\top \right\} (\bar{\boldsymbol{\Sigma}}_{t-1}^{1/2})^\top \tag{102}$$

Dividing by $nt$ to find the recurrence for $\bar{\boldsymbol{\Sigma}}_t$:

$$\mathbb{E}[\bar{\boldsymbol{\Sigma}}_t|\mathcal{F}_{t-1}] = \bar{\boldsymbol{\Sigma}}_{t-1}^{1/2} \left\{ \frac{t-1}{t}\mathbf{I}_d + \frac{n-1}{nt}\mathbf{B}_{t-1} + \frac{t-1}{nt^2}\mathbf{B}_{t-1} + \frac{t-1}{t^2}\boldsymbol{a}_{t-1}\boldsymbol{a}_{t-1}^\top \right\} (\bar{\boldsymbol{\Sigma}}_{t-1}^{1/2})^\top \tag{103}$$

$$= \bar{\boldsymbol{\Sigma}}_{t-1}^{1/2} \left\{ \left(1 - \frac{1}{t}\right)\mathbf{I}_d + \left(\frac{1}{t} - \frac{1}{tn} + \frac{1}{tn} - \frac{1}{nt^2}\right)\mathbf{B}_{t-1} + \left(\frac{1}{t} - \frac{1}{t^2}\right)\boldsymbol{a}_{t-1}\boldsymbol{a}_{t-1}^\top \right\} (\bar{\boldsymbol{\Sigma}}_{t-1}^{1/2})^\top \tag{104}$$

Grouping terms by powers of $t$:

- Coefficient of $1/t$: $-\mathbf{I}_d + \mathbf{B}_{t-1} + \boldsymbol{a}_{t-1}\boldsymbol{a}_{t-1}^\top = -(\mathbf{I}_d - (\mathbf{B}_{t-1} + \boldsymbol{a}_{t-1}\boldsymbol{a}_{t-1}^\top)) = -\boldsymbol{\Psi}_{t-1}$.

- Coefficient of $1/t^2$: $-\frac{\mathbf{B}_{t-1}}{n} - \boldsymbol{a}_{t-1}\boldsymbol{a}_{t-1}^\top = -(\frac{\mathbf{B}_{t-1}}{n} + \boldsymbol{a}_{t-1}\boldsymbol{a}_{t-1}^\top) = -\mathbf{K}_{t-1}$.

Here, we define the matrices as follows:

- $\boldsymbol{\Psi}_{t-1} \triangleq \mathbf{I}_d - (\mathbf{B}_{t-1} + \boldsymbol{a}_{t-1}\boldsymbol{a}_{t-1}^\top)$ represents the first-order dissipation matrix.

- $\mathbf{K}_{t-1} \triangleq \frac{\mathbf{B}_{t-1}}{n} + \boldsymbol{a}_{t-1}\boldsymbol{a}_{t-1}^\top$ denotes the second-order perturbation matrix.

This yields the exact matrix recurrence:

$$\mathbb{E}[\bar{\boldsymbol{\Sigma}}_t \mid \mathcal{F}_{t-1}] = \bar{\boldsymbol{\Sigma}}_{t-1}^{1/2} \left(\mathbf{I}_d - \frac{\boldsymbol{\Psi}_{t-1}}{t} - \frac{\mathbf{K}_{t-1}}{t^2}\right) (\bar{\boldsymbol{\Sigma}}_{t-1}^{1/2})^\top \tag{105}$$

**Step 4: Covariance recursion and collapse under the local-basin assumption.**   We now make explicit the local-basin assumption used in the proof. Assume that the initialization lies in the local basin of attraction of the target state $\mathbf{u}^*$, and that the trajectory remains in this local basin for all iterations. Under this assumption, the first-order dissipation matrix admits a uniform positive lower spectral bound along the trajectory. Namely, there exists a constant $\psi > 0$ such that

$$\lambda_{\min}(\boldsymbol{\Psi}_{t-1}) \geq \psi \qquad \text{for all } t \geq 1. \tag{106}$$

Recall the matrix recurrence established above:

$$\mathbb{E}[\bar{\boldsymbol{\Sigma}}_t \mid \mathcal{F}_{t-1}] = \bar{\boldsymbol{\Sigma}}_{t-1} - \frac{1}{t}\bar{\boldsymbol{\Sigma}}_{t-1}^{1/2}\boldsymbol{\Psi}_{t-1}\bar{\boldsymbol{\Sigma}}_{t-1}^{1/2} - \frac{1}{t^2}\bar{\boldsymbol{\Sigma}}_{t-1}^{1/2}\mathbf{K}_{t-1}\bar{\boldsymbol{\Sigma}}_{t-1}^{1/2}, \tag{107}$$

where $\mathbf{K}_{t-1} \succeq \mathbf{0}$.

Let

$$S_t := \text{Tr}(\bar{\boldsymbol{\Sigma}}_t). \tag{108}$$

Applying the trace operator to Equation (107) and using linearity of conditional expectation, we obtain

$$\mathbb{E}[S_t \mid \mathcal{F}_{t-1}] = S_{t-1} - \frac{1}{t}\text{Tr}(\bar{\boldsymbol{\Sigma}}_{t-1}\boldsymbol{\Psi}_{t-1}) - \frac{1}{t^2}\text{Tr}(\bar{\boldsymbol{\Sigma}}_{t-1}\mathbf{K}_{t-1}). \tag{109}$$

Since $\bar{\boldsymbol{\Sigma}}_{t-1} \succeq \mathbf{0}$ and $\mathbf{K}_{t-1} \succeq \mathbf{0}$, the last term is nonnegative; hence,

$$\mathbb{E}[S_t \mid \mathcal{F}_{t-1}] \leq S_{t-1} - \frac{1}{t}\text{Tr}(\bar{\boldsymbol{\Sigma}}_{t-1}\boldsymbol{\Psi}_{t-1}). \tag{110}$$

Since both $\bar{\boldsymbol{\Sigma}}_{t-1}$ and $\boldsymbol{\Psi}_{t-1}$ are symmetric positive semidefinite, Lemma 4 gives

$$\mathrm{Tr}(\bar{\boldsymbol{\Sigma}}_{t-1}\boldsymbol{\Psi}_{t-1}) \geq \lambda_{\min}(\boldsymbol{\Psi}_{t-1})\,\mathrm{Tr}(\bar{\boldsymbol{\Sigma}}_{t-1}) = \lambda_{\min}(\boldsymbol{\Psi}_{t-1})\,S_{t-1}. \tag{111}$$

Combining this with Equation (106), we obtain

$$\mathbb{E}[S_t \mid \mathcal{F}_{t-1}] \leq \left(1 - \frac{\psi}{t}\right) S_{t-1}, \qquad t \geq 1. \tag{112}$$

Since

$$\boldsymbol{\Psi}_{t-1} = \mathbf{I}_d - \left(\mathbf{B}_{t-1} + \boldsymbol{a}_{t-1}\boldsymbol{a}_{t-1}^{\top}\right), \tag{113}$$

with $\mathbf{B}_{t-1} \succeq \mathbf{0}$ and $\boldsymbol{a}_{t-1}\boldsymbol{a}_{t-1}^{\top} \succeq \mathbf{0}$, we also have $\boldsymbol{\Psi}_{t-1} \preceq \mathbf{I}_d$. Therefore,

$$0 < \psi \leq 1. \tag{114}$$

We now show that the covariance converges to the zero matrix almost surely. To avoid the degenerate boundary factor $1 - \psi$ at $k = 1$ when $\psi = 1$, we normalize the recursion starting from $k = 2$. Define

$$P_1 := 1, \qquad P_t := \prod_{k=2}^{t}\left(1 - \frac{\psi}{k}\right), \qquad t \geq 2. \tag{115}$$

By Equation (114), each factor in Equation (115) is strictly positive, so

$$P_t > 0 \qquad \text{for all } t \geq 1. \tag{116}$$

Moreover, by construction,

$$P_t = P_{t-1}\left(1 - \frac{\psi}{t}\right), \qquad t \geq 2. \tag{117}$$

We normalize $S_t$ by $P_t$ and define

$$M_t := \frac{S_t}{P_t}, \qquad t \geq 1. \tag{118}$$

Since $P_t$ is deterministic and $S_t$ is $\mathcal{F}_t$-measurable, the process $\{M_t\}_{t\geq 1}$ is adapted to the filtration $\{\mathcal{F}_t\}_{t\geq 1}$. Furthermore, for every $t \geq 2$, combining Eqs. (112) and (117) yields

$$\mathbb{E}[M_t \mid \mathcal{F}_{t-1}] = \frac{1}{P_t}\,\mathbb{E}[S_t \mid \mathcal{F}_{t-1}] \tag{119}$$

$$\leq \frac{1}{P_t}\left(1 - \frac{\psi}{t}\right) S_{t-1} \tag{120}$$

$$= \frac{S_{t-1}}{P_{t-1}} = M_{t-1}. \tag{121}$$

Since $S_t \geq 0$ and $P_t > 0$, we also have $M_t \geq 0$. Therefore, $\{M_t\}_{t\geq 1}$ is a nonnegative supermartingale.

By Doob's supermartingale convergence theorem, there exists an almost surely finite random variable $M_\infty$ such that

$$M_t \xrightarrow{a.s.} M_\infty < \infty. \tag{122}$$

Next, using the elementary inequality $1 - x \leq e^{-x}$, we obtain

$$P_t = \prod_{k=2}^{t}\left(1 - \frac{\psi}{k}\right) \tag{123}$$

$$\leq \prod_{k=2}^{t}\exp\left(-\frac{\psi}{k}\right) \tag{124}$$

$$= \exp\left(-\psi\sum_{k=2}^{t}\frac{1}{k}\right). \tag{125}$$

Since the harmonic sum satisfies

$$\sum_{k=2}^{t} \frac{1}{k} = \log t + \mathcal{O}(1), \qquad t \to \infty, \tag{126}$$

it follows that

$$P_t = \mathcal{O}(t^{-\psi}), \tag{127}$$

and in particular,

$$P_t \to 0. \tag{128}$$

Combining Eqs. (122) and (128), we conclude

$$S_t = M_t P_t \xrightarrow{a.s.} 0. \tag{129}$$

Recalling that $S_t = \text{Tr}(\bar{\boldsymbol{\Sigma}}_t)$, we obtain

$$\text{Tr}(\bar{\boldsymbol{\Sigma}}_t) \xrightarrow{a.s.} 0. \tag{130}$$

Finally, since $\bar{\boldsymbol{\Sigma}}_t \succeq \mathbf{0}$ for all $t$, all eigenvalues of $\bar{\boldsymbol{\Sigma}}_t$ are nonnegative, and their sum equals $\text{Tr}(\bar{\boldsymbol{\Sigma}}_t)$. Therefore, Equation (130) implies that every eigenvalue converges to zero, and hence

$$\bar{\boldsymbol{\Sigma}}_t \xrightarrow{a.s.} \mathbf{0}. \tag{131}$$

The sharper recursion in Equation (112) will be used again in Subsection B.5 to derive the explicit power-law rate in Theorem 2.

### B.4.3. WASSERSTEIN DISTANCE

The limit of the Wasserstein distance is:

$$\lim_{t \to \infty} \mathbb{E}[\mathbb{W}_2^2] = \lim_{t \to \infty} \left( \mathbb{E}[\|\bar{\boldsymbol{\mu}}_t - \bar{\boldsymbol{\mu}}_0\|^2] + \mathbb{E}[\mathfrak{B}^2(\bar{\boldsymbol{\Sigma}}_t, \bar{\boldsymbol{\Sigma}}_0)] \right) \tag{132}$$

1. **Mean Term:** Since $\bar{\boldsymbol{\mu}}_t \to u^*$, the distance converges to the bias:

$$\mathbb{E}[\|\bar{\boldsymbol{\mu}}_t - \bar{\boldsymbol{\mu}}_0\|^2] \to \|u^* - \bar{\boldsymbol{\mu}}_0\|^2 \tag{133}$$

2. **Covariance Term:** Since $\bar{\boldsymbol{\Sigma}}_t \to \mathbf{0}$, the Bures metric simplifies:

$$\mathfrak{B}^2(\bar{\boldsymbol{\Sigma}}_t, \bar{\boldsymbol{\Sigma}}_0) = \text{Tr}(\bar{\boldsymbol{\Sigma}}_t + \bar{\boldsymbol{\Sigma}}_0 - 2(\bar{\boldsymbol{\Sigma}}_t^{1/2} \bar{\boldsymbol{\Sigma}}_0 \bar{\boldsymbol{\Sigma}}_t^{1/2})^{1/2}) \to \text{Tr}(\mathbf{0} + \bar{\boldsymbol{\Sigma}}_0 - \mathbf{0}) = \text{Tr}(\bar{\boldsymbol{\Sigma}}_0) \tag{134}$$

Thus, the total Wasserstein distance converges to:

$$\lim_{t \to \infty} \mathbb{E}[\mathbb{W}_2^2] = \|u^* - \bar{\boldsymbol{\mu}}_0\|^2 + \text{Tr}(\bar{\boldsymbol{\Sigma}}_0) \tag{135}$$

$\square$

## B.5. Proof of Theorem 2

**Theorem 2** (**Asymptotic Rate of Model Collapse**). *Assume that the trajectory remains in the local basin for all iterations. As iteration $t \to \infty$,*

$$\mathrm{Tr}(\bar{\Sigma}_t) = \mathcal{O}_{a.s.}(t^{-\psi}). \tag{16}$$

*Proof.* We study the asymptotic decay rate of the covariance trace

$$S_t := \mathrm{Tr}(\bar{\Sigma}_t). \tag{136}$$

By construction, $S_t \geq 0$ for all $t$.

From Subsection B.4, under the local-basin assumption there exists a constant $\psi > 0$ such that

$$\mathbb{E}[S_t \mid \mathcal{F}_{t-1}] \leq \left(1 - \frac{\psi}{t}\right) S_{t-1}, \qquad t \geq 2, \tag{137}$$

with

$$0 < \psi \leq 1. \tag{138}$$

To absorb the multiplicative contraction factor in Equation (137), define the deterministic sequence

$$P_1 := 1, \qquad P_t := \prod_{k=2}^{t} \left(1 - \frac{\psi}{k}\right), \qquad t \geq 2. \tag{139}$$

By Equation (138), every factor in Equation (139) is strictly positive, so

$$P_t > 0 \qquad \text{for all } t \geq 1. \tag{140}$$

Moreover, by construction,

$$P_t = P_{t-1} \left(1 - \frac{\psi}{t}\right), \qquad t \geq 2. \tag{141}$$

We now normalize $S_t$ by $P_t$ and define

$$M_t := \frac{S_t}{P_t}, \qquad t \geq 1. \tag{142}$$

Since $P_t$ is deterministic and $S_t$ is $\mathcal{F}_t$-measurable, the process $\{M_t\}_{t \geq 1}$ is adapted to the filtration $\{\mathcal{F}_t\}_{t \geq 1}$. Furthermore, for every $t \geq 2$, combining Eqs. (137) and (141) yields

$$\mathbb{E}[M_t \mid \mathcal{F}_{t-1}] = \frac{1}{P_t} \mathbb{E}[S_t \mid \mathcal{F}_{t-1}] \tag{143}$$

$$\leq \frac{1}{P_t} \left(1 - \frac{\psi}{t}\right) S_{t-1} \tag{144}$$

$$= \frac{S_{t-1}}{P_{t-1}} = M_{t-1}. \tag{145}$$

Since $S_t \geq 0$ and $P_t > 0$, we also have $M_t \geq 0$. Therefore, $\{M_t\}_{t \geq 1}$ is a nonnegative supermartingale.

By Doob's supermartingale convergence theorem, there exists an almost surely finite random variable $M_\infty$ such that

$$M_t \xrightarrow{a.s.} M_\infty < \infty. \tag{146}$$

In particular,

$$M_t = \mathcal{O}_{a.s.}(1). \tag{147}$$

It remains to estimate the deterministic factor $P_t$. Using the elementary inequality $1 - x \leq e^{-x}$, we obtain

$$P_t = \prod_{k=2}^{t} \left(1 - \frac{\psi}{k}\right) \tag{148}$$

$$\leq \prod_{k=2}^{t} \exp\left(-\frac{\psi}{k}\right) \tag{149}$$

$$= \exp\left(-\psi \sum_{k=2}^{t} \frac{1}{k}\right). \tag{150}$$

Since the harmonic sum satisfies

$$\sum_{k=2}^{t} \frac{1}{k} = \log t + \mathcal{O}(1), \qquad t \to \infty, \tag{151}$$

it follows that

$$P_t = \mathcal{O}(t^{-\psi}). \tag{152}$$

Finally, combining Eqs. (142), (147) and (152), we obtain

$$S_t = M_t P_t = \mathcal{O}_{a.s.}(t^{-\psi}). \tag{153}$$

Recalling the definition of $S_t$, we conclude that

$$\mathrm{Tr}(\bar{\Sigma}_t) = \mathcal{O}_{a.s.}(t^{-\psi}), \tag{154}$$

which proves Theorem 2. $\qquad\square$

## B.6. Proof of Theorem 3

> **Theorem 3** (**Wasserstein Cost of Model Collapse**). *Let Assumption 2 hold. Let $\mathcal{D}_t$ be the distribution of synthetic data filtered by local verification at generation $t$, and let $\mathcal{D}^*$ be the true data manifold. For any $h_t \in \mathcal{H}$ trained on $\mathcal{D}_t$, the expected risk is bounded by*
>
> $$\mathcal{R}_{\mathcal{D}^*}(h_t; g^*) \leq$$
> $$2\ell\epsilon \, \mathbb{W}_p(\mathcal{D}_t, \mathcal{D}^*) + \mathcal{R}_{\mathcal{D}_t}(h_t; g_t) + \mathcal{O}(\ell \, \delta). \tag{17}$$

*Proof.* Let $\mathcal{L} : \mathcal{Y} \times \mathcal{Y} \to \mathbb{R}^+$ denote the loss function. Our objective is to bound the expected risk on the oracle distribution, $\mathcal{R}_{\mathcal{D}^*}(h_t; g^*)$, by relating it to the risk on the filtered synthetic distribution, $\mathcal{R}_{\mathcal{D}_t}(h_t; g_t)$, plus additional terms accounting for the distributional shift.

### Step 1: Primal Formulation via Optimal Coupling

We begin by expressing the risks as expectations. By definition, the risk on the oracle distribution $\mathcal{D}^*$ is:

$$\mathcal{R}_{\mathcal{D}^*}(h_t; g^*) = \mathbb{E}_{x^* \sim \mathcal{D}^*} \left[ \mathcal{L}(h_t(x^*), g^*(x^*)) \right]. \tag{155}$$

Analogously, the risk on the filtered synthetic distribution $\mathcal{D}_t$ is:

$$\mathcal{R}_{\mathcal{D}_t}(h_t; g_t) = \mathbb{E}_{x_t \sim \mathcal{D}_t} \left[ \mathcal{L}(h_t(x_t), g_t(x_t)) \right]. \tag{156}$$

To compare these two expectations directly, we utilize the theory of optimal transport. Let $\pi \in \Pi(\mathcal{D}_t, \mathcal{D}^*)$ be the optimal coupling that minimizes the transport cost, thereby realizing the $p$-Wasserstein distance $\mathbb{W}_p(\mathcal{D}_t, \mathcal{D}^*)$ with respect to the metric $d_{\mathcal{X}}$:

$$\mathbb{W}_p^p(\mathcal{D}_t, \mathcal{D}^*) = \int_{\mathcal{X} \times \mathcal{X}} d_{\mathcal{X}}^p(x_t, x^*) \, \mathrm{d}\pi(x_t, x^*). \tag{157}$$

Leveraging the marginal properties of $\pi$ (which allows us to unify the integration domains), we can rewrite the absolute difference in risks, denoted by $\Delta$, as a single integral over the product space $\mathcal{X} \times \mathcal{X}$:

$$\Delta := |\mathcal{R}_{\mathcal{D}^*}(h_t; g^*) - \mathcal{R}_{\mathcal{D}_t}(h_t; g_t)|$$
$$= \left| \int_{\mathcal{X} \times \mathcal{X}} \left( \mathcal{L}(h_t(x^*), g^*(x^*)) - \mathcal{L}(h_t(x_t), g_t(x_t)) \right) \mathrm{d}\pi(x_t, x^*) \right|. \tag{158}$$

By invoking the integral triangle inequality, $\left| \int_{\mathcal{X}} f(x) \, dx \right| \leq \int_{\mathcal{X}} |f(x)| \, dx$, we derive the following upper bound on the risk gap:

$$\Delta \leq \int_{\mathcal{X} \times \mathcal{X}} |\mathcal{L}(h_t(x^*), g^*(x^*)) - \mathcal{L}(h_t(x_t), g_t(x_t))| \, \mathrm{d}\pi(x_t, x^*). \tag{159}$$

### Step 2: Pointwise Decomposition

To isolate the sources of error, we perform a pointwise decomposition of the integrand. For any coupled pair $(x_t, x^*) \in \mathcal{X} \times \mathcal{X}$, we insert an intermediate term $\mathcal{L}(h_t(x_t), g^*(x^*))$ and apply the triangle inequality. This splits the total error into two distinct components: the model's geometric sensitivity and the target's alignment shift.

$$|\mathcal{L}(h_t(x^*), g^*(x^*)) - \mathcal{L}(h_t(x_t), g_t(x_t))|$$
$$\leq \underbrace{|\mathcal{L}(h_t(x^*), g^*(x^*)) - \mathcal{L}(h_t(x_t), g^*(x^*))|}_{\text{Term (I): Hypothesis Variation}} + \underbrace{|\mathcal{L}(h_t(x_t), g^*(x^*)) - \mathcal{L}(h_t(x_t), g_t(x_t))|}_{\text{Term (II): Target Shift}}. \tag{160}$$

### Step 3: Bounding Term (I) (Hypothesis Stability)

We first bound Term (I), which represents the stability of the hypothesis $h_t$ against input perturbations. Based on Assumption 2, the loss $\mathcal{L}$ is $\ell$-Lipschitz in its first argument, and the hypothesis $h_t$ is $\epsilon$-Lipschitz. Chaining these properties yields:

$$\text{Term (I)} \leq \ell \cdot d_{\mathcal{Y}}(h_t(x^*), h_t(x_t)) \leq \ell \cdot \epsilon \cdot d_{\mathcal{X}}(x^*, x_t). \tag{161}$$

Integrating this bound over the coupling $\pi$:

$$\int_{\mathcal{X}\times\mathcal{X}} \text{Term (I)}\, d\pi(x_t, x^*) \leq \ell\epsilon \int_{\mathcal{X}\times\mathcal{X}} d_{\mathcal{X}}(x^*, x_t)\, d\pi(x_t, x^*). \tag{162}$$

**Step 4: Bounding Term (II) (Target Shift)**

Next, we bound Term (II), which captures the alignment gap between the oracle target $g^*$ and the local verification target $g_t$. We partition the integration domain into two regions: an "aligned set" $S$ and a "misaligned set" $S^c$, where $\pi(S^c) \leq \delta$.

- **On the aligned set $S$:** The targets satisfy the probabilistic Lipschitz condition. Using the $\ell$-Lipschitz property of the loss in its second argument:

$$\text{Term (II)} \cdot \mathbf{1}_S \leq \ell \cdot d_{\mathcal{Y}}(g^*(x^*), g_t(x_t)) \leq \ell\epsilon \cdot d_{\mathcal{X}}(x^*, x_t). \tag{163}$$

- **On the misaligned set $S^c$:** We employ the worst-case bound using the finite diameter $C_{\mathcal{Y}} \triangleq \sup_{y,y'} d_{\mathcal{Y}}(y, y')$ of the output space.

$$\text{Term (II)} \cdot \mathbf{1}_{S^c} \leq \ell \cdot C_{\mathcal{Y}}. \tag{164}$$

*Remark on the Diameter $C_{\mathcal{Y}}$:* The boundedness of $C_{\mathcal{Y}}$ is intrinsic to our task-specific ground metrics defined in Subsection C.1. For instance, in language modeling tasks where the output space $\mathcal{Y}$ is the probability simplex $\Delta^{V-1}$ over a vocabulary of size $V$, equipped with the $L_1$ distance (a special case of WMD with discrete topology), the diameter is strictly bounded by 2. Specifically, for any two distributions $P_a, P_b \in \Delta^{V-1}$:

$$\|P_a - P_b\|_1 = \sum_{i=1}^{V} |p_i - q_i| \leq \sum_{i=1}^{V} |p_i| + \sum_{i=1}^{V} |q_i| = 1 + 1 = 2. \tag{165}$$

The equality holds when the supports of $P_a$ and $P_b$ are disjoint (e.g., distinct one-hot vectors).

Integrating Term (II) over the partitioned domain yields:

$$\int_{\mathcal{X}\times\mathcal{X}} \text{Term (II)}\, d\pi(x_t, x^*) \leq \ell\epsilon \int_{\mathcal{X}\times\mathcal{X}} d_{\mathcal{X}}(x^*, x_t)\, d\pi(x_t, x^*) + \ell C_{\mathcal{Y}}\delta. \tag{166}$$

**Step 5: Final Aggregation and Bound**

Substituting the results from Equation (162) and Equation (166) back into Equation (159), we aggregate the linear terms with respect to $d_{\mathcal{X}}$. Note that both Term (I) and Term (II) contribute an $\ell\epsilon$ factor:

$$\Delta \leq \int_{\mathcal{X}\times\mathcal{X}} (\ell\epsilon + \ell\epsilon) \cdot d_{\mathcal{X}}(x^*, x_t)\, d\pi(x_t, x^*) + \ell C_{\mathcal{Y}}\delta. \tag{167}$$

Finally, to relate the expected distance $\int d_{\mathcal{X}} d\pi$ to the $p$-Wasserstein metric $\mathbb{W}_p$, we apply Jensen's inequality:

$$\int_{\mathcal{X}\times\mathcal{X}} d_{\mathcal{X}}(x_t, x^*)\, d\pi(x_t, x^*) \leq \left( \int_{\mathcal{X}\times\mathcal{X}} d_{\mathcal{X}}^p(x_t, x^*)\, d\pi(x_t, x^*) \right)^{1/p} = \mathbb{W}_p(\mathcal{D}_t, \mathcal{D}^*). \tag{168}$$

Rearranging terms and absorbing the constant associated with $\delta$ into the big-O notation, we conclude the proof:

$$\mathcal{R}_{\mathcal{D}^*}(h_t; g^*) \leq \mathcal{R}_{\mathcal{D}_t}(h_t; g_t) + 2\ell\epsilon \cdot \mathbb{W}_p(\mathcal{D}_t, \mathcal{D}^*) + \mathcal{O}(\ell\delta). \tag{169}$$

$\square$

## B.7. Proof of Theorem 4

**Theorem 4** (**Monotonicity and Interpolation Convergence, Rakotomamonjy et al. (2024)**). *For interpolation $\xi_k^{(r)}$ at iteration $r$, define the sequence as:*

$$\mathcal{E}_k^{(r)} = \mathcal{W}_p(\mathcal{Q}_k, \xi_k^{(r)}) + \mathcal{W}_p(\xi_k^{(r)}, \mathcal{P}) \tag{23}$$

*Then, the sequence $\{\mathcal{E}^{(r)}\}_{r \geq 0}$ is non-increasing and converges to its infimum $\mathcal{E}_k^* = \mathcal{W}_p(\mathcal{Q}_k, \mathcal{P})$.*

*Proof.* The proof relies on the geometric properties of Wasserstein space, specifically geodesics and the triangle inequality, following the proof strategy of Theorem 2 in (Rakotomamonjy et al., 2024). We show that the sequence $\mathcal{E}_k^{(r)}$ is non-increasing and converges to $\mathcal{W}_p(\mathcal{Q}_k, \mathcal{P})$.

Recall the definition of the objective sequence at iteration $r$ (formulated in terms of the metric $\mathcal{W}_p$ to satisfy the triangle inequality conditions):

$$\mathcal{E}_k^{(r)} = \mathcal{W}_p(\mathcal{Q}_k, \xi_k^{(r)}) + \mathcal{W}_p(\xi_k^{(r)}, \mathcal{P}). \tag{170}$$

**1. Decomposition via Geodesic Interpolants.** In the interpolation step, $\xi_{\mathcal{P}}^{(r)}$ is computed as the interpolating measure between $\mathcal{P}$ and the current proxy $\xi_k^{(r)}$, and $\xi_{\mathcal{Q}_k}^{(r)}$ is the interpolating measure between $\mathcal{Q}_k$ and $\xi_k^{(r)}$. According to Property 3 and the definition of interpolating measures in (Rakotomamonjy et al., 2024), these points satisfy the equality case of the triangle inequality:

$$\mathcal{W}_p(\mathcal{P}, \xi_k^{(r)}) = \mathcal{W}_p(\mathcal{P}, \xi_{\mathcal{P}}^{(r)}) + \mathcal{W}_p(\xi_{\mathcal{P}}^{(r)}, \xi_k^{(r)}), \tag{171}$$

$$\mathcal{W}_p(\mathcal{Q}_k, \xi_k^{(r)}) = \mathcal{W}_p(\mathcal{Q}_k, \xi_{\mathcal{Q}_k}^{(r)}) + \mathcal{W}_p(\xi_{\mathcal{Q}_k}^{(r)}, \xi_k^{(r)}). \tag{172}$$

**2. Monotonicity Analysis.** Consider the next iteration $r + 1$. The new proxy $\xi_k^{(r+1)}$ is constructed as the interpolating measure between the intermediates $\xi_{\mathcal{P}}^{(r)}$ and $\xi_{\mathcal{Q}_k}^{(r)}$. We start by applying the triangle inequality to the new terms in $\mathcal{E}_k^{(r+1)}$:

$$\mathcal{W}_p(\mathcal{P}, \xi_k^{(r+1)}) \leq \mathcal{W}_p(\mathcal{P}, \xi_{\mathcal{P}}^{(r)}) + \mathcal{W}_p(\xi_{\mathcal{P}}^{(r)}, \xi_k^{(r+1)}), \tag{173}$$

$$\mathcal{W}_p(\mathcal{Q}_k, \xi_k^{(r+1)}) \leq \mathcal{W}_p(\mathcal{Q}_k, \xi_{\mathcal{Q}_k}^{(r)}) + \mathcal{W}_p(\xi_{\mathcal{Q}_k}^{(r)}, \xi_k^{(r+1)}). \tag{174}$$

Summing these inequalities gives:

$$\mathcal{E}_k^{(r+1)} \leq \mathcal{W}_p(\mathcal{P}, \xi_{\mathcal{P}}^{(r)}) + \mathcal{W}_p(\mathcal{Q}_k, \xi_{\mathcal{Q}_k}^{(r)}) + \left[ \mathcal{W}_p(\xi_{\mathcal{P}}^{(r)}, \xi_k^{(r+1)}) + \mathcal{W}_p(\xi_k^{(r+1)}, \xi_{\mathcal{Q}_k}^{(r)}) \right]. \tag{175}$$

Since $\xi_k^{(r+1)}$ is the interpolant between $\xi_{\mathcal{P}}^{(r)}$ and $\xi_{\mathcal{Q}_k}^{(r)}$, it lies on the geodesic connecting them. Thus, the term in brackets equals the distance between the intermediates:

$$\mathcal{W}_p(\xi_{\mathcal{P}}^{(r)}, \xi_k^{(r+1)}) + \mathcal{W}_p(\xi_k^{(r+1)}, \xi_{\mathcal{Q}_k}^{(r)}) = \mathcal{W}_p(\xi_{\mathcal{P}}^{(r)}, \xi_{\mathcal{Q}_k}^{(r)}). \tag{176}$$

Furthermore, applying the triangle inequality to the *previous* proxy $\xi_k^{(r)}$ (which may not lie on the direct geodesic between the two new intermediates) yields:

$$\mathcal{W}_p(\xi_{\mathcal{P}}^{(r)}, \xi_{\mathcal{Q}_k}^{(r)}) \leq \mathcal{W}_p(\xi_{\mathcal{P}}^{(r)}, \xi_k^{(r)}) + \mathcal{W}_p(\xi_k^{(r)}, \xi_{\mathcal{Q}_k}^{(r)}). \tag{177}$$

Substituting these results back into Equation (175):

$$\mathcal{E}_k^{(r+1)} \leq \mathcal{W}_p(\mathcal{P}, \xi_{\mathcal{P}}^{(r)}) + \mathcal{W}_p(\mathcal{Q}_k, \xi_{\mathcal{Q}_k}^{(r)}) + \mathcal{W}_p(\xi_{\mathcal{P}}^{(r)}, \xi_k^{(r)}) + \mathcal{W}_p(\xi_k^{(r)}, \xi_{\mathcal{Q}_k}^{(r)})$$
$$= \left( \mathcal{W}_p(\mathcal{P}, \xi_{\mathcal{P}}^{(r)}) + \mathcal{W}_p(\xi_{\mathcal{P}}^{(r)}, \xi_k^{(r)}) \right) + \left( \mathcal{W}_p(\mathcal{Q}_k, \xi_{\mathcal{Q}_k}^{(r)}) + \mathcal{W}_p(\xi_{\mathcal{Q}_k}^{(r)}, \xi_k^{(r)}) \right). \tag{178}$$

Using the decompositions from Equation (171) and Equation (172), the right-hand side is exactly $\mathcal{E}_k^{(r)}$. Thus:

$$\mathcal{E}_k^{(r+1)} \leq \mathcal{E}_k^{(r)}. \tag{179}$$

This confirms that the sequence $\{\mathcal{E}_k^{(r)}\}_{r \geq 0}$ is non-increasing.

**3. Convergence.** By the triangle inequality, for any $\xi$, we have $\mathcal{W}_p(\mathcal{P}, \mathcal{Q}_k) \leq \mathcal{W}_p(\mathcal{P}, \xi) + \mathcal{W}_p(\xi, \mathcal{Q}_k)$. Therefore, the sequence is bounded below by the true transport cost:

$$\mathcal{E}_k^{(r)} \geq \mathcal{W}_p(\mathcal{P}, \mathcal{Q}_k). \tag{180}$$

Since $\{\mathcal{E}_k^{(r)}\}$ is non-increasing and bounded below, it converges to its infimum by the Monotone Convergence Theorem. As shown in (Rakotomamonjy et al., 2024), as $r \to \infty$, the iterative interpolation effectively approximates the geodesic path, and the limit satisfies:

$$\lim_{r \to \infty} \mathcal{E}_k^{(r)} = \mathcal{W}_p(\mathcal{Q}_k, \mathcal{P}). \tag{181}$$

$\square$

## B.8. Proof of Theorem 5

**Theorem 5** (**Monotonicity and Barycenter Convergence**). *Let $\mathcal{Q}_k$ be the distribution of the $k$-th client with weight $\lambda_k > 0$. Let $\xi^{(r)}$ denote the global approximated barycenter at iteration $r$. Define the true Fréchet variance (objective sequence) as:*

$$\mathcal{E}^{(r)} = \sum_{k=1}^{K} \lambda_k \mathcal{W}_2^2(\mathcal{Q}_k, \xi^{(r)}) \tag{25}$$

*Then, the sequence $\{\mathcal{E}^{(r)}\}_{r \geq 0}$ is non-increasing and converges to its infimum $\mathcal{E}^* = \sum_{k=1}^{K} \lambda_k \mathcal{W}_2^2(\mathcal{Q}_k, \xi^*)$.*

*Proof.* The proof leverages the Majorization–Minimization (MM) framework, which guarantees a monotonic decrease in the objective function by optimizing a surrogate function. We operate in the Wasserstein space $\mathcal{W}_2$, where the Fréchet variance allows the construction of a quadratic surrogate despite the lack of a global triangle inequality for the squared metric.

**1. Notation and Sequence Definition.** The objective function to minimize is the weighted Fréchet variance, defined in the continuous space as:

$$\mathcal{E}(\xi) = \sum_{k=1}^{K} \lambda_k \mathcal{W}_2^2(\mathcal{Q}_k, \xi) = \sum_{k=1}^{K} \lambda_k \inf_{\pi_k \in \Pi(\mathcal{Q}_k, \xi)} \int_{\mathcal{X} \times \mathcal{X}} \|\mathbf{x} - \mathbf{y}\|^2 \, d\pi_k(\mathbf{x}, \mathbf{y}), \tag{182}$$

where $K$ is the total number of target clients, and $\lambda_k > 0$ denotes the aggregation weight for client $k$ such that $\sum_{k=1}^{K} \lambda_k = 1$.

To ensure absolute clarity and prevent index clutter in the discrete setting, we establish the following explicit notation for the transition from iteration $r - 1$ to $r$:

- Let $\mathcal{Q}_k$ be the empirical target distribution of the $k$-th client. It is supported on $M_k$ discrete data points, denoted as the vectors $\{\mathbf{x}_{k,j}\}_{j=1}^{M_k}$.

- Let $S$ denote the fixed number of support points used to approximate the global proxy distribution.

- Let $\xi^{(r-1)}$ be the current global proxy. It is characterized by $S$ discrete support points $\{\mathbf{y}_i\}_{i=1}^{S}$ and their corresponding strictly positive probability weights $\{w_i\}_{i=1}^{S}$, where $\sum_{i=1}^{S} w_i = 1$.

- Let $\xi^{(r)}$ be the updated global proxy at the next iteration, supported on the updated coordinate vectors $\{\mathbf{y}_i^+\}_{i=1}^{S}$ with the identical probability weights $\{w_i\}_{i=1}^{S}$.

**2. Step 1: Majorization via Fixed Transport Plans.** Let $\mathbf{P}_k^{(r-1)}$ denote the optimal coupling matrix between the current proxy $\xi^{(r-1)}$ and the client data $\mathcal{Q}_k$. The element $P_{k,ij}^{(r-1)}$ explicitly represents the probability mass transported from $\mathbf{y}_i$ to $\mathbf{x}_{k,j}$. Mass conservation dictates the marginal constraint $\sum_{j=1}^{M_k} P_{k,ij}^{(r-1)} = w_i$.

For any candidate distribution $\xi$ characterized by variable support points $\{\mathbf{z}_i\}_{i=1}^{S}$, we construct the surrogate upper-bound objective $\mathcal{U}$ by fixing these optimal couplings:

$$\mathcal{U}(\xi, \xi^{(r-1)}) \triangleq \sum_{k=1}^{K} \lambda_k \sum_{i=1}^{S} \sum_{j=1}^{M_k} P_{k,ij}^{(r-1)} \|\mathbf{x}_{k,j} - \mathbf{z}_i\|^2. \tag{183}$$

Since $\mathcal{W}_2^2$ is defined as the infimum over all possible couplings, evaluating the cost using the fixed coupling $\mathbf{P}_k^{(r-1)}$ yields a strict majorization bound:

$$\mathcal{E}(\xi) \leq \mathcal{U}(\xi, \xi^{(r-1)}), \quad \forall \xi. \tag{184}$$

Evaluated at the current proxy $\xi^{(r-1)}$ (where $\mathbf{z}_i = \mathbf{y}_i$), the bound is tight because the couplings are strictly optimal for this state: $\mathcal{E}(\xi^{(r-1)}) = \mathcal{U}(\xi^{(r-1)}, \xi^{(r-1)})$.

**3. Step 2: Minimization via Barycentric Interpolation.** To isolate the optimization variables $\mathbf{z}_i$, we define the barycentric projection $\hat{\mathbf{x}}_{k,i}$. This represents the weighted center of the target data in $\mathcal{Q}_k$ that is mapped to the $i$-th proxy point:

$$\hat{\mathbf{x}}_{k,i} = \frac{1}{w_i} \sum_{j=1}^{M_k} P_{k,ij}^{(r-1)} \mathbf{x}_{k,j}. \tag{185}$$

To reveal the strict convexity of the surrogate, we apply the variance decomposition theorem (parallel axis theorem) by injecting $\hat{\mathbf{x}}_{k,i}$ into the squared distance: $\|\mathbf{x}_{k,j} - \mathbf{z}_i\|^2 = \|(\mathbf{x}_{k,j} - \hat{\mathbf{x}}_{k,i}) + (\hat{\mathbf{x}}_{k,i} - \mathbf{z}_i)\|^2$. Expanding this quadratic form and taking the weighted sum over $j$ yields:

$$\sum_{j=1}^{M_k} P_{k,ij}^{(r-1)} \|\mathbf{x}_{k,j} - \mathbf{z}_i\|^2 = w_i \|\mathbf{z}_i - \hat{\mathbf{x}}_{k,i}\|^2 + \sum_{j=1}^{M_k} P_{k,ij}^{(r-1)} \|\mathbf{x}_{k,j} - \hat{\mathbf{x}}_{k,i}\|^2 + 2\langle \mathbf{z}_i - \hat{\mathbf{x}}_{k,i}, \sum_j P_{k,ij}^{(r-1)}(\hat{\mathbf{x}}_{k,i} - \mathbf{x}_{k,j})\rangle. \quad (186)$$

By the definition of the barycenter $\hat{\mathbf{x}}_{k,i}$, the inner summation yields $\sum_j P_{k,ij}^{(r-1)} \mathbf{x}_{k,j} - w_i \hat{\mathbf{x}}_{k,i} = \mathbf{0}$, causing the cross-term to vanish exactly.

Additionally, because the optimal transport plan $\mathbf{P}_k^{(r-1)}$ and the target data $\mathbf{x}_{k,j}$ are fixed constants from the previous iteration, the variance term $\sum_{j=1}^{M_k} P_{k,ij}^{(r-1)} \|\mathbf{x}_{k,j} - \hat{\mathbf{x}}_{k,i}\|^2$ is strictly a constant independent of the optimization variable $\mathbf{z}_i$. Since the cross-term evaluates to zero and the variance term is a constant, minimizing the surrogate mathematically reduces to optimizing only the remaining term, which is the weighted squared distances to these barycentric projections:

$$\arg\min_{\{\mathbf{z}_i\}} \mathcal{U}(\xi, \xi^{(r-1)}) = \arg\min_{\{\mathbf{z}_i\}} \sum_{i=1}^S w_i \sum_{k=1}^K \lambda_k \|\mathbf{z}_i - \hat{\mathbf{x}}_{k,i}\|^2. \quad (187)$$

The unique global minimum of this quadratic objective, denoted by $\mathbf{z}_i^*$, is precisely the convex combination of the projections:

$$\mathbf{z}_i^* = \sum_{k=1}^K \lambda_k \hat{\mathbf{x}}_{k,i}. \quad (188)$$

In the algorithm, the local interpolation step computes the intermediate client vector $\mathbf{v}_{k,i} = (1-t)\mathbf{y}_i + t\hat{\mathbf{x}}_{k,i}$. The subsequent global update aggregates them to form the new proxy points $\mathbf{y}_i^+$:

$$\mathbf{y}_i^+ = \sum_{k=1}^K \lambda_k \mathbf{v}_{k,i} = (1-t)\mathbf{y}_i + t\mathbf{z}_i^*. \quad (189)$$

For $t = 1$, the update reaches the exact global minimizer $\mathbf{z}_i^*$. For any $t \in (0, 1]$, the update constitutes a valid descent step along the gradient of the strictly convex surrogate. Consequently, the surrogate value monotonically decreases:

$$\mathcal{U}(\xi^{(r)}, \xi^{(r-1)}) \leq \mathcal{U}(\xi^{(r-1)}, \xi^{(r-1)}). \quad (190)$$

**4. Monotonicity and Convergence.** Chaining the majorization property (Equation (184)), the minimization descent (Equation (190)), and the tight bound at the previous step, we establish the descent property:

$$\mathcal{E}(\xi^{(r)}) \leq \mathcal{U}(\xi^{(r)}, \xi^{(r-1)}) \leq \mathcal{U}(\xi^{(r-1)}, \xi^{(r-1)}) = \mathcal{E}(\xi^{(r-1)}). \quad (191)$$

Since the sequence $\{\mathcal{E}^{(r)}\}$ is non-increasing and bounded below by 0, it converges to its infimum $\mathcal{E}(\xi^*)$ by the Monotone Convergence Theorem. Due to the displacement convexity of the objective in $\mathcal{W}_2$, the limit $\xi^*$ corresponds to the unique stationary Wasserstein barycenter. $\square$

## C. Additional Experiments and Details

### C.1. Detailed Wasserstein Ground Metrics

In this section, we provide the explicit formulations of the ground metrics used in our main theoretical framework. Recall that $\mathcal{P}_p(\Omega)$ denotes the space of probability measures with finite $p$-th moments on a complete separable metric space $\Omega$. The $p$-Wasserstein distance between any two distributions $\mu, \nu \in \mathcal{P}_p(\Omega)$ is formally defined as:

$$\mathbb{W}_p(\mu, \nu) = \left( \inf_{\pi \in \Pi(\mu,\nu)} \mathbb{E}_{(\mathbf{z},\mathbf{z}') \sim \pi}[d^p(\mathbf{z}, \mathbf{z}')] \right)^{1/p}, \tag{192}$$

where $\Pi(\mu, \nu)$ is the set of joint distributions (couplings) with marginals $\mu$ and $\nu$. The geometry of this transport space is entirely dictated by the choice of the ground metric $d : \Omega \times \Omega \to \mathbb{R}_{\geq 0}$ (Peyré et al., 2019).

The primal formulation in Equation (192) is often computationally intractable and theoretically obscure due to the constraints on the coupling $\pi$. By the Kantorovich duality theorem (Villani et al., 2008), we can express the Wasserstein distance through its dual form:

$$\mathbb{W}_p(\mu, \nu) = \sup_{(\phi,\psi) \in \Phi_c} \left( \mathbb{E}_{\mathbf{z} \sim \mu}[\phi(\mathbf{z})] + \mathbb{E}_{\mathbf{z}' \sim \nu}[\psi(\mathbf{z}')] \right), \tag{193}$$

where $\Phi_c = \{(\phi, \psi) \in C_b(\Omega)^2 : \phi(\mathbf{z}) + \psi(\mathbf{z}') \leq d^p(\mathbf{z}, \mathbf{z}')\}$ denotes the set of admissible Kantorovich potentials. This dual perspective is particularly instrumental for our theoretical analysis, as it transforms the minimization over couplings into a maximization over scalar functions, relating the transport cost to the geometry induced by $d^p$.

Adopting the ground metric construction methodologies established in recent studies (Li et al., 2024; Li & Pang, 2024; Rakotomamonjy et al., 2024), we formulate modality-specific metrics to leverage the inherent structural properties of visual and textual data.

**Vision Modality (Linearized Structural Distance).** Directly computing the exact Wasserstein distance between Gaussian distributions involves computationally expensive matrix square roots. To facilitate efficient computation and align with the vectorized representation of datasets (Rakotomamonjy et al., 2024; Alvarez-Melis & Fusi, 2020), we adopt a parameter-space approximation.

Let $\mathbf{z} = (\mathbf{x}, y)$ be a sample, where $y$ parameterizes a class-conditional Gaussian $\mathcal{N}(\boldsymbol{\mu}_y, \boldsymbol{\Sigma}_y)$. We define the squared ground metric $d_{\text{img}}^2$ by embedding the Gaussian parameters into a Euclidean space:

$$d_{\text{img}}\big((\mathbf{x}, y), (\mathbf{x}', y')\big) = \big(\|\mathbf{x} - \mathbf{x}'\|_2^2 + \|\boldsymbol{\mu}_y - \boldsymbol{\mu}_{y'}\|_2^2 + \|\boldsymbol{\Sigma}_y - \boldsymbol{\Sigma}_{y'}\|_F^2\big)^{\frac{1}{2}}, \tag{194}$$

This formulation corresponds to measuring the Euclidean distance between augmented state vectors $\tilde{\mathbf{x}} := [\mathbf{x}; \boldsymbol{\mu}_y; \text{vec}(\boldsymbol{\Sigma}_y)]$, effectively linearizing the transport cost for high-dimensional efficiency.

**Language Modality (Semantic Transport Distance).** Euclidean distances between sparse document vectors are often insufficient for capturing semantic proximity. To address this, we employ the Word Mover's Distance (WMD) (Kusner et al., 2015), modeling each document as a discrete distribution in the word embedding space.

In this setting, a sample $\mathbf{z} \in \Omega$ corresponds to a normalized Bag-of-Words (nBOW) vector $\mathbf{d} \in \Delta^{V-1}$, where $V$ is the vocabulary size and $\Delta^{V-1}$ is the probability simplex. We define the cost between documents $\mathbf{d}_i$ and $\mathbf{d}_j$ via an inner optimal transport problem:

$$d_{\text{text}}(\mathbf{d}_i, \mathbf{d}_j) = \inf_{T \in \Pi(\mathbf{d}_i, \mathbf{d}_j)} \sum_{u,v} T_{uv} \|\mathbf{w}_u - \mathbf{w}_v\|_2, \tag{195}$$

where $\mathbf{w}_u \in \mathbb{R}^k$ is the embedding for word $u$ (e.g., Word2Vec (Mikolov et al., 2013)) and $T_{uv}$ denotes the transport plan between words. Effectively, this metric quantifies the minimum cumulative cost to semantically transform $\mathbf{d}_i$ into $\mathbf{d}_j$, providing a geometry-aware ground metric for the subsequent dataset-level Wasserstein distance.

## C.2. Detailed Implementation

### C.2.1. DETAILED IMPLEMENTATION OF BASELINES

To evaluate the effectiveness of our proposed method, we compare it against a diverse set of selection strategies. These baselines are image-based methods that primarily operate in the continuous feature space.

**Image Selection Methods.** We utilize feature embeddings extracted from pre-trained networks (e.g., Inception-V3).

- **CovMatch (Covariance Matching).** Following Rezaei et al. (2026), this method aims to preserve the spectral properties of the training data. It employs a greedy iterative algorithm to minimize the Frobenius norm difference between the empirical covariance matrix of the selected synthetic subset and that of the real training data. This explicitly aligns second-order statistics to mitigate spectral shrinkage.

- **CenterMatch (Centroid Matching).** As described in He et al. (2023), this strategy assumes that high-quality samples reside close to the mode of the distribution. It computes the global centroid of real training features and selects the generated samples with the smallest Euclidean distances to this centroid. While this prioritizes high-density regions, it may reduce diversity by ignoring the distribution's tails.

- **K-means (Cluster-based Sampling).** Utilized in Lin et al. (2023), this baseline addresses the diversity limitations of centroid-based methods. It partitions the generated pool into $N$ clusters (where $N$ is the budget size) using K-means and selects the sample closest to each cluster center. This ensures uniform coverage of the synthetic manifold.

### C.2.2. DETAILED IMPLEMENTATION OF METRICS

To provide a comprehensive assessment of the generated data quality, we employ a combination of metrics measuring both distributional fidelity and sample diversity.

**Image Generation Metrics.** We utilize the Inception-V3 feature space to compute the following metrics, ensuring a standardized comparison with prior works.

- **Fréchet Inception Distance (FID).** This metric computes the Fréchet distance between two multivariate Gaussian approximations fitted to the Inception-V3 feature embeddings of real and generated images. Concretely, let $(\mu_r, \Sigma_r)$ and $(\mu_g, \Sigma_g)$ be the empirical means and covariance matrices of the real feature distributions $\mathcal{D}_r$ and generated feature distributions $\mathcal{D}_g$, respectively. The FID is defined as:

$$\text{FID}(\mathcal{D}_r, \mathcal{D}_g) = \|\mu_r - \mu_g\|_2^2 + \text{Tr}\left(\Sigma_r + \Sigma_g - 2(\Sigma_r \Sigma_g)^{1/2}\right) \tag{196}$$

  A lower FID indicates closer alignment of the generated distribution to the real distribution, reflecting higher visual quality and better mode coverage.

- **Precision and Recall.** To disentangle the evaluation of quality and diversity, we adopt the improved precision and recall metrics (Kynkäänniemi et al., 2019).
  - **Implementation Details:** We compute these metrics in the Inception-V3 feature space using $k$-nearest neighbor ($k$-NN) manifold estimation with $k = 5$. The estimation is performed using the generated sample pool (e.g., 10,000 samples per iteration) against the full training set to ensure statistical consistency.
  - **Precision (Fidelity):** This metric measures the fraction of generated images that lie within the estimated support (manifold) of the real data distribution. It quantifies how "realistic" the generated samples are.
  - **Recall (Diversity):** This metric measures the fraction of real images that lie within the estimated support of the generated data distribution. A significant drop in Recall is a strong indicator of *mode collapse*, implying the generator has failed to capture the full variation of the training data.

### C.2.3. DETAILED IMPLEMENTATION OF CONFIGURATIONS

We detail the model architectures, datasets, and training protocols used for both image and text generation tasks. All experiments were conducted following standard configurations established in prior literature to ensure fair comparability.

**Image Generation Framework.** For image synthesis, we adopt the standard Denoising Diffusion Probabilistic Models (DDPM) framework (Ho et al., 2020). Following the configuration in Shi et al. (2025), we train the model using a standard U-Net architecture with attention mechanisms at multiple resolution levels.

- **Datasets and Preprocessing.** In our experiments, we employ three benchmark datasets representing different levels of complexity. For the initial training phase (Generation $t = 0$), we utilize the 50,000 training samples of each training dataset to establish a strong generator.

  - **CIFAR-10:** This dataset comprises 60,000 diverse natural images spanning 10 distinct semantic classes (e.g., vehicles and animals). It is officially partitioned into a training set of 50,000 images and a test set of 10,000 images. We utilize the complete training set with a native resolution of $32 \times 32$. It serves as a fundamental benchmark to evaluate the model's capability in capturing global semantic structures in a low-resolution setting.
  - **CelebA:** This large-scale face-attribute dataset contains 202,599 images. We follow the standard protocol for the official split (162,770 training, 19,867 validation, and 19,962 testing). To ensure a consistent data scale across experiments, we randomly sample a subset of 50,000 images from the training split. All images are preprocessed to $32 \times 32$ via center-cropping. We map the hair color attributes into five distinct categories: 0: Black Hair, 1: Blond Hair, 2: Brown Hair, 3: Gray Hair, and 4: Other (including Bald or unknown).
  - **STL-10:** To evaluate performance on more diverse object structures, we utilize the STL-10 dataset, which consists of 5,000 labeled training images, 8,000 test images, and 100,000 unlabeled images. We randomly sample 50,000 images from the combined pool of labeled and unlabeled data to construct the training set. While the original resolution is $96 \times 96$, we resize the images to $32 \times 32$ to align with our model configuration. This dataset challenges the model to capture fine-grained details within a reduced resolution.

- **Training and Sampling Configuration:** We adopt the Denoising Diffusion Probabilistic Models (DDPM) (Ho et al., 2020) framework utilizing a standard U-Net (Ronneberger et al., 2015) architecture. The diffusion process is trained over $T = 1000$ timesteps employing a linear variance schedule. For efficient inference, we use the Denoising Diffusion Implicit Models (DDIM) (Song et al., 2021) algorithm with 50 sampling steps. In each generation cycle, we synthesize a candidate pool of $N = 4n$ samples (where $n = 50,000$) and apply our selection mechanism to filter the top-$n$ samples for the subsequent training iteration.

In all experimental settings, we enforce a consistent selection protocol to ensure a fair comparison. For a target training budget of $n$ samples per category, we initially generate a larger candidate pool consisting of $N = 4n$ synthetic images. We then apply our selection mechanism (as described in the methodology) to filter this candidate pool, retaining exactly $n$ representative samples while discarding the remaining $3n$ instances. Based on these selected samples, we evaluate three distinct strategies for constructing the final training set:

- **Replace:** This strategy prioritizes current data distributions. We discard all historical data and construct the training set using exclusively the $n$ newly selected samples from the current generation step. Consequently, the training set size is fixed at $n$.

- **Accumulate:** This strategy maximizes data volume. We append the $n$ newly selected samples to the existing historical dataset. As a result, the training set size increases by $n$ at each update step, utilizing the full history of selected synthetic data.

- **Accumulate-Subsample:** This strategy balances diversity and computational cost. We first combine the $n$ newly selected samples with the accumulated historical data. From this unified pool, we randomly draw a fixed-size subset of $n$ samples. This ensures the training set size remains constant at $n$ while maintaining a diverse mixture of current and historical distributions.

## C.3. Hardware

All models were trained and evaluated on a dedicated server running Ubuntu 20.04.2 LTS. The system is equipped with dual Intel® Xeon® Gold 6442Y CPUs (48 physical cores, 96 threads @ 2.60 GHz), approximately 503 GiB of system memory, and a GPU cluster consisting of 8 NVIDIA L40 GPUs (48 GB VRAM each). Detailed code and generation results for each round are available at ○ GitHub.

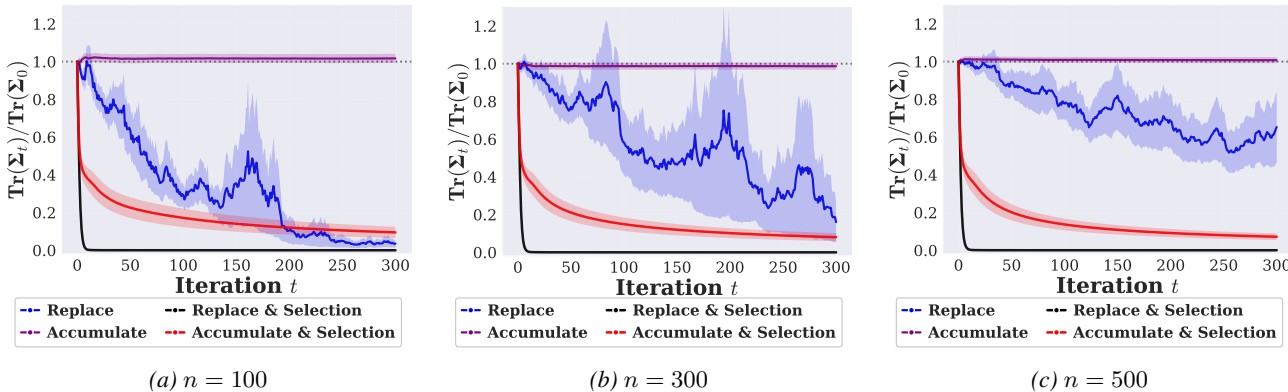

*(a)* $n = 100$        *(b)* $n = 300$        *(c)* $n = 500$

*Figure 6.* **Impact of Sample Size on Variance Collapse.** We visualize the evolution of the variance ratio $\mathrm{Tr}(\boldsymbol{\Sigma}_t)/\mathrm{Tr}(\boldsymbol{\Sigma}_0)$ over generations under varying generated sample sizes ($n$) in each iteration. While increasing the sample size reduces the collapse speed for the naive `Replace` baseline (Blue), the Replace Paradigm with biased selection method (Black) consistently exhibits rapid variance decay governed by the power-law dynamics described in Theorem 2, showing that selection bias dominates finite-sample effects.

## C.4. Multivariate Gaussian Modeling and Non-Gaussian Empirical Evidence

**Multivariate Gaussian Modeling**     While Section 3 focuses on the representative regime of $n = 300$ to illustrate the core collapse dynamics, we provide here a comprehensive specification of the experimental environment and extend our analysis to varying sample sizes ($n \in \{100, 500\}$) to demonstrate the universality of the observed phenomena.

**Experimental Settings and Hyperparameters.**     To empirically validate our theoretical findings, we perform controlled simulations on a synthetic multivariate Gaussian benchmark. The data dimension is fixed at $d = 10$, and the iterative training is tracked over $T = 300$ generations. We initialize the ground truth distribution as $\mathcal{N}(\boldsymbol{\mu}^*, \boldsymbol{\Sigma}^*)$ with randomly sampled parameters. For the selection mechanism, we design a biased utility function $U(\mathbf{x}) = -\|\mathbf{x} - \mathbf{u}^*\|^2$ that favors a local mode $\mathbf{u}^*$ distinct from the true mean $\boldsymbol{\mu}^*$. We enforce a strict selection pressure by setting the retention ratio to $\alpha = 0.05$, meaning only the top 5% of generated samples with the highest utility scores are used for training the next generation. We vary the generated sample size $n \in \{100, 300, 500\}$ in each iteration to study finite-sample effects.

**Results: The Dominance of Selection Bias.**     Figure 6 illustrates the evolution of the variance ratio $\mathrm{Tr}(\boldsymbol{\Sigma}_t)/\mathrm{Tr}(\boldsymbol{\Sigma}_0)$ across different sample sizes. The comparative results highlight two fundamentally different collapse regimes:

- **Buffering Effect against Selection:** When selection is applied to accumulated data (**Accumulate & Selection**), the variance decay is markedly slower than in the **Replace & Selection** case. Historical data act as a "memory anchor," diluting the concentration effect of biased synthetic samples. However, the downward trend persists, indicating that while accumulation delays collapse, it cannot fully neutralize the long-term distributional drift induced by a fixed utility function.

- **Mitigation in Naive Replacement ($n$-dependent):** In the absence of selection (the **Replace** baseline), the collapse rate is highly sensitive to the sample size. As observed in Figure 6a ($n = 100$) versus Figure 6c ($n = 500$), increasing the sample size significantly delays the variance decay. This aligns with standard statistical theory: larger $n$ reduces the variance of the sample covariance estimator (scaling roughly with $O(1/n)$), thereby preserving the distributional width for longer periods.

- **Invariance in Selection Dynamics ($n$-independent):** Conversely, **Replace & Selection** and **Accumulate & Selection** demonstrate a striking insensitivity to sample size. Whether $n = 100$ or $n = 500$, the variance collapses rapidly and follows a nearly identical trajectory. This observation is crucial: it indicates that the selection bias acts as the dominant force, overshadowing the benefits of increased data volume. The selection mechanism effectively truncates the distribution's tails at each step, forcing a contraction rate that adheres to the power-law dynamics predicted in Theorem 2. Consequently, simply scaling up the synthetic data generation (increasing $n$) fails to counteract the structural collapse induced by biased selection.

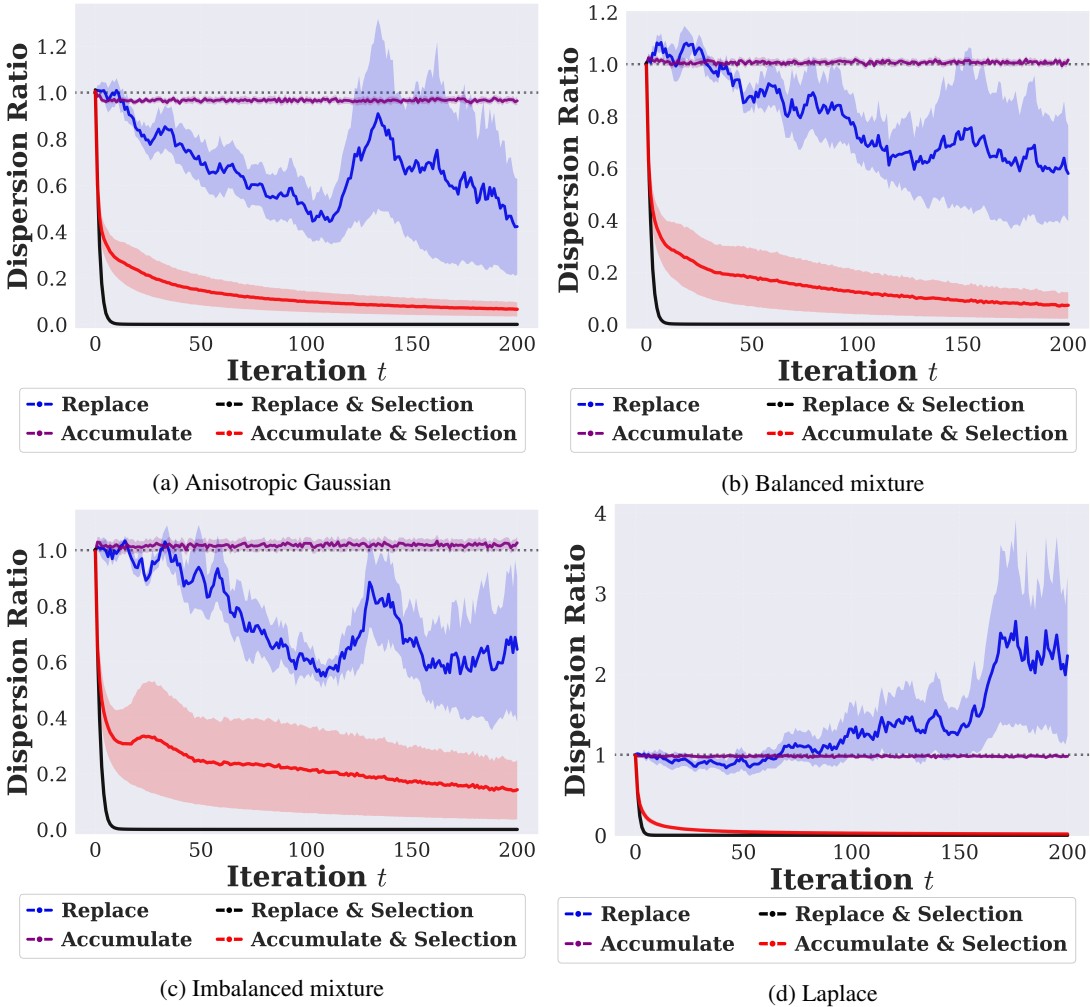

*Figure 7.* Recursive selection dynamics beyond the isotropic Gaussian model, reported for $n = 300$. Across structured, multimodal, imbalanced, and heavy-tailed distributions, local-reference selection sharply reduces normalized dispersion.

**Non-Gaussian recursive selection dynamics.** The theoretical results in Section 3 are derived under a Gaussian selection model, which provides a tractable setting for analyzing how local-reference filtering affects the mean, covariance, and long-term diversity of recursively generated data. This assumption is useful for obtaining explicit collapse dynamics and the associated power-law decay, but it also raises a natural question: whether the same qualitative mechanism persists when the data distribution departs from the Gaussian structure used in the analysis. We therefore include additional empirical checks that explore the selection process along several axes not covered by the main theory, including anisotropy, multimodality, component imbalance, heavy-tailed variation, semantic verifier bias, and weaker generative initialization.

These experiments should be interpreted as empirical robustness evidence rather than as formal extensions of Theorems 1 and 2. The goal is not to claim that the Gaussian power-law rate holds unchanged in all these settings. Instead, we examine whether the mechanism identified in the main text remains observable: when the verifier relies on a restricted local reference, recursive selection tends to preserve samples aligned with that reference while suppressing regions that are underrepresented, distant, or semantically outside the verifier's domain. Across the following settings, the results consistently support this qualitative interpretation.

We first consider synthetic distributions that isolate different forms of distributional structure beyond the isotropic Gaussian case. The anisotropic Gaussian control tests whether collapse persists when the distribution has direction-dependent variance. The balanced mixture setting introduces multimodality, allowing us to examine whether selection preserves multiple separated components or concentrates mass around a subset of them. The imbalanced mixture further introduces

*Table 3.* Final dispersion ratios at iteration $t = 200$. Lower values indicate stronger diversity collapse. The omitted REPLACE+SELECT column equals 0.0000 for every distribution family and sample size, and is therefore reported in the caption rather than as a degenerate all-zero column.

| Distribution | $n$ | Accumulate | Accumulate + Select | Replace | Replace + Select |
|---|---|---|---|---|---|
| Anisotropic Gaussian | 100 | 0.9894 | 0.0829 | 0.0556 | 0.0000 |
| | 300 | 0.9646 | 0.0652 | 0.4221 | 0.0000 |
| | 500 | 0.9736 | 0.0734 | 0.6692 | 0.0000 |
| Balanced mixture | 100 | 0.9455 | 0.0923 | 0.0258 | 0.0000 |
| | 300 | 1.0162 | 0.0729 | 0.5798 | 0.0000 |
| | 500 | 1.0034 | 0.0472 | 0.5604 | 0.0000 |
| Imbalanced mixture | 100 | 0.9699 | 0.0690 | 0.0790 | 0.0000 |
| | 300 | 1.0263 | 0.1422 | 0.6449 | 0.0000 |
| | 500 | 1.0010 | 0.0736 | 0.5615 | 0.0000 |
| Laplace | 100 | 1.0439 | 0.0169 | 0.1336 | 0.0000 |
| | 300 | 0.9803 | 0.0161 | 2.2282 | 0.0000 |
| | 500 | 0.9990 | 0.0149 | 1.0510 | 0.0000 |

minority modes, which are particularly relevant to our claim that biased selection can remove weakly represented regions of the distribution. Finally, the Laplace family introduces heavier tails, testing whether local-reference selection suppresses tail scale rather than only collapsing Gaussian-like variance. For each distribution family, we track the normalized dispersion $\text{DispRatio}(P_t) = \frac{\text{Disp}(P_t)}{\text{Disp}(P_0)}$, where smaller values indicate stronger loss of distributional diversity relative to the initial distribution. This metric provides a common scale across distribution families and sample sizes, allowing us to compare how recursive accumulation, replacement, and selection affect the preservation of distributional spread. In addition to this aggregate dispersion measure, we report family-specific diagnostics for the imbalanced mixture and Laplace settings, since these two cases expose more structured forms of collapse: erosion of minority components and contraction of heavy-tailed variation.

The trajectories in Figure 7 reveal several consistent patterns. First, accumulation without selection remains close to the initial dispersion across all four distribution families, indicating that recursive accumulation alone does not necessarily induce immediate diversity loss in these controlled settings. Second, local-reference selection introduces a strong dissipative effect. In the anisotropic Gaussian, balanced mixture, and imbalanced mixture settings, ACCUMULATE+SELECT decreases monotonically or near-monotonically from unit dispersion to a small residual level, whereas REPLACE+SELECT collapses almost immediately to numerical zero. This sharp contrast indicates that the diversity loss is not caused solely by recursive resampling, but by the interaction between recursion and selective filtering toward a restricted reference.

The behavior of the REPLACE baseline further clarifies the role of selection. Without selection, replacement dynamics can be noisy and distribution-dependent: in the mixture settings, dispersion decreases gradually but remains substantially above zero, while in the Laplace setting it can even expand beyond the initial dispersion due to heavy-tailed fluctuations. Once selection is applied, however, this instability is converted into rapid collapse. In particular, REPLACE+SELECT reaches a final dispersion ratio of 0.0000 for every tested distribution family and sample size in Table 3. This all-zero outcome is not a missing-data artifact; it reflects a degenerate regime in which selection repeatedly concentrates the generated distribution around the local reference and eliminates measurable spread.

The complete dispersion statistics in Table 3 also show that the phenomenon is stable across sample sizes. For ACCUMULATE, the final dispersion ratio remains close to one in most cases, ranging from 0.9455 to 1.0439 across the reported non-Gaussian families and sample sizes. In contrast, ACCUMULATE+SELECT consistently reduces dispersion by an order of magnitude or more, with final ratios below 0.15 in all reported cases and below 0.02 for the Laplace family. This pattern is important because it separates the effect of biased selection from finite-sample randomness: increasing the sample size from $n = 100$ to $n = 500$ does not remove the collapse trend. Instead, the same qualitative behavior persists across structured, multimodal, imbalanced, and heavy-tailed regimes.

Beyond aggregate dispersion, the imbalanced-mixture and Laplace settings provide targeted diagnostics for the type of

*Table 4.* Minority-mode diagnostics for the imbalanced mixture at iteration $t = 200$. Lower smallest-component weight and lower entropy indicate stronger erosion of low-probability modes.

| $n$ | Setting | Smallest Comp. Weight | Weight Entropy |
|---|---|---|---|
| 100 | Accumulate | 0.2794 | 0.8496 |
| | Accumulate + Select | 0.1648 | 0.5688 |
| | Replace | 0.0000 | 0.0000 |
| | Replace + Select | 0.0000 | 0.0000 |
| 300 | Accumulate | 0.2123 | 0.7436 |
| | Accumulate + Select | 0.1692 | 0.5717 |
| | Replace | 0.1087 | 0.2774 |
| | Replace + Select | 0.0000 | 0.0000 |
| 500 | Accumulate | 0.2434 | 0.7997 |
| | Accumulate + Select | 0.1591 | 0.5579 |
| | Replace | 0.2052 | 0.5403 |
| | Replace + Select | 0.0119 | 0.0653 |

*Table 5.* Tail-scale diagnostics for the Laplace family at iteration $t = 200$. Lower mean absolute scale indicates stronger contraction of heavy-tailed variation.

| $n$ | Setting | Mean Absolute Scale |
|---|---|---|
| 100 | Accumulate | 2.0087 |
| | Accumulate + Select | 0.2463 |
| | Replace | 0.3788 |
| | Replace + Select | 0.0007 |
| 300 | Accumulate | 1.9573 |
| | Accumulate + Select | 0.2413 |
| | Replace | 1.8039 |
| | Replace + Select | 0.0006 |
| 500 | Accumulate | 1.9828 |
| | Accumulate + Select | 0.2335 |
| | Replace | 1.6169 |
| | Replace + Select | 0.0006 |

diversity being lost. The imbalanced mixture tests whether selection erodes minority components rather than merely reducing total variance. As shown in Table 4, selection lowers both the smallest component weight and the component-weight entropy under accumulation, indicating that low-probability modes become less represented even when the overall recursive process remains stable. Under replacement, the effect is stronger: the minority component disappears completely for $n = 100$ and $n = 300$ under REPLACE+SELECT, and is reduced to a near-zero weight at $n = 500$. This supports the interpretation that local-reference selection preferentially suppresses modes that are weakly represented or distant from the verifier's reference.

The Laplace diagnostics in Table 5 show a complementary form of collapse. Since the Laplace family has heavier tails, mean absolute scale measures whether tail magnitude is retained. Accumulation preserves this scale near its initial level, while ACCUMULATE+SELECT reduces it from approximately $2.0$ to about $0.24$ across all sample sizes. REPLACE+SELECT further contracts the scale to nearly zero. Thus, biased filtering does not merely collapse multimodal structure; it can also remove heavy-tailed variation. Together, these diagnostics indicate that the dispersion collapse observed in Figure 7 corresponds to structured loss of distributional coverage, including minority-mode erosion and tail-scale contraction.

## C.5. Algorithms and Empirical Results

In this section, we present the pseudocode for our algorithm, followed by additional experiments, including **(i)** an analysis of the computational time of Algorithms 1 and 2 and **(ii)** the impact of varying Dirichlet settings.

---

**Algorithm 1** Scheme I: Collaborative Geodesic Interpolation & Greedy Selection

---

1: **Input:** Synthetic data $\mathcal{P}$, real data shards $\{\mathcal{Q}_k\}_{k=1}^K$, rounds $R$, budget $n$.
2: **Output:** Selected subset $\mathcal{I}$.
3: **// Phase 1: Geodesic Proxy Estimation** {*See Figure 2*}
4: Initialize proxies $\xi_k^{(0)}$ for all $k$
5: **for** $r = 0$ **to** $R-1$ **do**
6:     **for** $k = 1$ **to** $K$ **in parallel do**
7:         Interpolation: $\xi_{\mathcal{Q}_k}^{(r)} \leftarrow \arg\min_\xi \mathbb{W}_p(\mathcal{Q}_k, \xi) + \mathbb{W}_p(\xi, \xi_k^{(r)})$ {*See Property 2*}
8:         Interpolation: $\xi_{\mathcal{P}}^{(r)} \leftarrow \arg\min_\xi \mathbb{W}_p(\mathcal{P}, \xi) + \mathbb{W}_p(\xi, \xi_k^{(r)})$
9:         Update proxy: $\xi_k^{(r+1)} \leftarrow \arg\min_\xi \mathbb{W}_p\left(\xi_{\mathcal{P}}^{(r)}, \xi\right) + \mathbb{W}_p(\xi, \xi_{\mathcal{Q}_k}^{(r)})$
10:     **end for**
11: **end for**
12: Set final proxies $\xi_k^* \leftarrow \xi_k^{(R)}$ {*Converges via Theorem 4*}
13: **// Phase 2: Greedy Selection**
14: **for** $k = 1$ **to** $K$ **in parallel do**
15:     Compute scores: $\mathcal{S}_k(x_i) \leftarrow f^*(x_i) - \frac{1}{N-1}\sum_{j\neq i}^N f^*(x_j)$ {*Equation (18)*}
16:     Min–max normalization: $\tilde{\mathcal{S}}_k(x_i) = \frac{\mathcal{S}_k(x_i) - \min_j \mathcal{S}_k(x_j)}{\max_j \mathcal{S}_k(x_j) - \min_j \mathcal{S}_k(x_j)}$
17: **end for**
18: Initialize: $\mathcal{I} \leftarrow \emptyset$
19: Greedy selection: $\underset{\mathcal{I} \subseteq \{1,\dots,N\}}{\text{maximize}} \quad \sum_{k=1}^K g\left(\sum_{i\in\mathcal{I}}(1 - \tilde{\mathcal{S}}_k(x_i))\right) \quad$ s.t. $|\mathcal{I}| \leq n$.
20: **Return** $\mathcal{I}$

---

**Algorithm 2** Scheme II: Collaborative Barycenter Estimation

---

1: **Input:** Synthetic data $\mathcal{P}$, real data shards $\{\mathcal{Q}_k\}_{k=1}^K$, rounds $R$, selection ratio $\alpha$.
2: **Output:** Selected indices $\mathcal{I}$.
3: **// Phase 1: Offline Barycenter Estimation**
4: **Server:** Initializes global barycenter $\xi^{(0)}$ {*See Figure 3*}
5: **for** $r = 0$ **to** $R-1$ **do**
6:     **Server:** Broadcasts $\xi^{(r)}$ to all parties
7:     **for** $k = 1$ **to** $K$ **in parallel do**
8:         **Client $k$:** Computes local interpolation
9:         $\xi_k^{(r)} \leftarrow \arg\min_\xi \mathbb{W}_p(\mathcal{Q}_k, \xi) + \mathbb{W}_p\left(\xi, \xi^{(r)}\right)$
10:         **Client $k$:** Sends $\xi_k^{(r)}$ to Server
11:     **end for**
12:     **Server:** Updates barycenter {*See Property 1*}
13:     $\xi^{(r+1)} \leftarrow \arg\min_\xi \sum_{k=1}^K \mathbb{W}_p\left(\xi, \xi_k^{(r)}\right)$
14: **end for**
15: $\xi^* \leftarrow \xi^{(R)}$ {*Converges via Theorem 5*}
16: **// Phase 2: Calibrated Gradient Filtering**
17: **for each** $x_i \in \mathcal{P}$ **do**
18:     Compute Score: $\mathcal{S}(x_i) \leftarrow f^*(x_i) - \frac{1}{N-1}\sum_{j\neq i}^N f^*(x_j)$ {*Equation (18)*}
19: **end for**
20: $\mathcal{I} \leftarrow \arg\text{topk}(\{-\mathcal{S}(x_i)\}, k = \alpha|\mathcal{P}|)$
21: **Return** $\mathcal{I}$

---

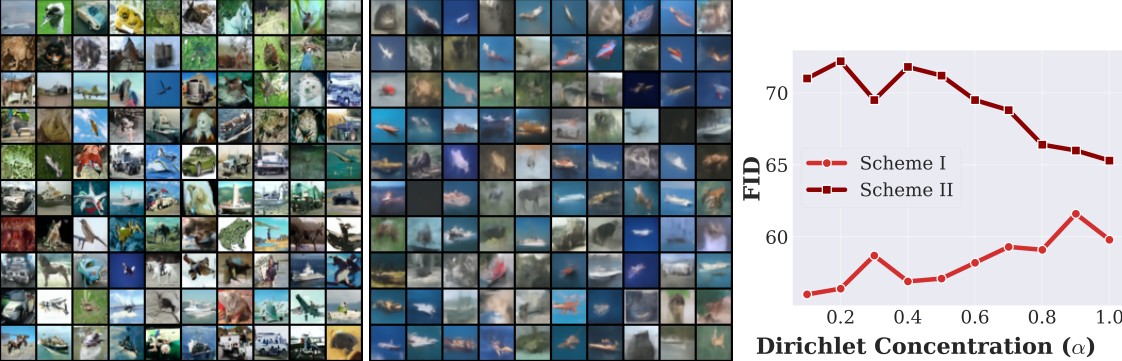

*Figure 8.* **Evolution of the Training Dataset under the Replace Paradigm. (Left)** Snapshot at Iteration 1. **(Middle)** Snapshot at Iteration 10. **(Right) Impact of Data Heterogeneity.** We report FID scores under varying Dirichlet concentration parameters ($\alpha$).

**Data Heterogeneity.**    As illustrated in Figure 8, `Scheme I` exhibits superior robustness in highly non-IID regimes ($\alpha < 0.5$) by effectively leveraging heterogeneity as a diversity source. In contrast, `Scheme II` shows significant performance gains as the data distribution becomes more homogeneous ($\alpha \to 1.0$) with lower computational cost.

- **Heterogeneity Transforms into Diversity.** Scheme I (Circles) consistently outperforms Scheme II (Squares) in terms of generation quality (lower FID), particularly in highly non-IID settings (low Dirichlet concentration $\alpha < 0.5$). As illustrated in the plot, Scheme I maintains a stable low FID ($\approx 56$) even under extreme heterogeneity. This validates the theoretical premise in Subsection 4.3: by computing interpolations along geodesics to multiple distinct targets rather than a single aggregated mean, Scheme I preserves the unique distributional modes of disjoint data shards.

- **Efficiency and Proxy Validity in Homogeneous Regimes.** As the data distribution becomes more homogeneous (Dirichlet $\alpha \to 1.0$), the performance gap between the two methods narrows significantly, with Scheme II showing a steep improvement in FID (dropping from $\approx 72$ to $\approx 65$). This indicates that when the divergence between local distributions is low, the Wasserstein barycenter (Scheme II) becomes a statistically valid proxy for the ground truth.

**Topic-local verification in recursive LLM training.**    We further examine whether local-reference selection exhibits a similar failure mode when the verifier bias is semantic rather than geometric. Following the verification-based recursive training setup of Feng et al. (2025), we fine-tune Llama-2-7B (Touvron et al., 2023) on the English subset of XLSum (Hasan et al., 2021). To instantiate a siloed semantic-reference setting, we partition XLSum by topic so that each entity observes only one topical subset as its local data. In the experiment shown in Figure 9, the verifier is constructed from the technology subset, and generated samples are filtered according to their ROUGE-based selection with this technology-local reference (Lin, 2004). Recursive training is then evaluated on held-out non-technology topics.

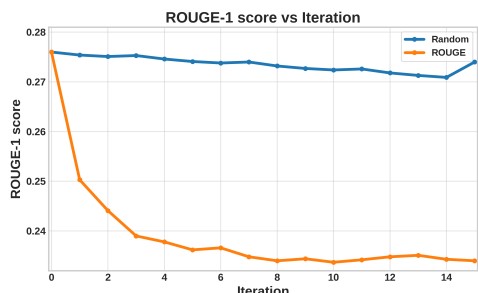

*Figure 9.* Held-out topic generalization under recursive training with a technology-local verifier. ROUGE-based local selection drops early and stabilizes below random selection.

This experiment is not intended as a comprehensive LLM benchmark. Its purpose is narrower: to test whether a verifier that enforces local semantic alignment can simultaneously degrade coverage outside its reference domain. In this sense, the setup reflects a low-resource verification condition: the verifier has access only to a restricted topical slice, while held-out topics behave as underrepresented regions of the target distribution. This framing is consistent with recent concerns that model collapse can be especially consequential when tail information is scarce or weakly represented (Jarvis et al., 2026). As shown in Figure 9, ROUGE-based local selection yields weaker held-out topic generalization than random selection. The degradation occurs early in recursive training and persists across later iterations, suggesting that filtering against a narrow topical reference can suppress samples useful for non-local semantic regions. This observation supports the broader mechanism analyzed in the main text: when the verifier's reference is semantically narrow, sample selection can become a source of coverage loss rather than a safeguard against collapse.

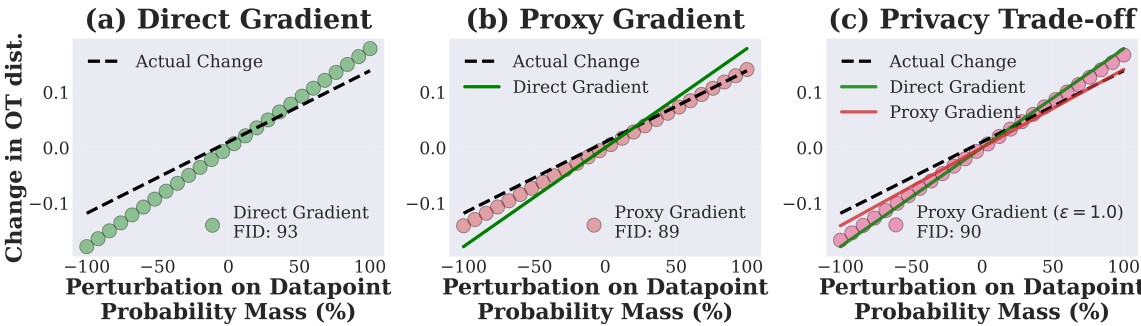

*Figure 10.* **Predicting the change in OT distance when increasing or reducing the probability mass of a point on the CIFAR-10 dataset. (a) Direct Gradient:** The predicted change in Wasserstein distance (Green points, ●) computed via standard dual potentials accurately matches the **actual numerical change** (Black dashed line, - - -), confirming the theoretical validity of using dual potentials as influence scores. **(b) Proxy Gradient:** The gradient estimated using the Proxy target (geodesic interpolation at $t = 0.5$) (Light red points, ●) aligns perfectly with the Direct Gradient, justifying the use of the proxy distribution for efficient selection. **(c) Differential Privacy:** Under a privacy budget of $\epsilon = 1.0$, the Noisy Proxy target (Red points, ●) remains highly correlated with the clean gradients despite the added noise, demonstrating the robustness of our method for private data selection.

## C.6. Differentially Private Optimal Transport

To provide formal privacy guarantees beyond the implicit protection of barycentric compression, we integrate the Differentially Private Optimal Transport (DPOT) framework (Lê Tien et al., 2019). This method leverages the Johnson–Lindenstrauss transform (Kenthapadi et al., 2013) to sanitize the pairwise distance matrix required for optimal transport. Let $\mathbf{X}_{\mathcal{P}} \in \mathbb{R}^{n \times d}$ and $\mathbf{X}_{\mathcal{Q}} \in \mathbb{R}^{m \times d}$ be the data matrices associated with probability measures $\mathcal{P}$ and $\mathcal{Q}$. The party holding $\mathcal{P}$ generates a random Gaussian projection matrix $\mathbf{M} \in \mathbb{R}^{d \times l}$ (entries i.i.d. $\sim \mathcal{N}(0, 1/l)$) and adds a noise matrix $\mathbf{\Delta}$ (entries $\sim \mathcal{N}(0, \sigma)$) to release a sanitized view $\tilde{\mathbf{X}}_{\mathcal{P}} = \mathbf{X}_{\mathcal{P}}\mathbf{M} + \mathbf{\Delta}$. The receiving party holding $\mathcal{Q}$ then projects their data as $\tilde{\mathbf{X}}_{\mathcal{Q}} = \mathbf{X}_{\mathcal{Q}}\mathbf{M}$ and approximates the cost matrix as $\tilde{\mathbf{C}}_{ij} = \|(\tilde{\mathbf{X}}_{\mathcal{P}})_i - (\tilde{\mathbf{X}}_{\mathcal{Q}})_j\|^2 - l\sigma^2$, where the term $l\sigma^2$ corrects the bias induced by the noise. This mechanism satisfies $(\epsilon, \delta)$-differential privacy provided that $\sigma \geq w\sqrt{2(\ln(1/2\delta) + \epsilon)}/\epsilon$, where $w$ is the sensitivity of the projection.

To validate the theoretical underpinnings of our data selection method, we analyze the sensitivity of the optimal transport (OT) distance to perturbations in the probability mass of individual data points. We conduct this experiment using subsets of the **CIFAR-10** dataset, with $N = 5,000$ samples randomly selected for both the source and target distributions.

Figure 10 visualizes the relationship between the *predicted* change in OT distance (derived from our gradient-based scoring) and the *actual* numerical change observed when re-solving the exact OT problem. The perturbation involves varying the probability mass of a single target point from $-100\%$ to $+100\%$. For the privacy-preserving scenario, we employ a privacy budget of $\epsilon = 1.0$, an interpolation factor of $t = 0.5$, and report the average predictions over 30 independent trials to account for the stochasticity of the DP mechanism.

- **(a) Validation of the Direct Gradient:** Figure 10(a) compares the gradient-based prediction using standard dual potentials (Green points) against the ground-truth change in Wasserstein distance (Black dashed line). This empirically verifies that the scoring mechanism accurately reflects the contribution of each data to the overall distributional distance.

- **(b) Effectiveness of the Proxy Gradient:** In Figure 10(b), we evaluate the accuracy of gradients estimated using a *Proxy Gradient* constructed via geodesic interpolation at $t = 0.5$ (Light red points). The proxy-based gradients exhibit a near-perfect correlation with the direct gradient (Green line). This result justifies the use of proxies for data selection, demonstrating that full transport to the final target is not strictly necessary to identify high-value samples.

- **(c) Robustness to Differential Privacy:** Figure 10(c) demonstrates the method's robustness under a strict privacy budget ($\epsilon = 1.0$). Despite the injection of noise into the proxy target construction (Red points), the estimated gradients maintain a strong positive correlation with the clean direct gradient (Green line). While the individual points exhibit variance due to the DP noise, the overall trend remains linear and directionally correct, ensuring that the selection mechanism remains effective even when the target distribution is private. We empirically show that there is a negligible difference in FID results between data selected via the noisy proxy and data selected via the direct method.

