# OpenReview forum: "When Sample Selection Bias Precipitates Model Collapse"
_ICML.cc/2026/Conference — ICML 2026 regular_

### Official Review · Reviewer_pAu7 · 2026-03-04

**Soundness:** 3
**Presentation:** 3
**Significance:** 3
**Originality:** 2
**Overall Recommendation:** 4
**Confidence:** 2

**Summary:**

This paper studies the model collapse problem due to recursive model retraining on dataset that contains model-generated synthetic data. Specifically, this paper focuses on characterizing the impact of sample selection on model collapse, which is formulated as diminishing variation in data. Based on the classical multivariate Gaussian analysis framework, the authors analyze the failure of accumulating paradigm (of merging synthetic data), which is enlightening and substantially different with the case without selection rule, even though the selection is often implemented for better data quality. Next, the authors propose Wasserstein-Gradient-Based Selection and demonstrates its theoretical properties and benefits. At last, experiments on classical datasets are conducted to validate the effectiveness of proposed methodology.

**Compliance With Llm Reviewing Policy:**

Affirmed.

**Final Justification:**

I maintain my overall recommendation of 4 (weak accept). The paper addresses an important problem, and I find the main technical results sound within the Gaussian/truncated-Gaussian setting. Its strengths are clear presentation, solid analysis in the stated regime, and a practically relevant Wasserstein-based selection method with empirical support.

The rebuttal addressed most of my concerns by clarifying the scope of the theory and adding discussion on stability and scalability. However, the question of how far the conclusions extend beyond the Gaussian setting remains only partially resolved, so my overall evaluation is unchanged.

Overall, I view this as a solid and useful contribution, though its broader impact is somewhat limited by the current scope of the analysis.

**Key Questions For Authors:**

1. Which parts of the collapse claim (variance-to-zero, power-law rate, dependence on $\alpha$/bias strength) remain true if the data distribution is non-Gaussian (e.g., mixture of Gaussians/heavy-tailed/multimodal)?
2. When is the Wasserstein-gradient score reliable under finite pruning?
3. Would the dual non-uniqueness influence the method? (see the weakness part mentioned above)

**Limitations:**

Yes.

**Strengths And Weaknesses:**

Strengths:
1. The problem is important and practically relevant.
2. The theoretical results are solid and relevant.
3. The paper is well-written and easy to follow.

Weaknesses:
1. **Lack of discussion beyond Gaussian case**. The collapse and power-law rate are derived in a very special regime (multivariate normal + top-$\alpha$ truncation). It remains unclear which qualitative conclusions are robust to non-Gaussian, multimodal, heavy-tailed, or highly anisotropic feature distributions that are more realistic for deep generative models/embeddings

2. **Non-uniqueness and instability of OT dual potentials are under-addressed**. In discrete OT, optimal plans and dual potentials can be non-unique; the "zero-sum convention" fixes a gauge but does not resolve multi-solution ambiguity. If $f^*$ is not unique, sample scores $S\left(x_i\right)$ may be unstable.

3. **Computational and practical scalability is insufficiently discussed.**

---

> ### Author Rebuttal · Authors · 2026-03-30
>
> We sincerely thank the reviewer for the thoughtful and supportive evaluation, and for the time and effort devoted to reviewing our paper. We appreciate the reviewer’s positive assessment of the problem, theory, and presentation. Below we address the concerns on generality beyond the Gaussian regime, OT dual non-uniqueness, and scalability. For clarity, W denotes Weakness and Q denotes Question.
> > **W1.** Lack of discussion beyond Gaussian case.
> >
> > **Q1.** Which parts of the collapse claim remain true if the data distribution is non-Gaussian?
>
> We agree that our current collapse analysis is developed in a controlled multivariate Gaussian / truncated-Gaussian regime, and we will clarify this scope in the revision. Our goal is to characterize the failure boundary of existing model-collapse theory under sample selection, so we intentionally build on the same framework as prior work to make the comparison controlled and precise.
>
> Accordingly, our claims should be interpreted at two levels. Quantitatively, the variance-to-zero result and the explicit power-law rate in Thms. 1&2 are proved in the Gaussian setting, and we do not claim that the same closed-form exponent must hold for arbitrary non-Gaussian distributions. Qualitatively, however, the broader mechanism is expected to persist: repeated locally biased filtering removes underrepresented modes/tails and increases discrepancy from the global manifold. This is also consistent with Theorem 3, which is stated more generally through a Wasserstein-based risk bound.
>
> To further address this concern, we additionally conducted non-Gaussian simulations, and the qualitative trend remains consistent with our theory. Due to space limits, we place the corresponding visualizations in the **[anonymous repository](https://anonymous.4open.science/r/When-Sample-Selection-Bias-Precipitates-Model-Collapse-D8D4/README.md)** and update the accompanying code for reproducibility. We will incorporate these results into  revision.
> > **W2.** Non-uniqueness and instability of OT dual potentials are under-addressed.
> >
> > **Q2.** When is the Wasserstein-gradient score reliable under finite pruning?
> >
> > **Q3.** Would the dual non-uniqueness influence the method?
>
> We thank the reviewer for this important point. We agree that the zero-sum convention only fixes the additive gauge, and does not by itself resolve dual non-uniqueness. In our setting, however, the key practical question is whether such multi-solution ambiguity leads to materially different sample rankings or pruning decisions.
>
> Empirically, rankings remain stable across perturbations. To probe the stability of the ranking signal in practice, we inject different levels of differential privacy perturbation into the OT computation and measure the Spearman correlation coefficient between the perturbed and unperturbed sample rankings.  As shown below, the induced rankings remain reasonably stable across datasets. This signal is most reliable for moderate pruning budgets, or in iterative recompute-and-prune settings where the perturbation remains local. We will clarify this distinction in the revision.
>
> Table A (Spearman)|$\epsilon=1$ | $\epsilon=5$ | $\epsilon=10$
> -|-|-|-
> CIFAR-10| .85| .87| .90
> STL-10| .72| .79| .82
> CelebA| .78| .83| .89
> > **W3.** Computational and practical scalability is insufficiently discussed.
>
> We agree that the scalability discussion should be made more visible. The current manuscript already contains complexity analysis and empirical runtime comparisons in **Appendix C.5 Remark 3**, but this is not sufficiently emphasized in the main text.
>
> We also add further empirical evidence on computational cost via the runtime results below.
>
> Table B (Hours) | CIFAR-10|STL-10|CelebA
> -|-|-|-
> CovMatch|1|1.25|1.2
> CenterMatch| 0.10|0.12|0.11
> K-means|11.8|14.6|17.9
> Scheme I|0.19|0.22|0.20
> Scheme II|0.17|0.20|0.18
>
> Our method is computationally efficient because it operates only on the proxy measures, whose support size is small (typically 10 to 100). Notably, Scheme II only needs to be computed once for a fixed set of real-data shards, whereas the competing methods must be recomputed at every generation. This makes Scheme II particularly attractive in recursive training, where repeated selection costs otherwise accumulate over generations. In the revision, we will move the complexity discussion into the main text and add an explicit runtime table to make the computational trade-off clearer.
>
> ---
> **In the revision, we will make 3 concrete changes to clarify the scope and practical implications of our method:**
>
>  (1) explicitly scope Theorems 1&2 to the Gaussian analytical regime, while adding non-Gaussian simulations and visualizations in repository;
>
>  (2) clarify that the Wasserstein-gradient score is used as a first-order ranking signal, and that dual non-uniqueness mainly affects ranking stability in practice;
>
>  (3) move the complexity discussion into the main text and add an explicit runtime table across datasets.

---

> > ### Author Rebuttal · Reviewer_pAu7 · 2026-04-01
> >
> > I would like to thank the authors for their detailed responses. Most of my concerns have been resolved, except for those regarding the Gaussian regime. Due to my limited familiarity with the specific literature in this sub-field, I will maintain my current evaluation.

---

> > > ### Author Response · Authors · 2026-04-02
> > >
> > > We sincerely thank the reviewer for the thoughtful and supportive evaluation and for the time and effort devoted to reading the paper, both of which helped make our contributions clearer.
> > >
> > > Since part of our contribution builds on analyses from prior literature, we provide the following references for context.
> > >
> > > - Specifically, we build on the same **Multivariate Gaussian Analysis Framework** as prior influential works **[Ref1]** and **[Ref2]**. This allows us to study the problem in a more controlled and transparent setting, and to clearly isolate how biased selection (e.g., local preference feedback or alignment-style filtering) can destabilize a system that was previously proven to remain stable under iterative training. **[Ref4]** also reviews several collapse-theory frameworks beyond the Gaussian setting (e.g., **linear models with ridge regression**).
> > >
> > > - In addition, we would like to further clarify the role of the **Multivariate Gaussian Analysis Framework**. As stated in **[Ref3]**, under statistical assumptions and large-sample limits, recursive dynamics can be predicted within the Gaussian framework. Therefore, although some analyses may be fragile in the LLM era, the insights provided by the Multivariate Gaussian Analysis Framework give us reason to believe that, in some not-uncommon scenarios, the trend-level predictions of our theory are still useful and can offer insight for future work. We also provide empirical evidence consistent with these trends in both image generation (**Sec. 5**) and text generation (additional results provided in the anonymous repository during the rebuttal period).
> > >
> > >   Moreover, since some non-Gaussian empirical evidence collected during the rebuttal period also shows trends similar to those predicted by our analysis, we plan to extend these theoretical results in future work.
> > >
> > > ---
> > >
> > > **Ref1** [AI Models Collapse When Trained on Recursively Generated Data](https://www.nature.com/articles/s41586-024-07566-y). **Nature 2024**
> > >
> > > **Ref2** [Collapse or Thrive: Perils and Promises of Synthetic Data in Self-Generating World](https://openreview.net/forum?id=buwLCdOHxO). **ICML 2025**
> > >
> > > **Ref3** [Universality of π²/6 Pathway in Avoiding Model Collapse](https://arxiv.org/html/2410.22812). arXiv 2025
> > >
> > > **Ref4** [Surveying Model Collapse: Empirics, Universality, and Implications](https://arulandu.com/assets/pdf/am254-model-collapse.pdf). arXiv 2025

---

### Official Review · Reviewer_pHpz · 2026-03-12

**Soundness:** 3
**Presentation:** 2
**Significance:** 3
**Originality:** 3
**Overall Recommendation:** 4
**Confidence:** 3

**Summary:**

This paper investigates the phenomenon of model collapse in recursive synthetic data training, focusing on the impact of biased data selection in siloed environments. Using a multivariate Gaussian framework, the authors prove that applying a biased selection filter (modeled as a locally concave utility function) within the "Accumulate" paradigm destroys its theoretical stability, inducing variance collapse at an asymptotic power-law rate $\mathcal{O}(t^{-\lambda_{\min}(\Psi_\infty)})$. To mitigate this, the paper proposes two collaborative, optimal transport-based data selection schemes (Geodesic Interpolation and Wasserstein Barycenter). These schemes compute proxy distributions to approximate the global data manifold without sharing raw data, allowing local actors to score synthetic samples using Wasserstein dual potentials. Empirical results on CIFAR-10, STL-10, and CelebA demonstrate that the proposed methods preserve diversity and outperform standard selection baselines in non-IID settings.

**Compliance With Llm Reviewing Policy:**

Affirmed.

**Key Questions For Authors:**

* **[GAP] Local vs. Global Restoring Force in Theorem 1:** Assumption 1 states the utility function is *locally* concave around $\mathbf{u}^{\*}$. Consequently, Lemma 1 (Page 17) explicitly assumes "$\mu$ is in a local neighborhood of $\mathbf{u}^*$" to guarantee the restoring force $\kappa > 0$. However, the proof of Theorem 1 (Appendix B.4, Eq 90-91) applies this bound globally from $t=0$ to prove convergence. If the initial mean $\bar{\mu}_0$ is outside this local neighborhood, the drift might not act as a restoring force, and the Lyapunov function $V_t$ might not contract. Please clarify if global concavity is required, or if an explicit assumption about the initialization $\bar{\mu}_0$ falling within the local basin of attraction is missing.
* **[GAP] Rigor of the deterministic approximation in Theorem 2:** In Appendix B.5 (Page 29), the proof states: "In the asymptotic regime ($t \to \infty$), we analyze the deterministic equivalent of the trace recurrence... $S_t \approx S_{t-1}(1 - \frac{\psi}{t} - \frac{\kappa}{t^2})$" (Eq. 122). While the subsequent Gamma function expansion (Eq 123-128) is mathematically sound for a deterministic sequence, the leap from the stochastic trace $S_t$ to this deterministic sequence without bounding the martingale difference sequence (the stochastic fluctuations of the covariance estimator) or the approximation error of $\Psi_{t-1} \to \Psi_\infty$ is unproven. Can the authors replace this section with a rigorous bound derived directly from the supermartingale $M_t$ constructed in Appendix B.4?
* **Probabilistic Cross-Lipschitzness definition:** In Assumption 2 (Page 5), the paper assumes $(\epsilon, \delta)$-probabilistic cross-Lipschitzness between $g_t$ and $g^*$. While cited to Just et al. (2023), the exact mathematical definition is required in the main text or appendix to make the proof of Theorem 3 (specifically Eq. 139 and 140) self-contained.
* **Privacy leakage of the proxy:** In Scheme I, the proxy $\xi^{(r)}$ is transmitted between parties. Since $\xi^{(r)}$ is a discrete distribution (a set of support points and weights), does sharing it leak the exact feature vectors of the synthetic data? While Appendix C.6 discusses DPOT, the main text should briefly clarify the exact nature of the "proxy" being transmitted to avoid overclaiming privacy.
* Page 4, Line 209: "Mean Drift ($\lambda_{t-1}$): Defined as $\lambda_{t-1} \triangleq \mathbb{E}[\eta_{i,t}|\mathcal{F}_{t-1}]$ ".

The notation $\lambda$ is typically reserved for eigenvalues (as used later in $\lambda_{\min}$); consider using a different symbol for the mean drift to prevent notation overloading.
* Page 40, Figure 6 caption: Typo "thethe Replace Pradigm".
* Line71, Q1 has grammatical problem.

**Limitations:**

See weaknesses and problems.

**Strengths And Weaknesses:**

## Strengths

1. **Theoretical characterization of a realistic failure mode:** The extension of the Gaussian analysis framework to include biased selection in the Accumulate paradigm (Theorems 1 and 2) provides a rigorous explanation for why local verification fails in federated/siloed settings.
2. **Elegant martingale convergence proof:** The proof of Theorem 1 (Appendix B.4) cleverly constructs a supermartingale $M_t = S_t / P_t$ to establish the almost sure convergence of the covariance trace to zero, handling the anisotropic dynamics cleanly.
3. **Principled algorithmic design:** The translation of the theoretical Wasserstein generalization bound (Theorem 3) into a practical scoring mechanism using Kantorovich dual potentials (Eq. 18) and decentralized proxy estimation (Schemes I & II) is well-motivated and computationally analyzed.
4. **Comprehensive empirical validation:** The experiments effectively isolate the impact of data heterogeneity (Dirichlet $\alpha$) and demonstrate the specific failure modes of heuristic baselines (e.g., CenterMatch) compared to the proposed OT-based methods.

## Weaknesses

1. **Claim-guarantee alignment in Theorem 3:** Theorem 3 bounds the generalization risk using the global Wasserstein distance $\mathcal{W}_p(\mathcal{D}_t, \mathcal{D}^*)$. However, the proposed algorithms (Schemes I and II) perform a greedy, sample-wise selection based on first-order variations (Eq. 18) to a proxy distribution.
   * *Why it matters:* The text implies that the algorithm minimizes the bound in Theorem 3, but greedy selection on marginal dual potentials does not guarantee the minimization of the global Wasserstein distance, especially under a strict cardinality constraint $|I| \le n$.
   * *What to fix:* Add a clarifying remark after Theorem 3 or in Section 4.1 acknowledging that while Theorem 3 motivates aligning $\mathcal{D}_t$ with $\mathcal{D}^*$, the proposed gradient-based scoring is a first-order greedy approximation and does not come with a global optimality guarantee for the subset selection problem.
2. **Presentation of Computational Complexity:** The main text (Sections 4.3 and 4.4) claims that Scheme II overcomes the scalability limitations of Scheme I. However, the actual computational complexity analysis that substantiates this claim is entirely relegated to Appendix C (Remark 3, Page 42).
   * *Why it matters:* Scalability is the primary motivation for introducing Scheme II. Without stating the $\mathcal{O}(\cdot)$ bounds in the main text, the reader cannot evaluate the magnitude of the efficiency gain or the feasibility of the method.
   * *What to fix:* Move the final complexity bounds from Remark 3 ($\mathcal{O}(R \cdot L(N+M+S)S + nNK)$ vs. $\mathcal{O}(T LMS + LNS)$) into the main text of Section 4.4 to explicitly quantify the scalability improvement.

---

> ### Author Rebuttal · Authors · 2026-03-30
>
> We sincerely thank the reviewer for the thoughtful and supportive evaluation. We are also very grateful for the considerable time spent checking our proofs and definitions, which will help make our theory more robust. Below we provide clarifications and revisions to address your concerns. For clarity, **W** denotes Weakness and **Q** denotes Question.
> > **W1. Claim-guarantee alignment in Thm. 3.** Add a clarifying remark after Thm. 3 or in Sec. 4.1.
>
> We agree that Them 3 is a global risk bound, whereas Scheme I is a practical first-order selection rule under cardinality constraints. We do not claim that Scheme I directly minimizes the global Wasserstein objective. We thus will add an explicit clarification at the end of Sec. 4.4 to avoid any possible misunderstanding.
> > **W2. Presentation of Computational Complexity.** Move the final complexity bounds from Remark 3 into the main text of Sec. 4.4.
>
> Thank you for the suggestion. We will move the final complexity bounds into the main text of Sec. 4.4 to make scalability gain explicit. In addition, since the final version allows extra page, we will add more empirical evidence (preliminary results are already provided in our response to `Reviewer pAu7`'s **W3** due to  word limit) comparing the efficiency of different algorithms to make our statements more clear.
> > **Q1. Local vs. Global Restoring Force in Thm. 1.** Clarify if an assumption about initialization falling within the local basin of attraction is missing.
>
> We agree that an explicit assumption about the initialization falling within local basin of attraction is missing. This statement is required to ensure the effectiveness of selection mechanism. We will update Thm. 1 and App. B.4 to state this explicitly.
> > **Q2. Rigor of deterministic approximation in Thm 2.**
>
> We thank the reviewer again for the careful examination of the proof. We agree that current derivation based on deterministic surrogate is not sufficiently rigorous. Following your suggestion, we replace it with stochastic argument based on the supermartingale $M_t$ constructed in App. B.4. A proof sketch of revised App. B.5 is as follows.
>
> Let $S_t=\operatorname{Tr}(\bar\Sigma_t)$.
>
> From stochastic recurrence in App. B.4, we have $\mathbb E[S_t\mid \mathcal F_{t-1}] \le (1-\frac{\psi}{t})S_{t-1}$ with $0<\psi\le 1$. Define deterministic sequence $P_1=1;\ P_t:=P_{t-1}\left(1-\frac{\psi}{t}\right),\ t\ge 2$,
>
> We normalize $S_t$ by $P_t$ and define $M_t=S_t/P_t.$
> Since $P_t$ is deterministic and $S_t$ is $\mathcal F_t$-measurable, the process $\{M_t\}$ is adapted to $\{\mathcal F_t\}$. Moreover,
>
> $\mathbb E[M_t\mid \mathcal F_{t-1}]=\frac{1}{P_t}\mathbb E[S_t\mid \mathcal F_{t-1}]\le\frac{1}{P_t}(1-\frac{\psi}{t})S_{t-1}=S_{t-1}/P_{t-1}=M_{t-1},$
>
> Also, $M_t\ge 0$ as $S_t\ge 0$ and $P_t>0$. Hence $\{M_t\}$ is a nonnegative supermartingale. By Doob’s supermartingale convergence,
>
> $M_t \xrightarrow{a.s.} M_\infty<\infty.$
>
> Using $1-x\le e^{-x}$ and $\sum_{k=2}^{t}\frac{1}{k}=\log t+ O(1)$ as $t\to\infty$,
>
> $P_t=\prod_{k=2}^t(1-\frac\psi k)\le\exp(-\psi\sum_{k=2}^t\frac1k)=O(t^{-\psi}).$
>
> Consequently,
>
> $\text{Tr}(\bar\Sigma_t)=S_t=M_tP_t=\mathcal O_{a.s.}(t^{-\psi}).$
>
> This yields the claimed power-law upper bound directly from the stochastic inequality, without introducing a deterministic recurrence.
> > **Q3. Probabilistic Cross-Lipschitzness definition.** The definition is required in the main text or appendix to make the proof of Thm. 3 self-contained.
>
> We will provide the formal definition of probabilistic Cross-Lipschitzness in App. B.1 to make the proof self-contained.
> > **Q4. Privacy leakage of the proxy.**
>
> We agree that the privacy claim should be stated more precisely. Our intended claim is not that proxy sharing itself provides a formal privacy notion. In non-DP setting, the proxy only avoids raw-data exchange and instead communicates compressed proxy measures, and formal privacy guarantee is provided only when optional DP-OT layer is added. We will also add a privacy-utility ablation of privacy budget to evaluate the privacy-utility trade-off and to clarify the level of privacy protection achieved when a DP-OT layer is enabled.
> > **Q5,6,7.** (1) notation overloading for $\lambda$; (2) typo “thethe Replace Pradigm”; (3) grammatical problem.
>
> We thank the reviewer for catching these issues. We will rename the mean-drift notation $\lambda_t$ to $a_t$, fix the typo, and correct the grammar.
>
> ---
> ▪  **In the revision, we will further strengthen the rigor of the definitions and proofs to avoid any possible misunderstanding:**
>
>  (1) add missing local-basin assumption in Thm 1;
>
>  (2) replace deterministic approximation in Thm 2 with supermartingale-based proof;
>
>  (3) add formal definition of probabilistic Cross-Lipschitzness;
>
>  (4) clarify claim-guarantee relation behind Scheme I / Thm 3;
>
>  (5) move complexity bounds into Sec. 4.4;
>
>  (6) refineprivacy statement and add a privacy-utility ablation;
>
>  (7) fix notation, typo, and grammar.

---

> > ### Author Rebuttal · Reviewer_pHpz · 2026-04-03
> >
> > I do not have further questions.

---

> > > ### Author Response · Authors · 2026-04-03
> > >
> > > We sincerely thank the reviewer once again for the time and effort devoted to reviewing our paper, and for the supportive evaluation. Your helpful feedback has helped us further refine our analysis and clarify the presentation of our contributions.

---

### Official Review · Reviewer_ZDzC · 2026-03-18

**Soundness:** 3
**Presentation:** 3
**Significance:** 3
**Originality:** 3
**Overall Recommendation:** 5
**Confidence:** 4

**Summary:**

The paper tackles the existential risk of model collapse in recursive synthetic data training, specifically focusing on how sample selection bias in siloed environments accelerates this decay. While data selection is often proposed as a solution, the authors theoretically prove that local, biased filters in data silos drive diversity loss following a power law. To mitigate this, they propose a collaborative framework using Wasserstein geometry and geodesic interpolations (or Barycenters) to score synthetic data against a global reference without exchanging raw, private data.

**Compliance With Llm Reviewing Policy:**

Affirmed.

**Key Questions For Authors:**

1. MIA as a Proxy for Privacy: Could the authors provide empirical results using Membership Inference Attacks (MIA) to quantify exactly how much privacy is preserved by sharing geodesic interpolants versus raw synthetic data?

2. Distributed vs. Direct Performance: In Table 1, why is the performance of Scheme II (Barycenter) generally slightly lower (higher FID) than Scheme I (Geodesic)? Does this imply that the Barycenter is a coarser approximation of the global manifold, or are more communication rounds (R) required to reach the fidelity of pairwise geodesic interpolations?

**Limitations:**

yes

**Strengths And Weaknesses:**

Strengths

Strong Theoretical Results in Variety of Settings:

Diversity Decay Quantification: The authors prove that biased selection in the Accumulate paradigm causes variance to collapse at an asymptotic power-law rate.

Wasserstein Cost of Collapse: Theorem 3 bounds the generalization risk on the true manifold, showing that without global verification, error is dominated by the Wasserstein distance between the filtered synthetic distribution and the true data manifold.

Convergence Guarantees: They provide formal proofs (Theorems 4 and 5) that their collaborative geodesic and barycenter estimation schemes are monotonic and converge to the true global metrics.

Experimental Validation:

Empirical results across CIFAR-10, STL-10, and CelebA demonstrate that the proposed schemes (Scheme I & II) consistently achieve the best performance, outperforming baselines like CenterMatch and CovMatch.

The approach remains robust in both IID and non-IID (Dirichlet) partitions, where traditional baselines often fail to beat simple random sampling.

Weaknesses

Performance with Differential Privacy (DP): While the authors integrate a DP-OT framework, the quality of selection under strict privacy budgets is a concern. Although Figure 10(c) suggests gradients remain correlated at ϵ=1.0, the potential for accuracy degradation in more complex, high-dimensional manifolds remains an open question.

Weakly Defined Privacy Notion: The introduction highlights the necessity of privacy in hospitals and banks, but the "privacy" provided by barycentric compression is largely described as making reconstruction "ill-posed" rather than offering a formal cryptographic or statistical guarantee without the optional DP layer

---

> ### Author Rebuttal · Authors · 2026-03-30
>
> We sincerely thank the reviewer for the thoughtful and supportive evaluation, and for recognizing both the theoretical and empirical strengths of our work. Below, we clarify several details to address your concerns. For clarity, **W** denotes *Weakness*, **Q** denotes *Question*, and **R** denotes *Reference*.
>
> **Privacy (W1, W2, Q1**).
>
> > **W1.** Performance under strict privacy budgets may degrade, especially in complex or high-dimensional settings.
> >
> > **W2.** The privacy notion is weakly defined, since ill-posed reconstruction is not a formal privacy guarantee.
> >
> > **Q1.** Could the authors provide MIA results to quantify how much privacy is preserved by sharing geodesic interpolants versus raw synthetic data?
>
> Regarding **W1**, we agree that our current DP results do not yet fully characterize how utility degrades under stronger privacy constraints. To address this, we will add a privacy-utility ablation that includes the privacy budget and evaluates both the stability of the OT-based selection signal and the downstream selection quality across datasets and heterogeneity levels. Due to limited academic computational resources during the rebuttal period, we have completed a partial additional experiment for Scheme II for other datasets, reported in **Table A**. In the full revision, we will further report the Spearman correlation between sample scores with and without DP noise, so as to quantify how well the global ranking induced by the selection signal is preserved as privacy noise increases.
>
> Regarding **W2**, we agree that the privacy claim should be stated more precisely. Our intended claim is not that proxy sharing provides a formal privacy notion. In non-DP setting, the mechanism only avoids raw-data exchange and instead communicates compressed proxy measures. This reduces exposure, but it does not by itself constitute a statistical guarantee. In our framework, a formal privacy guarantee is provided only when the optional DP-OT layer is added. We will revise the introduction and methodology to make this separation explicit.
>
> Regarding **Q1**, we thank the reviewer for raising this highly meaningful question. We would like to clarify that a well-defined MIA in our setting requires a well-specified threat model and evaluation protocol. In our framework, the communicated object is a geodesic interpolant as in **R1**, rather than a raw example or a standard model output. Therefore, an interpretable MIA evaluation must specify what the attacker observes and what auxiliary knowledge is available. Although designing such an attack against the geodesic interpolant in **R1** would be an interesting direction for future work, it is beyond the scope of the current paper.
>
> Nevertheless, as a complementary formal perspective, we report DP results using a fixed privacy budget on all datasets in **Table A**. While DP is not a substitute for an empirical MIA benchmark, prior work has shown that it provides formal bounds relevant to membership inference and has been studied as a principled defense or risk-control mechanism against MIA **[R2, R3]**.
>
> | Table A  | FID  | Spearman |
> | -------- | ---- | -------- |
> | CIFAR-10 | 96   | 0.85     |
> | STL-10   | 89   | 0.72     |
> | CelebA   | 84   | 0.78     |
>
> ---
>
> **Methodology Analysis (Q2).**
>
> > **Q2.** Why is Scheme II generally slightly worse than Scheme I in Table 1? Is barycenter a coarser approximation of global manifold, or are more communication rounds required?
>
> We thank the reviewer for this insightful question. Intuitively, when local distributions have weak overlap or substantial noise and barycenter estimation is sensitive to regularization and optimization, the resulting proxy can become overly smooth and lose fine local information, which is particularly detrimental in strongly non-IID settings. By contrast, Scheme I retains client-specific geodesic proxies and therefore preserves heterogeneous local modes more explicitly. Additional communication rounds are unlikely to be the main cause here, since both our theory and experiments indicate that the barycenter converges within few rounds. We will revise the discussion around Table 1 to clarify this point.
>
> ---
>
> ▪  **In the revision, we will make 3 concrete changes to avoid any possible misunderstanding:**
>
>  (1) explicitly distinguish reduced exposure through proxy sharing from formal privacy guarantees in the main text;
>
>  (2) add a privacy-utility ablation over the different privacy budget in App. C.6 to characterize when the OT-based selection signal remains stable; and
>
>  (3) clarify in App. C.5 that the gap between Scheme I and II is due to the practical sensitivity of barycenter estimation to heterogeneity, regularization, and noise, rather than insufficient rounds.
>
> ---
> **R1** Federated Wasserstein Distance. ICLR 2024.
>
> **R2** From Differential Privacy to Bounds on Membership Inference: Less can be More. TMLR 2024.
>
> **R3** Attack-Aware Noise Calibration for Differential Privacy. NeurIPS 2024.

---

> > ### Author Rebuttal · Reviewer_ZDzC · 2026-04-04
> >
> > The authors addressed all my comments and I will maintain my score.

---

> > > ### Author Response · Authors · 2026-04-05
> > >
> > > We sincerely appreciate your valuable comments, which have significantly improved the clarity of our manuscript and further strengthened its contributions.

---

### Official Review · Reviewer_h5Ke · 2026-03-24

**Soundness:** 2
**Presentation:** 2
**Significance:** 2
**Originality:** 2
**Overall Recommendation:** 2
**Confidence:** 4

**Summary:**

This work utilizes Wasserstein geometry to overcome the sample selection bias problem. Based on some assumptions, the authors provide some theoretical results, and the effectiveness is proved by some toy experiments.

**Compliance With Llm Reviewing Policy:**

Affirmed.

**Key Questions For Authors:**

Please refer to the weaknesses part.

**Limitations:**

Please refer to the weaknesses part.

**Strengths And Weaknesses:**

Strength:
- The theoretical results are solid.

Weaknesses:
- Assumption aspect: The authors claim to solve a realistic problem, and the problem setting or story (from introduction) seems practical. However, through the theoretical analysis, the assumptions are over-idealized (Gaussian assumptions, locally concave utility functions). I understand that they are common assumptions for theoretical analysis, but they are quite limited in wild.
- Small-scale models and datasets: The datasets and models used are old. As the authors mentioned generative models, can this method be useful in LLM domains, which I think is the most useful generative model recently. And again, in this domain, are the assumptions still hold? Any evidence?
- Experiment settings: The paper's core motivation is addressing data silos and data scarcity. Yet, in their experimental setup, they note that partitioned data yields weak initial models, so they "first utilize the training set of size n=50,000 to establish a strong generator". If the collaborating institutions already possess 50,000 high-quality real images upfront to train a robust base model , the initial premise of extreme data fragmentation and scarcity is somewhat undermined. Therefore, I think the authors overclaimed the practical value of this method.

---

> ### Author Rebuttal · Authors · 2026-03-30
>
> We sincerely thank the reviewer for the constructive comments. Below are our clarifications. For clarity, **W** denotes Weakness and **R** denotes Reference.
>
> > **W1.** Theoretical analysis relies on idealized assumptions
>
> - **Our goal is to characterize the failure boundary of existing model collapse theory.** We thus intentionally build on the same framework as prior influential works **R1** and **R2**, so that the comparison is controlled: when selection is introduced, under what conditions does the existing stability guarantee break down?
> - **Our analysis is predictive rather than fully descriptive of all settings.** As stated in **R3**, under statistical assumptions and large sample limits, recursive dynamics can be predicted within Gaussian framework. Our experiments are also consistent with the analysis, e.g., power law diversity decay in Thm. 2 & Fig. 4. Moreover, Thm 3 provides a more general Wasserstein-based risk perspective without over-idealized assumptions.
> - **Nevertheless, we empirically conducted non-Gaussian simulations,** and the trend remains consistent with our prediction. Due to words limits, we placed visualizations in  **[anonymous repository](https://anonymous.4open.science/r/When-Sample-Selection-Bias-Precipitates-Model-Collapse-D8D4)** and updated repo for reproducibility. We will incorporate the results into revision.
>
> ▪ **In the revision, we will revise App. A&C to clearly state purpose and limitations of assumptions, and to distinguish predictive role of theory from descriptive account of all practical recursive training settings to highlight our contributions.**
>
> ---
> > **W2.** The datasets and models are old and small scale; it is unclear whether method is useful in LLM
>
> - **We adopted the well-established setup in R4 & R5,** both of which study collapse with diffusion models. Specifically, **R4** evaluated DDPM based only on CIFAR10 and **R5** evaluated on CIFAR10, FFHQ, MNIST. We follow the same line, using CIFAR10, FFHQ, CelebA.
> - **We agree that LLMs are an important setting. To address the concern, we extend text experiments** to cover both image& text modalities.
>
> Following **R6**, we fine-tune Llama-2-7B on English subset of XLSUM. We partition XLSUM by topic so that each entity only has access to 1 topical subset (e.g. tech.) as local data. We then filter generated data using only tech topic and evaluate ROUGE-1 on non-tech topics. We compare with the ROUGE selection strategy in **R6**, which utilizes ROUGE to quantify alignment between generated outputs and real data
> TableA|Tech|Non-Tech
> -|-|-
> Random|.29|.27
> ROUGE|.32|.24
> Scheme I|.31|.29
>
> TableA shows that even ROUGE strategy, which is referred to in **R6** as oracle selection and used as performance upper bound, degrades on non-tech topics. We also include more visualizations in `repo`, showing that selecting based on English subset alone can also degrade performance on underrepresented languages. These results suggest that a topic-local verifier can improve in-topic alignment while hurting broader coverage;
>
> ▪ **In the revision, we will incorporate these text-generation results and discuss the same failure mode in LLM-based recursive training.**
>
> ---
> > **W3.** Using 50k images to establish a strong generator seems to undermine data scarcity motivation
>
> - **Our claim is about the failure of selection under siloed verification, rather than about training a generator from scratch under extreme scarcity.** In fact, in many settings, one does not pretrain a model from scratch within each institution. Instead, one starts from a strong pretrained foundation model, and then performs subsequent finetuing or alignment using limited data or local preference feedback, and our setup is intended to capture this stage.
> - **Our goal is to explore whether a strong generator remains stable under biased verification during recursive training.** This setting is practically relevant: even a generator with broad initial coverage can still suffer severe diversity collapse under siloed filtering, as shown in Fig. 4.
> - **We agree weak initializer case should be ablated. We thus add weak generator experiment** on CIFAR-10 using a 5k sample initializer. The same effect appears as shown in Table B.
> TableB|FID
> -|-
> CenterM.|151
> CovM.|144
> Random|142
> Scheme I|109
>
> ▪ **In the revision, we will strengthen motivation for strong-initializer setting and add weak-initializer ablations on more datasets in App. C.**
>
> ---
> **R1** AI Models Collapse When Trained on Recursively Generated Data. Nature2024
>
> **R2** Collapse or Thrive: Perils and Promises of Synthetic Data in Self-Generating World. ICML2025
>
> **R3** Universality of π²/6 Pathway in Avoiding Model Collapse
>
> **R4** Self-Consuming Generative Models with Curated Data Provably Optimize Preferences. NIPS2025
>
> **R5** A Closer Look at Model Collapse: From a Generalization-to-Memorization Perspective. NIPS2025
>
> **R6** Beyond Model Collapse: Scaling up with Synthesized Data Requires Verification. ICLR2025

---

### Decision · Program_Chairs · 2026-04-30

**Decision:**

Accept (regular)

**Comment:**

In the theoretical contribution of the work, multivariate Gaussian settings are studied. The use of Wasserstein geometry to overcome the sample selection bias is appreciated by all the reviewers. The theoretical analysis applies to canonical and simplified scenarios, which is typical of theoretical works.

The experiments are more for a proof of concept, and are smaller scale and simpler tasks. In response, the authors added a Llama-2-7B experimental results.